# Convex Efficient Coding

**William Dorrell, Peter E. Latham**[*]
Gatsby Computational Neuroscience Unit
University College London
dorrellwec@gmail.com

**James Whittington**[*]
Department of Experimental Psychology
University of Oxford
* Co-Senior Authors

## Abstract

Why do neurons encode information the way they do? Normative answers to this question model neural activity as the solution to an optimisation problem; for example, the celebrated efficient coding hypothesis frames neural activity as the optimal encoding of information under efficiency constraints. Successful normative theories have varied dramatically in complexity, from simple linear models (Atick & Redlich, 1990), to complex deep neural networks (Lindsay, 2021). What complex models gain in flexibility, they lose in tractability and often understandability. Here, we split the difference by constructing a set of tractable but flexible normative representational theories. Instead of optimising the neural activities directly, following Sengupta et al. (2018), we optimise the representational similarity, a matrix formed from the dot products of each pair of neural responses. Using this, we show that a large family of interesting optimisation problems are convex. This family includes problems corresponding to linear and some non-linear neural networks, and problems from the literature not previously recognised as convex, such as modified versions of semi-nonnegative matrix factorisation or nonnegative sparse coding. We put these findings to work in three ways. First, we provide the first necessary and sufficient identifiability result for a form of semi-nonnegative matrix factorisation. Second, we show that if neural tunings are 'different enough' then they are uniquely linked to the optimal representational similarity, partially justifying the use of single neuron tuning analysis in neuroscience. Finally, we use the tractable nonlinearity of some of our problems to explain why dense retinal codes, but not sparse cortical codes, optimally split the coding of a single variable into ON and OFF channels. In sum, we identify a space of convex problems, and use them to derive neural coding results.

## 1 Introduction

Neural activity forms the substrate of intelligence in both brains and artificial neural networks. Towards understanding these systems, much work has therefore tried to frame and solve neural coding puzzles; for example, is neural activity a temporal or a rate code (Gautrais & Thorpe, 1998), or why do we find Gabor filters in both visual cortex (Jones & Palmer, 1987) and Convolutional Neural Networks trained to recognise objects (Yosinski et al., 2014). Normative approaches provide compelling models. They frame neural responses as the solutions to optimisation problems, the most classical of which is the efficient coding hypothesis (Attneave, 1954; Barlow, 1961); loosely, 'neurons use the representation that most efficiently encodes the required information'. Finding a match between optimal encodings and the brain provides evidence that these neurons can be thought of as solving the proposed optimisation problem, naturally leading to predictions and broader functional theories. These approaches have been successful, for example in studying tuning curves (Laughlin, 1981).

However, the tractability of these theories varies significantly. Some problems admit pleasing analytic solutions, outlining clearly how representations arise from the optimisation problem (Atick & Redlich, 1992). More often this is not the case. For example, neural activity is often modelled using task-optimised neural networks (Hinton et al., 2006; Yamins et al., 2014; Lindsay, 2021; Kell et al., 2018; Botvinick & Plaut, 2006; Wang et al., 2018; Zipser & Andersen, 1988; Tanaka, 2016; Goldstein et al., 2022; Gauthier & Levy, 2019), which are largely analytically impregnable. Even

successful simpler models like sparse coding (Olshausen & Field, 1996) or nonnegative matrix factorisation (Lee & Seung, 1999) are not fully understood (for example, it is not known under what conditions sparse coding generates Gabor filters). These complexities hinder potential insights.

A lot of work therefore attempts to 'open the black box', either by studying trained networks (e.g. (Sussillo & Barak, 2013)), or using simpler tractable models. Here, we build on a recent example of a tractable optimisation problem. Following a long line of related work (Pehlevan & Chklovskii, 2019), Sengupta et al. (2018) study a similarity matching objective—which loosely measures the alignment between the similarity of a pair of neural encodings and the similarities of their corresponding inputs—and show that the optimal nonnegative representations of compact variables are place cells if there are unlimited neurons. The technical novelty of their approach lies in showing that the optimisation problem can be written as a convex optimisation problem over the set of representational dot-product similarity matrices (the matrices of dot-products of neural representations in each task condition). In so doing, Sengupta et al. (2018) unlock the application of convexity to cutting-edge problems in neuroscience. However, this approach has not been extensively used, perhaps due to the bespoke nature of the similarity matching objective.

In this work, we broaden this approach. In section 2, we show that we can write a large family of objectives and constraints as convex over the set of representational dot-product similarity matrices. Arbitrary sets of convex constraints and objectives can then be composed to create a variety of interesting optimisation problems, each of which are convex. Many of these problems correspond to linear, or even nonlinear, neural networks with different regularisation schemes. Indeed, using these components, some problems from the literature can be reframed as convex problems. In this paper, we study three of these problems of relevance to AI and neuroscience.

First, in section 3, we study an existing problem not previously recognised as convex: nonnegative-affine autoencoding of a set of sources. Previous works studied orthogonally embedded sources and derived a set of necessary and sufficient conditions such that, optimally, each latent neuron encodes a single source, and used this to model patterns of modularity and mixed-selectivity in the entorhinal cortex (Whittington et al., 2023; Dorrell et al., 2025). We generalise to the case of linearly (rather than orthogonally) mixed sources. This change seems small, but in so doing, we derive the first necessary and sufficient conditions for the identifiability of related matrix factorisation problems.

Second, in section 4, we use the same theory to link population-level representational similarity to single neuron tuning curves. We derive conditions under which a given representational similarity matrix is created by a unique set of single-neuron tuning curves. Naturally one may expect many tuning curves to be associated with a given representational similarity, since representations can often be rotated arbitrarily, scrambling the neural tuning curves, without changing representational similarity or task performance. If this were the case, looking at the activity of single neurons would not be a useful signal for inferring the function of a neural population. Here we show that this is broadly not the case, since the convex constraint of neural activity being non-negative breaks rotational symmetry. For problems with this constraint (which applies to biological neurons), we derive a set of sufficient conditions that outline how, if the tuning curves are 'different enough', all optimal solutions will contain the same neural responses. In so doing, we provide justification for studying tuning curves, linking them precisely to the representation's optimality.

Finally, in section 5 we use our tractable convexity on a *non-linear* problem to answer a neural coding puzzle. Retinal neurons display ON-OFF splitting, in which a single variable is encoded in a pair of oppositely rectified neurons (Euler et al., 2014). This splitting has been understood as an efficiency: splitting the stimulus reduces the range of each neuron, saving energy (Sterling & Laughlin, 2015; Gjorgjieva et al., 2014). But not all variables are ON-OFF coded and existing theories cannot explain what governs this. We show that the transition between ON-OFF and pure coding is driven by the variable's sparsity, matching conjecture (Sterling & Laughlin, 2015).

In sum, we greatly expand the set of problems amenable to convex analysis and apply these results to coding puzzles and identifiability results relevant to both AI researchers and neuroscientists alike.

## 2 A FAMILY OF CONVEX REPRESENTATIONAL OPTIMISATIONS

In this section we establish the convexity of a series of representational optimisation problems, beginning with a simple example.

## 2.1 A Motivating Example

We consider optimisation problems over representations — $d_z$-dimensional vectors of neural activity for each of $N$ datapoints: $\boldsymbol{z}^{[i]} \in \mathbb{R}^{d_z}$, $i = 1, ..., N$. For example, we might require a set of targets, $\boldsymbol{y}^{[i]} \in \mathbb{R}^{d_y}$ to be linearly decodable from our representation,

$$\boldsymbol{y}^{[i]} = \boldsymbol{W}_{\text{out}} \boldsymbol{z}^{[i]}. \tag{1}$$

There are many feasible $\{\boldsymbol{z}^{[i]}\}_{i=1}^{N}$ and $\boldsymbol{W}_{\text{out}}$, and so for neurobiological realism and interpretability we additionally constrain the neural activity to be nonnegative ($\boldsymbol{z}^{[i]} \geq 0$), and penalise the energy use, either through firing rates or synaptic weights, the latter in line with evidence that the largest energy cost of spiking is synaptic transmission (Harris et al., 2012). We use the L2 norm for mathematical convenience, but consider a modified L1 activity loss in appendix A. Hence our problem,

$$\min_{\boldsymbol{W}_{\text{out}}, \{\boldsymbol{z}^{[i]}\}_{i=1}^{N}} \left( \langle ||\boldsymbol{z}^{[i]}||^2 \rangle_i + \lambda ||\boldsymbol{W}_{\text{out}}||_F^2 \right) \quad \text{subject to} \quad \boldsymbol{z}^{[i]} \geq 0, \quad \boldsymbol{W}_{\text{out}} \boldsymbol{z}^{[i]} = \boldsymbol{y}^{[i]}. \tag{2}$$

Where we use the physicists notation $\langle \rangle$ for average. This is a simple instantiation of the efficient coding hypothesis; $\boldsymbol{W}_{\text{out}} \boldsymbol{z}^{[i]} = \boldsymbol{y}^{[i]}$ ensures $\boldsymbol{z}^{[i]}$ encodes information about $\boldsymbol{y}^{[i]}$, and subject to this we maximise the efficiency by minimising the energy loss over both representation, $\boldsymbol{z}^{[i]}$, and weights, $\boldsymbol{W}_{\text{out}}$. Yet, even simple models like this have widespread use in neuroscience (Dordek et al., 2016; Sorscher et al., 2019; Martín-Sánchez et al., 2025; Huang et al., 2026). As written, this problem is not convex[1]. We use a classic approach to convexification (Shor, 1987; Lasserre, 2009; Anstreicher, 2012; Pena et al., 2015) following Sengupta et al. (2018), and find that, if $d_z > N$, problems like these can be framed as convex optimisations over the set of representational dot-product similarity matrix, defined as:

$$(\boldsymbol{Q})_{ij} = (\boldsymbol{z}^{[i]})^T \boldsymbol{z}^{[j]}. \tag{3}$$

We can show this by showing each of the constraints and functions are convex:

- *Firing Cost:*

$$\langle ||\boldsymbol{z}^{[i]}||^2 \rangle_i = \frac{1}{N} \sum_i (\boldsymbol{z}^{[i]})^T \boldsymbol{z}^{[i]} = \frac{1}{T} \text{Tr}[\boldsymbol{Q}]. \tag{4}$$

  This is a linear, and hence convex, function of $\boldsymbol{Q}$.

- *Weight Cost:* Since the representation is linearly decodable, the min-norm choice of readout weight matrix is given by the pseudoinverse. Defining the data and representation matrices:

$$\boldsymbol{Y} \in \mathbb{R}^{d_y \times N}, [\boldsymbol{Y}]_{:,i} = \boldsymbol{y}^{[i]}, \qquad \boldsymbol{Z} \in \mathbb{R}^{d_z \times N}, [\boldsymbol{Z}]_{:,i} = \boldsymbol{z}_i, \qquad \boldsymbol{W}_{\text{out}} = \boldsymbol{Y} \boldsymbol{Z}^{\dagger}. \tag{5}$$

  where $\boldsymbol{Z}^{\dagger}$ denotes the pseudoinverse. Then the weight cost becomes:

$$||\boldsymbol{W}_{\text{out}}||_F^2 = \text{Tr}[\boldsymbol{W}_{\text{out}}^T \boldsymbol{W}_{\text{out}}] = \text{Tr}[\boldsymbol{Y}^T \boldsymbol{Y} \boldsymbol{Z}^{\dagger} \boldsymbol{Z}^{\dagger,T}] = \text{Tr}[\boldsymbol{Y}^T \boldsymbol{Y} \boldsymbol{Q}^{\dagger}]. \tag{6}$$

  We show in appendix A that is a convex function of $\boldsymbol{Q}$.

- *Nonnegativity:* The set of dot-product matrices of nonnegative vectors ($\boldsymbol{Q} = \boldsymbol{Z}^T \boldsymbol{Z}$, $\boldsymbol{Z} \geq 0$) form a convex set called the set of completely positive matrices (Berman & Shaked-Monderer, 2003), as in Sengupta et al. (2018).

- *Decodability:* Requiring that the targets can be linearly decoded from the representation limits the set of allowed similarity matrices to those in which there is some subspace encoding the labels, a set we also show is convex in appendix A.

Each objective and constraint is convex, and combinations of convex functions and sets are convex (Boyd & Vandenberghe, 2004), hence the problem is convex. This ensures that all locally optimal representational dot-product similarity matrices, $\boldsymbol{Q}$, are globally optimal. This will prove theoretically useful. However, as we explain in section 6, optimising over the set of completely positive matrices is NP-hard (Dickinson & Gijben, 2014) stopping us from developing practical convex optimisation algorithms.

---

[1]For example, consider two feasible solutions that differ by a neuron swap:

$$\boldsymbol{W}_{\text{out}} \boldsymbol{z}^{[i]} = \begin{bmatrix} 1 & 0 \\ 0 & 1 \end{bmatrix} \begin{bmatrix} y_1^{[i]} \\ y_2^{[i]} \end{bmatrix}, \quad \tilde{\boldsymbol{W}}_{\text{out}} \tilde{\boldsymbol{z}}^{[i]} = \begin{bmatrix} 0 & 1 \\ 1 & 0 \end{bmatrix} \begin{bmatrix} y_2^{[i]} \\ y_1^{[i]} \end{bmatrix}, \quad \frac{\boldsymbol{W}_{\text{out}} + \tilde{\boldsymbol{W}}_{\text{out}}}{2} \frac{\boldsymbol{z}^{[i]} + \tilde{\boldsymbol{z}}^{[i]}}{2} = \begin{bmatrix} \frac{1}{2}(y_1^{[i]} + y_2^{[i]}) \\ \frac{1}{2}(y_1^{[i]} + y_2^{[i]}) \end{bmatrix}.$$

As shown by the last equation, despite both solutions satisfying eq. (1), their convex combination does not.

## 2.2 A Family of Convex Problems

In appendix A we present a set of representational constraints and objectives and show that each is convex over the set of representational similarity matrices. Further, we combine them to construct a series of problems, a few of which we now highlight.

**Regularised Linear/Affine Networks** Linear/Affine neural networks have proved to be popular tractable models in neuroscience and machine learning; for example in studying learning dynamics Saxe et al. (2013); Braun et al. (2022). With L2 weight regularisation and sufficient width, we show the optimisation problem posed by these networks is convex. Adding a nonnegativity constraint breaks rotational symmetry, making the neuron basis meaningful and allowing one to use these networks to ask neural tuning questions. For example, Whittington et al. (2023); Dorrell et al. (2025) use these models to reason about why neural recordings in the entorhinal cortex are sometimes modular, with disjoint sets of neurons encoding space and reward, and sometimes mixed-selective, with the same neurons showing tuning to both space and reward. While nonnegativity makes the optimisation problem over $Z$ non-convex, when reframed over the dot-product matrix $Z^T Z$ the problem becomes convex, which we use in section 3 to derive a novel tight identifiability criterion.

**Tractable Nonlinear Problems** An appealing feature of Sengupta et al. (2018) is its ability to tractably model nonlinear tuning curves, by optimising a nonnegative representation to minimise a similarity matching loss ($\text{Tr}[GQ]$ for a positive-definite matrix $G$). We extend this, by showing that optimisations over nonlinear, but linear(affine)-decoded representations, as in section 2.1, are convex. These models have been used in neuroscience to model grid cells (Dordek et al., 2016; Sorscher et al., 2019; 2023; Schøyen et al., 2023; Tang et al., 2024) (via the nonnegative PCA), place field remapping (Martín-Sánchez et al., 2025), or zebrafish visual responses (Huang et al., 2026), and we use them in section 5 to model retinal coding. Further, they correspond to classic sparse coding or matrix factorisation approaches that have found success in modelling Gabor filters (Olshausen & Field, 1996) and learning parts-based representations (Lee & Seung, 1999). Future work could therefore use the uncovered convexity as analytic traction to understand these phenomena. In appendix A, we also show that the arbitrary nonlinearity can be replaced by a ReLU, allowing us to show that wide, regularised one-hidden layer ReLU networks are convex, a result which seems similarly promising, and related to recent work (Pilanci & Ergen, 2020; Zeger & Pilanci, 2025; Wang et al., 2025) (see section 6 for further comparison). Finally, appendix A also shows that even the affine readout constraint can be dropped, by optimising a nonlinear similarity matching loss– $\text{Tr}[Se^Q]$ for positive-definite $S$–which has been previously used as a model of multifield place cells (Dorrell et al., 2023). The flexibly nonlinear input and output mappings implicit in this problem suggest an interesting model for studying the internal representations of deep networks, such as nonlinear disentangling autoencoders (Higgins et al., 2017).

## 3 IDENTIFIABILITY OF SEMI-NONNEGATIVE MATRIX FACTORISATION

We now reframe a semi-nonnegative matrix factorisation algorithm as convex optimisation, permitting us to derive necessary and sufficient conditions under which the 'true' factors are recovered.

**Background** Matrix factorisation problems seek to break a matrix, such as the label matrix, $Y \in \mathbb{R}^{d_Y \times N}$, into two meaningful parts $Y = AS$, $A \in \mathbb{R}^{d_Y \times d_s}$, $S \in \mathbb{R}^{d_s \times N}$. For example, dictionary learning seeks to learn a dictionary, $A$, and sources, $S$, that fit the data while using a sparse $S$. In general, many choices of $A$ and $S$ can fit $Y$. For example, given one feasible pair $(A, S)$ such that $Y = AS$, inserting any invertible $d_s \times d_s$ matrix, $B$, will generate another feasible pair, $(AB, B^{-1}S)$, since: $ABB^{-1}S = AS = Y$. Therefore, this problem is only usefully posed in settings with more structure, either via constraining the allowed factorisations or regularising their choice. For example, *semi-nonnegative matrix factorisation* problems constrain one of the matrices to be nonnegative, while sparse coding uses an L1 regularisation on the source matrix.

**Identifiability** Identifiability results outline when a problem is well-posed. By well-posed we mean the following. First, the data-generating assumptions are true, so in the case of matrix factorisation problems, that the data really is linearly generated from two sub-matrices (otherwise how do we know what the model 'should' do?). Second, the proposed algorithm will recover the true data-generating factors. Previous work has extensively studied the identifiability of matrix factori-

sation, as we'll review later. However, to the best of our knowledge, all results are either necessary or sufficient, but not both. Here, we use our convexity results, section 2, to derive necessary and sufficient identifiability conditions for a neural network based factorisation algorithm with applications in theoretical neuroscience: nonnegative-affine autoencoders.

**Nonnegative Affine Autoencoding**   We derive such an identifiability result for a particular matrix factorisation problem, nonnegative affine autoencoding, which has been previously used as a model of neural data (Whittington et al., 2023; Dorrell et al., 2025). Data is generated via $X = AS$ and fed to the network: an autoencoder with two affine layers and a non-negativity constraint on the hidden layer activity. The network weights are optimised to perfectly reconstruct $X$ while minimising the L2 weight and activity norms. We ask when, up to a constant shift, the data-generating factors $A$ and $S$ are recovered in the optimal weights, $W_{\text{out}}$, and activities, $Z$, of the autoencoder. In other words, we look for identifiability conditions under which $Z = \Pi S + b\mathbf{1}$, meaning the latents recover the sources up to an arbitrary offset $b$, and a 'rectangular permutation' matrix, $\Pi$, with at most one non-zero entry in each row (and at least $d_s$ non-zero entries). Previous work has derived conditions when $A$ is an orthogonal matrix, and used this to understand patterns of modularity and mixed-selectivity in the entorhinal cortex (Whittington et al., 2023; Dorrell et al., 2025). Here, we use the convexity of the optimisation over the set of representational similarity matrices, $Q = Z^T Z$, section 2, to derive identifiability results for the case of arbitrary $A$. This problem is formalised below.

**Problem 1** (Nonnegative Affine Autoencoding). *We are provided a dataset, $\{x^i\}_{i=1}^N$, $x^{[i]} \in \mathbb{R}^{d_x}$, generated linearly from a dataset of sources $\{s^{[i]}\}_{i=1}^N$, $s^{[i]} \in \mathbb{R}^{d_s}$, $d_x > d_s$, and a full-column-rank mixing matrix $A \in \mathbb{R}^{d_x \times d_s}$: $x^{[i]} = As^{[i]}$. We use weights, $W_{\text{in}} \in \mathbb{R}^{d_z \times d_x}$ and $W_{\text{out}} \in \mathbb{R}^{d_x \times d_z}$, and biases, $b_{\text{in}} \in \mathbb{R}^{d_z}$ and $b_{\text{out}} \in \mathbb{R}^{d_x}$, to define a nonnegative affine-encoding of this dataset, $\{z^{[i]}\}_{i=1}^N$, $z^{[i]} \in \mathbb{R}^{d_z}$, $d_z > d_x$ using the following constrained optimisation problem :*

$$\min_{W_{\text{in}}, b_{\text{in}}, W_{\text{out}}, b_{\text{out}}} \langle ||z^{[i]}||_2^2 \rangle_i + \lambda \left( ||W_{\text{in}}||_F^2 + ||W_{\text{out}}||_F^2 \right),$$

$$\text{s.t.} \quad z^{[i]} = W_{\text{in}} x^{[i]} + b_{\text{in}}, \; x^{[i]} = W_{\text{out}} z^{[i]} + b_{\text{out}}, \; z^{[i]} \geq 0. \tag{7}$$

Previous work using orthogonal $A$ found that identifiability in this model is governed by the 'spread' of the sources. If the distribution of sources is 'sufficiently rectangular' the optimal representation recovers the sources (Dorrell et al., 2025). Precisely, if the convex hull of the data engulfs a particular easily-calculable set it is sufficiently rectangular, fig. 1A, and the optimal solution recovers the sources; otherwise it mixes them.

Returning to arbitrary $A$ complicates things. If the encodings of two sources are aligned, i.e. $A_i^T A_j$ is nonzero, where $A_i$ is the $i$th column of $A$, then weight regularisation encourages the encodings of those sources to align. This effect can cause the optimal solution to mix sources when it would otherwise modularise, or even modularise when it would otherwise mix[2], fig. 1. Our result improves the 'sufficient rectangularity' of previous work by adapting it appropriately to the mixing, $A$. In particular, we measure this adaptive rectangularity with a tight scattering condition using a quadratic form calculated from the dataset covariance and minima, and the mixing gram matrix $A^T A$.

**Definition 1** (Tight Scattering). *Given the same sources, $\{s^{[i]}\}_{i=1}^N$, and mixing matrix, $A$, as problem 1, Generate the mean-zero sources, $\bar{s}_d^{[i]} = s_d^{[i]} - \langle s_d^{[i]} \rangle_i$, and assume, without loss of generality, that $|\min_i \bar{s}_d^{[i]}| \leq \max_i \bar{s}_d^{[i]}$ for each $d$ (if not satisfied simply redefine $s_d$ as $-s_d$). Stack the mean-zero sources into a matrix, $\bar{S} \in \mathbb{R}^{d_S \times N}$ with elements $\bar{S}_{di} = \bar{s}_d^{[i]}$ and construct the diagonal $D \in \mathbb{R}^{d_S \times d_S}$ and the symmetric $F \in \mathbb{R}^{d_S \times d_S}$ matrices:*

$$D_{jj} = \sqrt{\frac{\langle (\bar{s}_j^{[i]})^2 \rangle_i + (\min_i \bar{s}_j^{[i]})^2 + \lambda((A^T A)^{-1})_{jj}}{\lambda(A^T A)_{jj}}}, \quad F = \lambda D(A^T A)D - \lambda(A^T A)^{-1} - \Sigma \tag{8}$$

*where $\Sigma = \frac{1}{T}\bar{S}\bar{S}^T$ is the covariance. Use these matrices to construct the following set:*

$$E = \{x | x^T F^{-1} x = 1\} \tag{9}$$

---

[2]This arises because the 'range effects' explored in previous work, and these weight effects are both signed: if $A_i^T A_j$ is positive (negative) the weight loss encourages the source encodings to positively (negatively) align. If range and weight effects misalign the modular solution can be optimal even if it isn't using orthogonal $A$.

*The sources are tightly scattered with respect to $A$ if the following conditions hold:*

- $Conv(\bar{S}) \supseteq E$

- $Conv(\bar{S})^* \cap bdE^* = \{\lambda e_k, \lambda \neq 0 \in \mathbb{R}, k = 1, ..., d_S\}$ *where the $*$ denotes the dual cone, bd the boundary, and $e_k$ the $k^{th}$ canonical basis vector, $(e_k)_l = \delta_{lk}$.*

The first condition ensures that the convex hull of the sources, $\text{Conv}(\bar{S})$, engulfs the $A$-dependent ellipse. The second is a technical condition that ensures the boundary of the ellipse, $bdE$, and the convex hull of the sources only touch along basis directions.

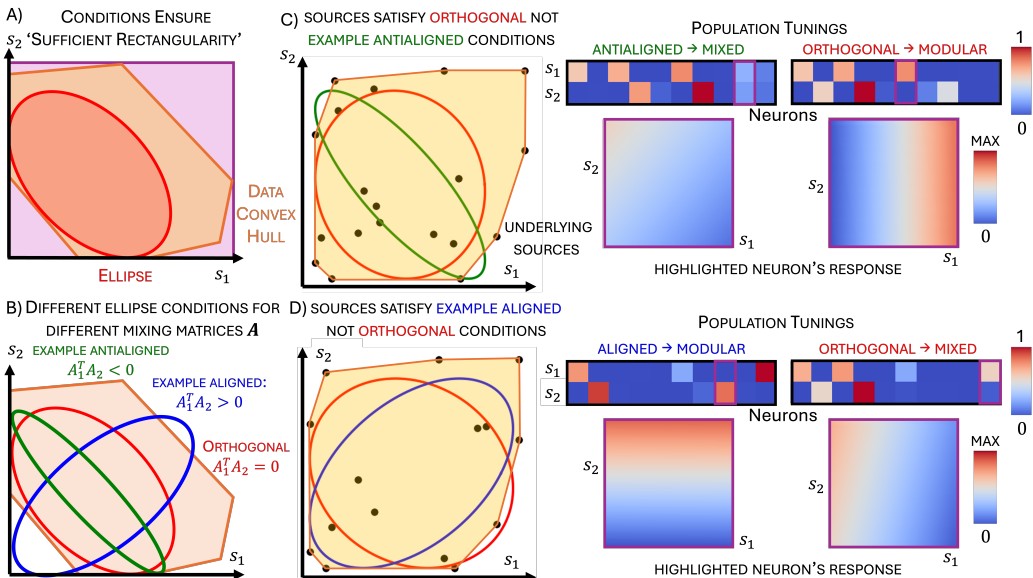

Figure 1: **(A)** We schematise the the identifiability conditions for two sources; the conditions specify a set (e.g. the red ellipse that depends on $A$ in the observed data) that the convex hull of the underlying empirical source distribution has to engulf. If this condition is satisfied the empirical source data is 'rectangular enough', there is no better linear transformation, and the sources are recovered, else the optimal representation is mixed. **(B)** When the data consists of linearly mixed sources via $A$, then this equates to warping the identifiability conditions: either via aligning (blue) or antialigning (green). **(C)** Source alignment can make the optimal solution mix when it would have otherwise modularised. We show such a dataset in which the orthogonal but not an example antialigned identifiability conditions are satisfied. Matching the theory, numerical solutions are modular for the orthogonally encoded sources (rightmost column), but not for the antialigned (middle column). We plot the linear conditional mutual information (Hsu et al., 2023; Dorrell et al., 2025) between each neuron and source scaled by the neuron's peak activity. Below we display a highlighted (purple) neuron's tuning to sources. **(D)** Similarly, source alignment can cause recovery of sources that would otherwise not be. We show an example dataset where aligning the sources by a specific amount causes the warped identifiability conditions to be satisfied (blue). Matching this, the aligned sources are recovered (middle column), but not the orthogonal ones (right column).

Our main theorem simply states that if and only if the sources are tightly scattered the problem is identifiable, i.e., up to scaling and permutation $Z$ and $W_{\text{out}}$ will recover $S$ and $A$. Our theorem relates the empirical dataset to identifiability, so is inherently a finite-sample result. To the best of our knowledge, this is the first identifiability condition for semi-nonnegative matrix factorisation that is both necessary and sufficient, and we suspect similar ideas can generalise to dictionary learning.

**Theorem 1** (Identifiability of Nonnegative Affine Autoencoders)**.** *In the same setting as problem 1, if and only if the matrix $\bar{S}$ is tightly scattered with respect to $A$ then the optimal positive affine autoencoder recovers the sources, i.e. each neuron's activity is an affine function of at most one source, and every source is encoded by at least one neuron.*

The proof is in appendix B, and in appendix C we derive a similar imperfect reconstruction result in which fitting is enforced by a mean-square error term.

**Previous Work**   The identifiability of various matrix factorisation problems has been extensively studied, and our condition is best seen as a generalisation of earlier sufficient scattering results. Donoho & Stodden (2003) studied nonnegative matrix factorisation and showed that if the data satisfied a scattering condition (that the convex cone of datapoints surrounded an 'ice-cream cone' in the postive orthant) the model was identifiable. Hu & Huang (2023) present a state-of-the-art sufficient identifiability condition for dictionary learning regarding the scattering of a sign-permuted dataset around a unit ball. This is similar both to our work, to work on polytope matrix factorisation (Tatli & Erdogan, 2021a;b) and to sufficient spread conditions for identifiabilty of nonnegative matrix factorisation (Huang et al., 2013). These works involve scattering relative to a set defined independently of the mixing matrix $A$. In contrast, our work studies a different factorisation algorithm which lets us harness the convexity of the problem, and is the only work to adapt the scattering condition to both $A$ and the dataset covariance. This allows us to derive the first *necessary* and sufficient criterion for semi-nonnegative matrix factorisation, as all the cited examples are only sufficient. The other works (Whittington et al., 2023; Dorrell et al., 2025) that study this form of neural network matrix factorisation, the affine nonnegative autoencoder, are far less general as they only only develop conditions for the simpler case where $A$ is an orthogonal matrix.

## 4    WHEN DOES OPTIMALITY IMPLY UNIQUE SINGLE NEURON TUNING?

We have described conditions for recovery of the linearly-mixed 'true' factors in semi-nonnegative matrix factorisation via the construction of nonnegative affine autoencoders with hidden representation, $z$. This result relied upon framing the problem in terms of the representational similarity matrix, $Q = Z^T Z$, which summarises the population structure. This link between population representational similarity and single neuron coding of sources raises the tantalising possibility of formally understanding how individual neural tuning curves relate to neural manifolds, and inferring the brain's underlying algorithm from population activity and single neurons alike.

Classical work in neuroscience involves correlating neural activity with task-relevant variables to infer function (e.g., evidence of whitening operations (Atick & Redlich, 1990; 1992), or conceptual navigation strategies (Constantinescu et al., 2016)). This assumes that activity patterns are linked to function. However, results in machine learning have shown how fragile the representational-function link can be without further assumptions. The same function can be implemented by many neural networks (Baldi & Hornik, 1989), and the internal representations of those networks might differ radically, confounding attempts to use correlations between neural activity and task-relevant variables to infer function. This point was reinforced by a recent paper that analytically studied the solution space of linear neural networks, finding that without further assumptions the representational similarity matrices of networks performing the same function are completely unrelated (Braun et al., 2025). Similar concerns arise when using single-neuron tuning curves. Most simply, rotating the neural population scrambles the tuning curves while leaving the representational similarity unchanged, potentially meaning the observed neural basis bears little relationship to function. Indeed, concern over the relevance of single neuron responses has led to debate over the correct level of neuroscientific enquiry (Barack & Krakauer, 2021).

Here, we construct a defence of traditional neuroscience approaches. We begin at the population level, defending the use of representational similarity matrices. Then we develop theory that outlines situations in which even single-neuron tuning curves are tightly coupled to optimality.

**Unique Representational Similarity Matrices** There is no unique neural network for a given function. This is simply illustrated by adding additional neurons to a network that are disconnected from the output, shaping representational similarity, but not affecting function. However, if the network is regularised, then irrelevant neurons will have low activity and so won't affect representational similarly. Indeed, Braun et al. (2025) find that regularised linear networks have unique representational similarity matrices (those with either the objectives O1&O4, or O3&O4, in the language of appendix A). Our framework extends this argument, showing that a large family of problems beyond regularised linear neural networks are convex over the set of representational similarity matrices, making the choice of representational similarity far from arbitrary.

**Unique Single Neuron Responses** We now go one step further: *when should we expect every network that optimally implements the same function to exhibit same single-neuron tuning curves?* For this to happen, not only does a function need a unique representational similarity (as above), but also rotational symmetry must be broken. This is most naturally done by constraining neural activity to be nonnegative (Dordek et al., 2016; Sengupta et al., 2018; Whittington et al., 2023; Dorrell et al., 2025). Thus, to answer the above question, we derive conditions under which nonnegativity ensures that all globally optimal representations are related only by the inherent permutation symmetry; in other words, all optimal solutions contain the same set of neurons, but shuffled.

**Precise Problem and Sufficient Conditions** We study a nonnegative representation, $\boldsymbol{Z}^*$, which has the globally optimal similarity structure: $\boldsymbol{Z}^{*T}\boldsymbol{Z}^* = \boldsymbol{Q}^*$. For another representation to be globally optimal it must have the same dot-product similarity, $\boldsymbol{Z}^T\boldsymbol{Z} = \boldsymbol{Q}^\star$, and thus must be an orthogonal transform of the first representation: $\boldsymbol{Z} = \boldsymbol{O}\boldsymbol{Z}^*$. Which $\boldsymbol{Z}$ are possible? This problem therefore directly asks how tightly neural tuning curves ($\boldsymbol{Z}$) are linked to representational optimality ($\boldsymbol{Q}^*$).

Following a similar logic to Theorem 1, we now show that $\boldsymbol{O}$ is constrained to be a permutation matrix – meaning every optimal representation contains the same neural tuning curves – only when the neural response patterns, or tuning curves, satisfy one of a family of sufficient scattering conditions. Intuitively, if the neural tuning curves are 'different enough' from one another no rotation can preserve the optimality of the representation. This is described in the following theorem, roofs are presented in appendix D.

**Theorem 2** (Tight Scattering Implies Unique Neurons). *If a given neural dataset $\{\boldsymbol{z}^{[i]}\}_{i=1}^T$ satisfies the following scattering constraints with respect to a set $E = \{\boldsymbol{x} + \langle\boldsymbol{z}^{[i]}\rangle_i | \boldsymbol{x}^T\boldsymbol{F}^{-1}\boldsymbol{x} = 1\}$ for any positive definite matrix $\boldsymbol{F}$ with diagonal equal to $(\langle\boldsymbol{z}^{[i]}\rangle_i)^2$ then all orthogonal matrices such that $\boldsymbol{O}\boldsymbol{z}^{[i]} \geq 0$ are permutation matrices:*

- *$Conv(\{\boldsymbol{z}^{[i]}\}_i) \supseteq E$*

- *$Conv(\{\boldsymbol{z}^{[i]}\}_i)^* \cap bd(E^*) = \{\lambda\boldsymbol{e}_k | \lambda \in \mathbb{R}, k = 1, ..., d_S\}$*

For each choice of $\boldsymbol{F}$ you get an ellipse, $E$, circumscribed by the range of the data fig. 2A. The first term ensures that the convex hull of the neural responses ($Conv(\{\boldsymbol{z}^{[i]}\}_i)$) 'swallows' the ellipse, while the second is a technical condition that ensures the intersections between the boundaries of the ellipse, $bd(E)$, and the convex hull are along the axes. Intuitively, as long as there is at least one $\boldsymbol{F}$ (and hence $E$) such that these are satisfied, the joint-distribution of neural activities is 'rectangular enough' such that any rotation will necessarily push some activity negative, fig. 1A, and hence the optimal tunings are unique. This can be seen as an extension of the tight scattering results section 3: as long as you're tightly scattered with respect to at least one $\boldsymbol{A}$-dependent ellipse it must be impossible to rotate the code while preserving positivity, else our identifiability results would be void. In fig. 2, we demonstrate an example of a pair of tuning curves that do (fig. 2B) and don't (fig. 2C,D) satisfy these conditions. We have proved the sufficiency of this set of conditions, though we conjecture that they might additionally be necessary.

**Grid Cell Modularisation** As a neuroscientific example, we now apply these results to understand why the famous grid cell tuning curves appear in discrete modules. Grid cells are neurons with a lattice receptive field to position (Hafting et al., 2005) and they come in groups called modules: grid cells within the same module share the same lattice receptive field (but translated) (Stensola et al., 2012), and each module has a different lattice frequency. Here we answer the question why each module has its own set of neurons (as opposed to mixing in the population). To do this, we consider two hypothetical grid cells from two different modules and test whether they satisfy our identifiability conditions. We find that, if the lattice frequency of one module is close to an integer multiple of the other, they don't satisfy our identifiability conditions, otherwise, they do. Thus, the fact that we observe grid cells in discrete modules is a consequence of the non-integer frequency relationship between grid modules. If one module lattice was an integer multiple of the other the population would have done better by using a mixed encoding. This matches the finding that, in order to encode space effectively efficiently, modules are not integer multiples of one another (Sreenivasan & Fiete, 2011; Wei et al., 2015; Dorrell et al., 2023; Schaeffer et al., 2023; Dorrell et al., 2025), and hence are identifiable.

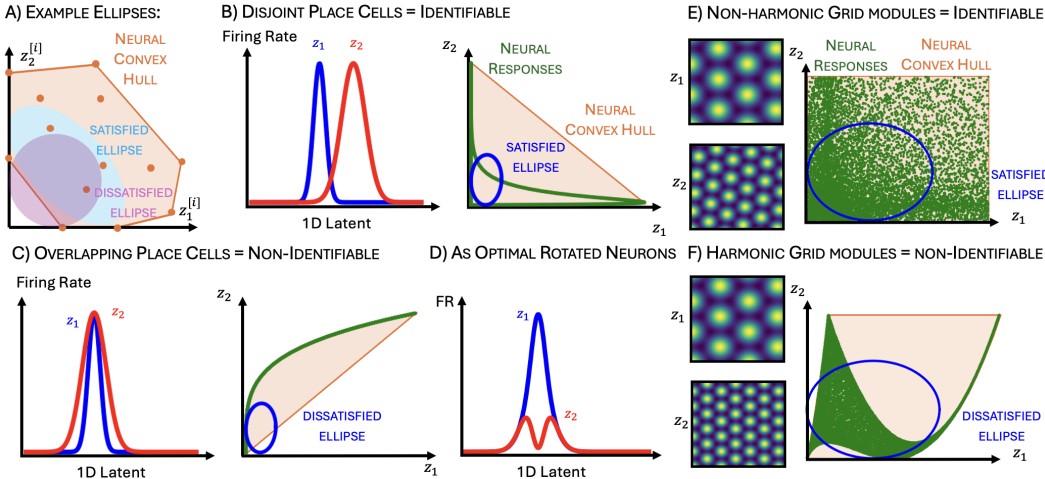

Figure 2: **A)** Illustration of identifiability condition: the condition states that the convex hull of the neural responses must engulf one of a set of ellipses. Plotting the data for two neurons, and drawing two ellipses for different choices of $\boldsymbol{F}$ shows us an example where the ellipse is not (purple) and is (blue) engulfed. Since there is at least one, the condition is satisfied. **(B)** We try this on two neurons, tuned like place cells to a single 1D latent. Plotting the neural responses and an example ellipse shows us that these tunings are identifiable, since the convex hull of the data engulfs the ellipse. However, **(C)** moving the place cells to overlap leads to non-identifiability. Indeed, **(D)** we can rotate the responses to find two different tuning curves with the same optimal dot-product structure. **(E)** We apply this to grid cells. Grid cells from two different modules are identifiable as long as their wavevectors are not integer multiples of one another, otherwise **(F)** they become non-identifiable, and we can numerically find a rotation of these neural responses that preserves nonnegativity.

## 5 TRACTABLE NONLINEAR THEORY OF ON-OFF CODING

Promisingly, many of our convex models are nonlinear (section 2, appendix A). Here, we use a convex nonlinear model to confirm a neural coding conjecture relating the optimality of ON-OFF coding to sparsity. In an ON-OFF code, a single variable is split into two oppositely rectified encodings (Euler et al., 2014), and in some settings this has been shown to be energy efficient (Gjorgjieva et al., 2014). However, not all variables are ON-OFF coded, and it has been conjectured that sparsity governs whether to use an ON-OFF code (Sterling & Laughlin, 2015). Existing theories that exhibit ON-OFF coding are either intractable neural networks (Ocko et al., 2018; Jun et al., 2022) or rely on direct model enumeration (Gjorgjieva et al., 2014), stymieing efforts to derive the parameters that govern ON-OFF optimality. Here, we use a tractable normative model to confirm the conjectured role of sparsity, and analytically derive a simple threshold.

We study a nonnegative affine-decodable representation of a single variable, $\boldsymbol{z}(I)$, similar to eq. (2):

$$\min(\langle||\boldsymbol{z}(I)||^2\rangle_{p(I)} + \lambda||\boldsymbol{w}||_F^2), \qquad \text{subject to} \qquad \boldsymbol{w}^T\boldsymbol{z}(I) + b = I, \qquad \boldsymbol{z}(I) \geq 0 \qquad (10)$$

In appendix E we use the KKT conditions to find an optimal solution, which convexity then guarantees is unique. We find two regimes. In the first, all neurons linearly encode the stimulus; in the second, the neurons split into ON and OFF channels, fig. 3A. Sparsity governs this transition: if $I$ is dense, channel-splitting lowers the firing rates. However, if the variable is sufficiently sparse (e.g. $I \geq 0$ but $I = 0$ often), it is better to ensure that the encoding of $I = 0$ uses low firing rates, leading to a single channel code, fig. 3B. We analytically derive this threshold sparsity, matching simulations, fig. 3C:

$$\text{Prob}(I = 0) > \frac{\langle I\rangle^2_{p(I)}}{\langle I^2\rangle_{p(I)}}. \qquad (11)$$

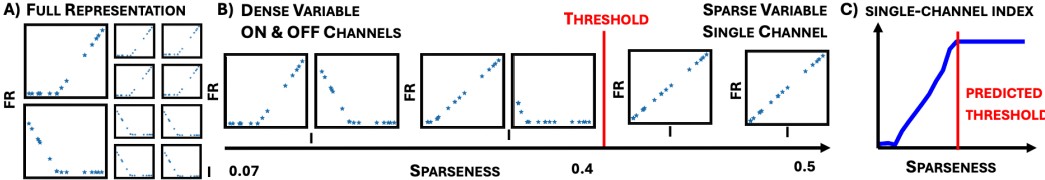

Figure 3: **A)** We numerically find a solution to eq. (10), and plot the firing rates of each neuron in the population as a function of $I$. We find that all tuning curves correspond to either OFF or ON channels. **B)**. We add a point mass at $I = 0$ to the distribution over $I$, thus increasing the sparsity. For low sparsity the optimal representation contains both ON and OFF neurons. As sparsity increases the coding range of the ON neurons increase while the range of the OFF neurons decreases, until at the threshold (eq. 11) the OFF neuron disappears and we are left with ON neurons only. We display only the unique tuning curves, the population is comprised of copies of these tuning curves. **C)** We quantify the degree of single channel coding, appendix E, and find it slowly increases to a ceiling at the predicted sparsity threshold, corresponding to a completely single channel representation.

## 6  DISCUSSION

In this work we showed that a variety of interesting representational optimisation problems could be written as convex optimisations over the set of representational similarity matrices. We used this to derive a tight matrix-factorisation identifiability criterion, section 3, to understand the link between tuning curves and optimality, section 4, and used the tractable nonlinearity of these theories to understand ON-OFF coding section 5. These results sound an optimistic note for classic neuroscience: if we can correctly frame the relevant neural computation and constraints, it seems that representation and function are tightly coupled, including often at the single neuron level, section 4. Given this optimism, we hope future work will use this paper's framework to develop analytic theories for nonlinear phenomena in both neuroscience and machine learning.

**Literature: Convex Analysis & Neural Networks** A lot of work has, as here, reframed the optimisation of neural networks as convex optimisation. Earlier approaches studied the optimal addition of single neurons to existing networks and found this was convex (Bengio et al., 2005; Bach, 2017). Neural tangent kernel approaches showed that, in the infinite width, training a neural network was convex, but this limit often removes interesting representational learning phenomenon (Jacot et al., 2018). More recent work has shown that the entire standard neural network problem can be framed as a convex optimisation for 1-hidden layer ReLU networks (Pilanci & Ergen, 2020; Ergen & Pilanci, 2021). These results have been extensively developed and used to understand neural network phenomenology (Sahiner et al., 2020; Zeger & Pilanci, 2025), and likely have much to offer neuroscience. They differ technically from ours in the variables used to construct the convex problem (we use representational similarity matrices, they use a version of the weights). Further, they stick tightly to the neural network problem setting, while we follow the more neuroscience approach of optimising a representation, allowing us flexibility at the cost of less relevance for machine learning.

**Limitation 1: Many Neurons** First, If the number of neurons is larger than the number of datapoints, the rank of $Q$ is unconstrained, and our optimisation problems are convex. Unfortunately, restricting the neuron number, and hence rank of $Q$, is a non-convex constraint. One general fix that future work could usefully explore is to replace the rank constraint with its convex relaxation–a constraint on the nuclear norm of $Q$ (sum of singular values). Further, section 3 demonstrates a more bespoke workaround: if the solution to the unconstrained problem uses few neurons, it will also be the solution in the neuron-constrained setting, letting us relax the neuron constraint, appendix B.5.

**Limitation 2: Computational Intractability** A possible dividend from a convex reformulation is efficient optimisation algorithms. These, however, rely on efficiently testing whether the proposed solution is a member of the feasible set. Unfortunately determining membership of the set of completely positive matrices is NP-hard (Dickinson & Gijben, 2014). One approach is to approximate the set with a hierarchy of increasingly precise enclosing sets (Nishijima & Nakata, 2024). Alternatively, we could optimise directly over $Z$, and use the conditions in section 4 to try and prove global optimality. However, verifying sufficient scattering conditions, like those in section 4, is often NP-hard. though fortunately there is work tackling this problem (Gillis & Luce, 2024).

**Reproducibility Statement** The appendices include all our mathematical proofs, and code for the small simulations we ran to test the theoretical results can be found at: `https://github.com/WilburDoz/Convex_Efficient_Coding_ICLR_2026.git`.

**Acknowledgements** The authors thank Pierre Glaser for discussions about convexity, Tim Behrens and Kris Jensen for careful readings of earlier drafts, James Fitzgerald for discussions about related optimisation problems, and especially Erin Grant for extensive reading and discussion, and Nicholas Gillis for extensive comments.

We thank the following funding sources: Gatsby Charitable Foundation (GAT3755; W.D. & P.E.L); Sir Henry Wellcome Post-doctoral Fellowship (222817/Z/21/Z; J.C.R.W); European Research Council Starting Grant (NARFB/101222868; J.C.R.W).

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

# A  A FAMILY OF CONVEX OBJECTIVES, CONSTRAINTS, AND PROBLEMS

In this section we demonstrate that a suite of interesting problems can be rewritten as convex optimisation problems over the set of representational similarity matrices. In so doing, we are inspired by (Sengupta et al., 2018) which, to our knowledge, is the only other result of this sort. Their case considers nonnegative similarity matching. We broaden this to include (nonnegative) linear neural networks, and autoencoders using either a ReLU, unconstrained, or no nonlinearity.

We study a family of optimisation problems; each of which we reframe as optimisations over the set of representational similarity. These problems are made by composing a set of objectives and constraints. Since both the sum of convex functions, and the intersection of convex sets, are convex (Boyd & Vandenberghe, 2004) we show the convexity of each component objectives or constraints, which we can then combine to show convexity of a large set of problems. We begin by describing the family of problems, then we study convexity of each constraint set, before finally studying the convexity of the objectives.

## A.1  FAMILY OF PROBLEMS

We consider optimising a representation, $\boldsymbol{Z} \in \mathbb{R}^{d_z \times N}$, where $N$ is the number of datapoints and $d_z$ is the number of neurons. We will assume $d_z \geq N$ when needed. We will prove that the following constraints and objectives are convex over the set of representational similarity matrices.

### CONSTRAINTS

C1  Nonnegativity: $\boldsymbol{Z} \geq 0$ (As in Sengupta et al. (2018))

C2  Bounded Firing Rates: $\boldsymbol{Z} \leq k$

C3  Perfect affine decodability of a dataset, $\boldsymbol{Y} \in \mathbb{R}^{d_y \times N}$: $\boldsymbol{W}_{\text{out}}\boldsymbol{Z} + \boldsymbol{b}_{\text{out}}\boldsymbol{1}^T = \boldsymbol{Y}$. (Can be relaxed to linear)

C4  Full rank affine relationship between representation and input dataset such that $\boldsymbol{Y}$ can be decoded, $\boldsymbol{X} \in \mathbb{R}^{d_x \times N}$: $\boldsymbol{W}_{\text{in}}\boldsymbol{X} + \boldsymbol{b}_{\text{in}}\boldsymbol{1}^T = \boldsymbol{Z}$, $\text{rank}(\boldsymbol{W}_{\text{in}}) = d_x$. (Can be relaxed to linear)

C5  A ReLU relationship between representation and input dataset, $\boldsymbol{X} \in \mathbb{R}^{d_x \times N}$: $\boldsymbol{Z} = \text{ReLU}(\boldsymbol{W}_{\text{in}}\boldsymbol{X} + \boldsymbol{b}_{\text{in}}\boldsymbol{1}^T)$

C6  A firing rate constraint: $||\boldsymbol{z}^{[i]}||_2^2 \leq k \forall i$. (As in Sengupta et al. (2018))

### OBJECTIVES

O1  L2 activity loss: $\langle ||\boldsymbol{z}^{[i]}||^2 \rangle_i$

O2  A modified L1 activity loss: $\sum_{d=1}^{d_z} ||\boldsymbol{z}_d||_1^2$, where $\boldsymbol{z}_d \in \mathbb{R}^{d_z}$ is neuron $d$'s response vector, when the representation is nonnegative (C1)

O3  An L2 input weight loss when the relationship with input data is affine (C4): $||\boldsymbol{W}_{\text{in}}||_F^2$

O4  An L2 output weight loss when the data is perfectly affine decodable (C3): $||\boldsymbol{W}_{\text{out}}||_F^2$

O5  An affine L2 reconstruction error with weight regularisation on the output weights: $\lambda_w ||\boldsymbol{W}_{\text{out}}||_F^2 + ||\boldsymbol{W}_{\text{out}}\boldsymbol{Z} + \boldsymbol{b}_{\text{out}}\boldsymbol{1}^T - \boldsymbol{Y}||_F^2$

O6  Similarity matching for a dataset $\boldsymbol{X}$, as in (Sengupta et al., 2018): $-\text{Tr}[(\boldsymbol{X}^T\boldsymbol{X} - \alpha\boldsymbol{1}\boldsymbol{1}^T)\boldsymbol{Z}^T\boldsymbol{Z}]$.

O7  Nonlinear similarity matching with input similarity $\boldsymbol{G} \in \mathbb{R}^{N \times N}$ and elementwise exponentiation: $\text{Tr}[\boldsymbol{G}e^{\boldsymbol{Z}^T\boldsymbol{Z}}]$.

We list below some problems that can be constructed from this menu:

- C3, C4, & O3, O5: A regularised, linear network as in Saxe et al. (2019).

- C1, C3, C5, & O2, O5: A regularised one-layer ReLU neural network; such as a sparse autoencoder.

- C1, C3, C4, & O1, O3, O4: A perfectly fitting positive affine autoencoder, as in Whittington et al. (2023); Dorrell et al. (2025).

- C1, C3, C4, & O3, O4: A regularised linear neural network with a positivity constraint..
- C1, C3, & O2, O4/O5: A nonlinear but affine-decodable representation such as the last layer of an unconstrained neural network.
- C1, C6, & O6: Nonnegative similarity matching, as in Sengupta et al. (2018).
- C1, C6, & O7: A representation that nonlinearly matches the data similarity, such as used to model multi-field place cells (Dorrell et al., 2023).

## A.2 CONVEX FEASIBLE SETS

We show the convexity of the feasible set of representational similarity matrices for each constraint.

**C1: Positive Representations** Following (Sengupta et al., 2018)), the set of dot product matrices with unconstrained ranks formed from vectors that are themselves positive ($Q = Z^T Z$ for $Z \geq 0$) form a convex set called the set of completely positive matrices (Berman & Shaked-Monderer, 2003). As long as we allow many neurons so there is no rank constraint on $Q$, we're good.

**C2: Bounded Firing Rates** We generalise slightly the normal proof of convexity of the set of completely positive matrices. Instead, we say that a dot-product similarity matrix is made from bounded firing rates if it can be written as the product of a representation where all elements are smaller than some constant $k$: $Q = Z^T Z$, where $Z \leq k$. We can equivalently write this as $Q = \sum_{l=1}^{L} z_d z_d^T$ for some number $L$. Convexly combining such matrices preserves this property:

$$Q_\lambda = \lambda Q + (1 - \lambda)\tilde{Q} = \lambda \sum_{l=1}^{L} z_d z_d^T + (1 - \lambda) \sum_{l=1}^{\tilde{L}} \tilde{z}_d \tilde{z}_d^T \tag{12}$$

Therefore, since $Q_\lambda$ can be written as the sum of outer products of vectors that are all smaller than $k$, it is also a member of the set, making the set convex.

**C3: Perfect Affine Decodability** To be affine decodable the representation must contain a subspace that encodes the demeaned labels. We can find the relevant subspace using the reduced SVD: $\bar{Y} = U S V^T$. The reduced SVD is the same as the SVD but you remove all singular vectors with singular value equal to zero. Denoting the rank of $\bar{Y}$ with $\rho$, this gives us a diagonal square positive definite matrix $S \in \mathbb{R}^{\rho \times \rho}$, and two rectangular orthogonal matrices $U \in \mathbb{R}^{d_x \times \rho}$, $V \in \mathbb{R}^{T \times \rho}$. Let's break down the representation into the part within the span of $V$ and the part living in the orthogonal complement, $V_\perp \in \mathbb{R}^{T \times (d_z - \rho)}$:

$$\bar{Z} = AV + BV_\perp \qquad A \in \mathbb{R}^{d_Z \times \rho} \quad B \in \mathbb{R}^{d_Z \times (d_Z - \rho)} \tag{13}$$

In order for the demeaned labels to be linearly decodable $A$ must have full column rank and the columns of $B$ must be linearly independent from the columns of $A$. These are constraints on the representation. What do they imply for the representational similarity matrices?

$$\bar{Q} = \bar{Z}^T \bar{Z} = [V \quad V_\perp] \begin{bmatrix} A^T A & A^T B \\ B^T A & B^T B \end{bmatrix} \begin{bmatrix} V^T \\ V_\perp^T \end{bmatrix} = [V \quad V_\perp] \begin{bmatrix} Q_A & Q_C \\ Q_C^T & Q_B \end{bmatrix} \begin{bmatrix} V^T \\ V_\perp^T \end{bmatrix} \tag{14}$$

We therefore relate the two constraints on $\bar{Z}$ to two constraints on $\bar{Q}$:

1. $A$ having full column rank is equivalent to $A^T A$ being invertible, this is easy to see.
2. $B$ and $A$ having linearly independent columns is equivalent to the generalised schur complement of $\bar{Q} = Q_A - Q_C Q_B^\dagger Q_C$ being invertible, this is less obvious.

Our proof will proceed in stages. First we will demonstrate the equivalence stated in point 2. Then we will show that each of these two properties are preserved under convex combination of the representational similarity matrix. Finally, we will relate the demeaned representational similarity back to $Q$.

**Equivalence of Decodability Conditions** First we go in one direction and show that if $Q_A - Q_C Q_B^\dagger Q_C$ is singular that implies the column spaces of $A$ and $B$ overlap. Singular means:

$$(Q_A - Q_C Q_B^\dagger Q_C)x = 0 \qquad \text{for some} \quad x \neq 0 \tag{15}$$

We can develop this into:

$$\boldsymbol{A}^T(\mathbb{1} - \boldsymbol{B}(\boldsymbol{B}^T\boldsymbol{B})^\dagger\boldsymbol{B}^T)\boldsymbol{A}\boldsymbol{x} = \boldsymbol{A}^T(\mathbb{1} - \boldsymbol{B}\boldsymbol{B}^\dagger(\boldsymbol{B}^\dagger)^T\boldsymbol{B}^T)\boldsymbol{A}\boldsymbol{x} = 0 \tag{16}$$

Now using the fact that $\boldsymbol{B}\boldsymbol{B}^\dagger$ is the orthogonal projector onto the columnspace of $\boldsymbol{B}$, and orthogonal projectors are hermitian and idempotent, this implies:

$$\boldsymbol{A}^T(\mathbb{1} - \boldsymbol{B}\boldsymbol{B}^\dagger)\boldsymbol{A}\boldsymbol{x} = 0 \tag{17}$$

$(\mathbb{1} - \boldsymbol{B}\boldsymbol{B}^\dagger)$ is the orthogonal projector onto the kernel of $\boldsymbol{B}$. Since $\boldsymbol{A}\boldsymbol{x}$ is a linear combination of the columns of $\boldsymbol{A}$, it lives in the span of $\boldsymbol{A}$. Projecting this vector will only produce a vector with zero component in the span of $\boldsymbol{A}$ (i.e. $\boldsymbol{A}^T\boldsymbol{y} = 0$) if it is entirely set to 0. Hence the condition is that $(\mathbb{1} - \boldsymbol{B}\boldsymbol{B}^\dagger)\boldsymbol{A}\boldsymbol{x} = 0$. In english, $(\mathbb{1} - \boldsymbol{B}\boldsymbol{B}^\dagger)\boldsymbol{A}\boldsymbol{x} = 0$ means that $\boldsymbol{A}\boldsymbol{x}$ lives entirely within the span of the columns of $\boldsymbol{B}$ implying the columns of $\boldsymbol{A}$ and $\boldsymbol{B}$ are not linearly independent.

Now in the reverse direction, if the columns of $\boldsymbol{A}$ and $\boldsymbol{B}$ are not linearly independent then some combination of columns of columns of $\boldsymbol{A}$ equals some combination of columns of $\boldsymbol{B}$: $\boldsymbol{A}\boldsymbol{a} = \boldsymbol{B}\boldsymbol{b}$. Using the previous logic, this implies that:

$$(\boldsymbol{Q}_A - \boldsymbol{Q}_C\boldsymbol{Q}_B^\dagger\boldsymbol{Q}_C)\boldsymbol{a} = \boldsymbol{A}^T(\mathbb{1} - \boldsymbol{B}\boldsymbol{B}^\dagger)\boldsymbol{A}\boldsymbol{a} = \boldsymbol{A}^T(\mathbb{1} - \boldsymbol{B}\boldsymbol{B}^\dagger)\boldsymbol{B}\boldsymbol{b} = 0 \tag{18}$$

Since $\boldsymbol{B}\boldsymbol{b}$ lives in the span of the columns of $\boldsymbol{B}$ so its projection onto the orthogonal complement of $\boldsymbol{B}$ is zero. Hence $\boldsymbol{Q}_A - \boldsymbol{Q}_C\boldsymbol{Q}_B^\dagger\boldsymbol{Q}_C$ is non-invertible, demonstrating the stated equivalence.

**Convex Combinations - Condition 1** Now we show that these two properties are preserved under convex combination. Take the convex combination of two matrices, $\boldsymbol{Q}_1$ and $\boldsymbol{Q}_2$, that satisfy conditions 1 and 2 above:

$$\boldsymbol{Q}_\lambda = \lambda\boldsymbol{Q}_1 + (1 - \lambda)\boldsymbol{Q}_2 = [\boldsymbol{V} \quad \boldsymbol{V}_\perp] \begin{bmatrix} \lambda\boldsymbol{Q}_{A,1} + (1 - \lambda)\boldsymbol{Q}_{A,2} & \lambda\boldsymbol{Q}_{C,1} + (1 - \lambda)\boldsymbol{Q}_{C,2} \\ \lambda\boldsymbol{Q}_{C,1}^T + (1 - \lambda)\boldsymbol{Q}_{C,2}^T & \lambda\boldsymbol{Q}_{B,1} + (1 - \lambda)\boldsymbol{Q}_{B,2} \end{bmatrix} \begin{bmatrix} \boldsymbol{V}^T \\ \boldsymbol{V}_\perp^T \end{bmatrix} \tag{19}$$

where $\lambda \in [0, 1]$. First, the convex combination of two positive definite matrices is positive definite, therefore $\lambda\boldsymbol{Q}_{A,1} + (1 - \lambda)\boldsymbol{Q}_{A,2}$ is positive definite, satisfying the first condition.

**Convex Combinations - Condition 2** Now we turn to the convex combinations of nonsingular generalised schur complements. First we show that the generalised schur complement is a matrix concave function - a very simple generalisation of an exercise from (Boyd & Vandenberghe, 2004) using a proof technique from maths stack exchange user p.s.. Then we use that to easily demonstrate the invertibility of the generalised schur complement.

Consider the map that takes a positive semidefinite matrix to its generalised schur complement:

$$f : S_+^n \to S^\rho \qquad \boldsymbol{Q} = \begin{bmatrix} \boldsymbol{Q}_A & \boldsymbol{Q}_C \\ \boldsymbol{Q}_C^T & \boldsymbol{Q}_B \end{bmatrix} \to \boldsymbol{Q}_A - \boldsymbol{Q}_C\boldsymbol{Q}_B^\dagger\boldsymbol{Q}_C^T \tag{20}$$

A function is matrix concave if its the hypograph is a convex set (Boyd & Vandenberghe, 2004). The hypograph is:

$$\text{hypo}f = \{(\boldsymbol{Q}, \boldsymbol{T})|f(\boldsymbol{Q}) \succcurlyeq T, \boldsymbol{Q} \in S_+^n, \boldsymbol{T} \in S^\rho\} \tag{21}$$

We make use of the following characterisation of when a matrix is positive semidefinite using the generalised schur complement (Gallier, 2010; 2011):

$$\boldsymbol{Q} = \begin{bmatrix} \boldsymbol{Q}_A & \boldsymbol{Q}_C \\ \boldsymbol{Q}_C^T & \boldsymbol{Q}_B \end{bmatrix} \succcurlyeq 0 \iff \boldsymbol{Q}_B \succcurlyeq 0, \quad (\mathbb{1} - \boldsymbol{Q}_B\boldsymbol{Q}_B^\dagger)\boldsymbol{Q}_C^T = 0, \quad \boldsymbol{Q}_A - \boldsymbol{Q}_C\boldsymbol{Q}_B^\dagger\boldsymbol{Q}_C^T \succcurlyeq 0 \tag{22}$$

Since $\boldsymbol{Q}_B \succcurlyeq 0$ and $(\mathbb{1} - \boldsymbol{Q}_B\boldsymbol{Q}_B^\dagger)\boldsymbol{Q}_C = 0$, we can proceed with the proof as on maths stack exchange:

$$f(Q) = \boldsymbol{Q}_A - \boldsymbol{Q}_C\boldsymbol{Q}_B^\dagger\boldsymbol{Q}_C^T \succcurlyeq \boldsymbol{T} \iff \begin{bmatrix} \boldsymbol{Q}_A - \boldsymbol{T} & \boldsymbol{Q}_C \\ \boldsymbol{Q}_C^T & \boldsymbol{Q}_B \end{bmatrix} \succcurlyeq 0 \iff \boldsymbol{Q} - \begin{bmatrix} \boldsymbol{T} & 0 \\ 0 & 0 \end{bmatrix} \succcurlyeq 0 \tag{23}$$

Defining the linear map:

$$L(\boldsymbol{Q}, \boldsymbol{T}) = \boldsymbol{Q} - \begin{bmatrix} \boldsymbol{T} & 0 \\ 0 & 0 \end{bmatrix} \tag{24}$$

Then:

$$\text{hypo}f = \{(\boldsymbol{Q}, \boldsymbol{T}) | L(\boldsymbol{Q}, \boldsymbol{T}) \in S_+^n, \boldsymbol{Q} \in S_+^n, \boldsymbol{T} \in S^\rho\} \tag{25}$$

And because $S_+^n$ is a convex set, so is the hypograph, hence the generalised schur complement is matrix concave.

Given that the generalised schur complement is matrix concave, convex combinations satisfy:

$$f(\lambda \boldsymbol{Q}_1 + (1 - \lambda)\boldsymbol{Q}_2) \succcurlyeq \lambda f(\boldsymbol{Q}_1) + (1 - \lambda)f(\boldsymbol{Q}_2) \succ 0 \tag{26}$$

where the last line follows from the invertibility of both $f(\boldsymbol{Q}_1)$ and $f(\boldsymbol{Q}_2)$. Therefore $f(\lambda \boldsymbol{Q}_1 + (1 - \lambda)\boldsymbol{Q}_2)$ is positive definite and hence convex.

**Demeaned to Meaned $\boldsymbol{Q}$** We have therefore shown that the set of $\bar{\boldsymbol{Q}}$ corresponding to affine decodable representations is convex. $\bar{\boldsymbol{Q}}$ and $\boldsymbol{Q}$ are clearly related:

$$\bar{\boldsymbol{Q}} = (\boldsymbol{Z} - \frac{1}{T}\boldsymbol{Z}\mathbf{1}\mathbf{1}^T)^T(\boldsymbol{Z} - \frac{1}{T}\boldsymbol{Z}\mathbf{1}\mathbf{1}^T) = \boldsymbol{Q} - \frac{1}{T}\boldsymbol{Q}\mathbf{1}\mathbf{1}^\mathbf{T} - \frac{1}{T}\mathbf{1}\mathbf{1}^T\boldsymbol{Q} + \frac{\mathbf{1}^T\boldsymbol{Q}\mathbf{1}}{T^2}\mathbf{1}\mathbf{1}^T = P(\boldsymbol{Q}) \tag{27}$$

And, since this is linear in $\boldsymbol{Q}$, it is easy to check that, matching intuition, the demeaned representational similarity of the convex combination of two representational similarity matrices is the convex combination of their demeaned partners:

$$P(\boldsymbol{Q}_\lambda) = P(\lambda \boldsymbol{Q} + (1 - \lambda)\tilde{\boldsymbol{Q}}) = \lambda P(\boldsymbol{Q}) + (1 - \lambda)P(\tilde{\boldsymbol{Q}}) \tag{28}$$

Hence, a convex combination of affine decodable $\boldsymbol{Q}$ is also affine decodable.

**C4: Affine Input** First let's rewrite the affine constraint more simply. Let's define:

$$\boldsymbol{X}' = \begin{bmatrix} \boldsymbol{X} \\ \mathbf{1} \end{bmatrix} \qquad \boldsymbol{W}_{\text{in}}' = [\boldsymbol{W}_{\text{in}} \quad \boldsymbol{b}_{\text{in}}] \quad \text{such that} \quad \boldsymbol{Z} = \boldsymbol{W}_{\text{in}}'\boldsymbol{X}' \tag{29}$$

Now, take two dot product matrices formed from representations that are affine functions of the inputs: $\boldsymbol{Q} = \boldsymbol{Z}^T\boldsymbol{Z}$, $\boldsymbol{Z} = \boldsymbol{W}_{\text{in}}'\boldsymbol{X}'$, and $\tilde{\boldsymbol{Q}} = \tilde{\boldsymbol{Z}}^T\tilde{\boldsymbol{Z}}$, $\tilde{\boldsymbol{Z}} = \tilde{\boldsymbol{W}}_{\text{in}}'\boldsymbol{X}'$. Take their convex combination:

$$\lambda \boldsymbol{Q} + (1 - \lambda)\tilde{\boldsymbol{Q}} = \boldsymbol{X}'^T(\lambda \boldsymbol{W}_{\text{in}}'^T\boldsymbol{W}_{\text{in}}' + (1 + \lambda)\tilde{\boldsymbol{W}}_{\text{in}}'^T\tilde{\boldsymbol{W}}_{\text{in}}')\boldsymbol{X}' \quad \lambda \in [0, 1] \tag{30}$$

Both $\boldsymbol{W}_{\text{in}}'^T\boldsymbol{W}_{\text{in}}'$ and $\tilde{\boldsymbol{W}}_{\text{in}}'^T\tilde{\boldsymbol{W}}_{\text{in}}'$ are positive semi-definite matrices, therefore their convex combination will be as well. Further any positive semidefinite matrix can be decomposed into two parts, therefore we can write $\lambda \boldsymbol{W}_{\text{in}}'^T\boldsymbol{W}_{\text{in}}' + (1 + \lambda)\tilde{\boldsymbol{W}}_{\text{in}}'^T\tilde{\boldsymbol{W}}_{\text{in}}' = \hat{\boldsymbol{W}}_{\text{in}}'^T\hat{\boldsymbol{W}}_{\text{in}}'$ for some matrix $\hat{\boldsymbol{W}}_{\text{in}}'$. Hence

$$\lambda \boldsymbol{Q} + (1 - \lambda)\tilde{\boldsymbol{Q}} = \boldsymbol{X}'^T\hat{\boldsymbol{W}}_{\text{in}}'^T\hat{\boldsymbol{W}}_{\text{in}}'\boldsymbol{X}' = (\hat{\boldsymbol{W}}_{\text{in}}'\boldsymbol{X}')^T(\hat{\boldsymbol{W}}_{\text{in}}'\boldsymbol{X}') = \hat{\boldsymbol{Z}}^T\hat{\boldsymbol{Z}} \tag{31}$$

Hence the convex combination of two similarity matrices of affine functions of the input data will also be such a similarity matrix.

**C5: ReLU Input** Let's say you have two representation that are ReLU and affine related to their inputs:

$$\boldsymbol{Z} = \text{ReLU}(\boldsymbol{W}_{\text{in}}\boldsymbol{X} + \boldsymbol{b}_{\text{in}}\mathbf{1}^T) \qquad \tilde{\boldsymbol{Z}} = \text{ReLU}(\tilde{\boldsymbol{W}}_{\text{in}}\boldsymbol{X} + \tilde{\boldsymbol{b}}_{\text{in}}\mathbf{1}^T) \tag{32}$$

With their respective representational similarity matrices, $\boldsymbol{Q} = \boldsymbol{Z}^T\boldsymbol{Z}$ and $\tilde{\boldsymbol{Q}} = \tilde{\boldsymbol{Z}}^T\tilde{\boldsymbol{Z}}$. Given enough neurons we can always create another representation $\boldsymbol{Z}_\lambda$ with representational similarity matrix $\lambda \boldsymbol{Q} + (1 - \lambda)\tilde{\boldsymbol{Q}}$:

$$\boldsymbol{Z}_\lambda = \text{ReLU}\left(\begin{bmatrix} \sqrt{\lambda}\boldsymbol{W}_{\text{in}} \\ \sqrt{1 - \lambda}\tilde{\boldsymbol{W}}_{\text{in}} \end{bmatrix}\boldsymbol{X} + \begin{bmatrix} \sqrt{\lambda}\boldsymbol{b}_{\text{in}}\mathbf{1}^T \\ \sqrt{1 - \lambda}\tilde{\boldsymbol{b}}_{\text{in}}\mathbf{1}^T \end{bmatrix}\right) = \begin{bmatrix} \sqrt{\lambda}\boldsymbol{Z} \\ \sqrt{1 - \lambda}\tilde{\boldsymbol{Z}} \end{bmatrix} \tag{33}$$

**C6: Constrained firing rate norm** If the diagonals of both $\boldsymbol{Q} = \boldsymbol{Z}^T\boldsymbol{Z}$ and $\tilde{\boldsymbol{Q}} = \tilde{\boldsymbol{Z}}^T\tilde{\boldsymbol{Z}}$ are below some constant $k$, then the same will be true for convex combinations of $\boldsymbol{Q}$ and $\tilde{\boldsymbol{Q}}$

### A.3 CONVEX OBJECTIVES

Here we show the convexity of each proposed objective.

**O1: L2 Activity**   The activity loss can be easily rewritten:

$$\frac{1}{T}||\boldsymbol{Z}||_F^2 = \frac{1}{T}\operatorname{Tr}[\boldsymbol{Z}^T\boldsymbol{Z}] = \frac{1}{T}\operatorname{Tr}[\boldsymbol{Q}] \tag{34}$$

which since it is a linear function of $\boldsymbol{Q}$ is clearly a convex function.

**O2: L1 Activity**   We define a particular form of the L1 activity loss:

$$\sum_{d=1}^{d_z}||\boldsymbol{z}_d||_1^2 = \sum_d(\sum_i|z_d^{[i]}|)^2 = \sum_{d,i,j}|z_d^{[i]}||z_d^{[j]}| \tag{35}$$

However, if the representation is nonnegative we can drop the absolute value. Then we can see this is a linear and therefore convex function of $\boldsymbol{Q}$:

$$\sum_{d,i,j}z_d^{[i]}z_d^{[j]} = \sum_{i,j}\boldsymbol{z}^{[i],T}\boldsymbol{z}^{[j]} = \sum_{i,j}Q_{i,j} = \mathbf{1}^T\boldsymbol{Q}\mathbf{1} \tag{36}$$

**O3: L2 Input Weights**   (under the constrain that the relationship between input data and representation is affine) We know that $\boldsymbol{Z} = \boldsymbol{W}_{\text{in}}\boldsymbol{X} + \boldsymbol{b}_{\text{in}}\mathbf{1}^T$. The work of mapping the mean between these two representations can be at least partly done by the bias, correcting any errors left by the linear map. Therefore the min-norm choice of linear map just maps the demeaned data to the demeaned representation:

$$\boldsymbol{W}_{\text{in}} = \bar{\boldsymbol{Z}}\bar{\boldsymbol{X}}^\dagger \tag{37}$$

This makes its frobenius norm:

$$\operatorname{Tr}[\bar{\boldsymbol{X}}^\dagger\bar{\boldsymbol{X}}^{T,\dagger}\bar{\boldsymbol{Z}}^T\bar{\boldsymbol{Z}}] = \operatorname{Tr}[\bar{\boldsymbol{X}}^\dagger\bar{\boldsymbol{X}}^{T,\dagger}(\boldsymbol{Z}-\boldsymbol{Z}\mathbf{1}\mathbf{1}^T)^T(\boldsymbol{Z}-\boldsymbol{Z}\mathbf{1}\mathbf{1}^T)] = \operatorname{Tr}[\bar{\boldsymbol{X}}^\dagger\bar{\boldsymbol{X}}^{T,\dagger}\boldsymbol{Q}] \tag{38}$$

since $\bar{\boldsymbol{X}}\mathbf{1} = \mathbf{0}$, and the columns of $\bar{\boldsymbol{X}}^\dagger$ has the same span as the rows of $\bar{\boldsymbol{X}}$, so $\mathbf{1}^T\bar{\boldsymbol{X}}^\dagger = \mathbf{0}$. This is a linear, and therefore convex, function of $\boldsymbol{Q}$.

**O4: L2 Readout Weights**   Similarly to the previous result, the min-norm readout linear map has to map the demeaned representation to the demeaned data:

$$\boldsymbol{W}_{\text{out}}\bar{\boldsymbol{Z}} = \bar{\boldsymbol{Y}} \tag{39}$$

The min-norm choice is given by the pseudoinverse:

$$\boldsymbol{W}_{\text{out}} = \bar{\boldsymbol{Y}}\bar{\boldsymbol{Z}}^\dagger \tag{40}$$

We have to use the pseudoinverse because $\boldsymbol{W}_{\text{out}}$ is only constrained in a subset of directions, in the others we can freely set the weights to zero. The best way to see these two spaces is, as in the previous derivation of the convexity of C3, to construct the reduced SVD of the de-meaned data matrix, $\bar{\boldsymbol{Y}} = \boldsymbol{U}\boldsymbol{S}\boldsymbol{V}^T$. For a rank $\rho$ matrix this leaves a diagonal square positive definite matrix, $\boldsymbol{S} \in \mathbb{R}^{\rho\times\rho}$, and two rectangular orthogonal matrices $\boldsymbol{U} \in \mathbb{R}^{d_x\times\rho}$, $\boldsymbol{V} \in \mathbb{R}^{T\times\rho}$. Now let's again break down the representation into the part within the span of $\boldsymbol{V}$ and the part living in the orthogonal complement, $\boldsymbol{V}_\perp \in \mathbb{R}^{T\times(d_Z-\rho)}$:

$$\bar{\boldsymbol{Z}} = \boldsymbol{A}\boldsymbol{V}^T + \boldsymbol{B}\boldsymbol{V}_\perp^T \qquad \boldsymbol{A} \in \mathbb{R}^{d_Z\times\rho} \quad \boldsymbol{B} \in \mathbb{R}^{d_Z\times(d_Z-\rho)} \tag{41}$$

In order to be linearly decodable $\boldsymbol{A}$ must be full column rank, and the columns of $\boldsymbol{A}$ must be linearly independent of the columns of $\boldsymbol{B}$.

Now we can develop the weight loss:

$$||\boldsymbol{W}_{\text{out}}||_F^2 = \operatorname{Tr}[\bar{\boldsymbol{X}}\bar{\boldsymbol{Z}}^\dagger(\bar{\boldsymbol{Z}}^\dagger)^T\bar{\boldsymbol{X}}^T] \tag{42}$$

Using the SVD of $\bar{\boldsymbol{Z}}$ it is clear that $\boldsymbol{Z}^\dagger(\bar{\boldsymbol{Z}}^\dagger)^T = (\bar{\boldsymbol{Z}}^T\bar{\boldsymbol{Z}})^\dagger = \bar{\boldsymbol{Q}}^\dagger$, giving:

$$||\boldsymbol{W}_{\text{out}}||_F^2 = \operatorname{Tr}[\boldsymbol{S}^2\boldsymbol{V}^T\bar{\boldsymbol{Q}}^\dagger\boldsymbol{V}] \tag{43}$$

To show this is a convex function we use the notion of matrix convexity. A function that maps to the set of positive semidefinite matrices, $S_+^m$ is matrix convex if:

$$f : \mathbb{R}^n \to S_+^m \qquad f(\theta m_1 + (1-\theta)m_2) \preccurlyeq \theta f(m_1) + (1-\theta)f(m_2) \qquad \forall m_1, m_2, \theta \in [0,1] \quad (44)$$

where $\preccurlyeq$ denotes the Loewner ordering on positive semidefinite matrices:

$$m_1, m_2 \in S_+^n \quad m_1 \preccurlyeq m_2 \quad \iff \quad m_2 - m_1 \in S_+^n \tag{45}$$

We quote the following result on the matrix convexity of the projection pseudoinverse (Silvey, 2013; Nordström, 2011):

$$\boldsymbol{V}^T(\lambda \boldsymbol{Q}_1 + (1-\lambda)\boldsymbol{Q}_2)^\dagger \boldsymbol{V} \preccurlyeq \lambda \boldsymbol{V}^T \boldsymbol{Q}_1^\dagger \boldsymbol{V} + (1-\lambda)\boldsymbol{V}^T \boldsymbol{Q}_2^\dagger \boldsymbol{V} \qquad \boldsymbol{Q}_1, \boldsymbol{Q}_2 \in S_+^n, \boldsymbol{V} \in \mathbb{R}^{n \times m} \quad (46)$$

Under the condition that the range of the projection is within the span of both matrices:

$$\mathbb{C}(\boldsymbol{V}) \subset \mathbb{C}(\boldsymbol{Q}_1) \cap \mathbb{C}(\boldsymbol{Q}_2) \tag{47}$$

where $\mathbb{C}$ denotes the column space or range of a matrix.

We can see that in our case this condition is satisfied due to the linear decodability assumption. To see this, let's start with a vector, $\boldsymbol{a} \in \mathbb{C}(\boldsymbol{V})$ and show that it is also in the columnspace of any linearly decodable representational similarity matrix. Since $\boldsymbol{a} \in \mathbb{C}(\boldsymbol{V})$, $\boldsymbol{a} = \boldsymbol{V}\boldsymbol{b}$. If $\boldsymbol{a}$ was not in $\mathbb{C}(\bar{\boldsymbol{Q}})$ then the following would be zero:

$$\bar{\boldsymbol{Q}}\boldsymbol{a} = \bar{\boldsymbol{Z}}^T \bar{\boldsymbol{Z}} \boldsymbol{V}\boldsymbol{b} = (\boldsymbol{V}\boldsymbol{A}^T + \boldsymbol{V}_\perp \boldsymbol{B}^T)\boldsymbol{A}\boldsymbol{V}^T \boldsymbol{V}\boldsymbol{b} + (\boldsymbol{V}\boldsymbol{A}^T + \boldsymbol{V}_\perp \boldsymbol{B}^T)\boldsymbol{B}\boldsymbol{V}_\perp^T \boldsymbol{V}\boldsymbol{b} = (\boldsymbol{V}\boldsymbol{A}^T + \boldsymbol{V}_\perp \boldsymbol{B}^T)\boldsymbol{A}\boldsymbol{b} \tag{48}$$

This comprises two terms living in orthogonal spaces, so both must be zero independently, i.e. $\boldsymbol{A}^T \boldsymbol{A}\boldsymbol{b} = 0$, and $\boldsymbol{B}^T \boldsymbol{A}\boldsymbol{b}$. Since $\boldsymbol{A}$ has full column rank, $\boldsymbol{A}^T \boldsymbol{A}$ is invertible, $\boldsymbol{A}^T \boldsymbol{A}\boldsymbol{b} = 0$ implies that $\boldsymbol{b} = 0$, which is also in the column space of $\boldsymbol{V}$, meaning the inclusion is satisfied.

Finally, if a function, $f$, is matrix convex then $\boldsymbol{x}^T f(x)\boldsymbol{x}$ is convex for all $\boldsymbol{x}$ (Boyd & Vandenberghe, 2004). We can see that this implies:

$$\|\boldsymbol{W}_{\text{out}}\|_F^2 = \text{Tr}[\boldsymbol{S}^2 \underbrace{\boldsymbol{V}^T \bar{\boldsymbol{Q}}^\dagger \boldsymbol{V}}_{f(\bar{\boldsymbol{Q}})}] = \sum_{i=1}^{\rho} s_k^2 \boldsymbol{e}_k^T f(\bar{\boldsymbol{Q}})\boldsymbol{e}_k \tag{49}$$

where $\boldsymbol{e}_k$ are canonical basis vectors and $s_k$ are the singular values of $\bar{\boldsymbol{X}}$. Each $\boldsymbol{e}_k^T f(\bar{\boldsymbol{Q}})\boldsymbol{e}_k$ is convex thanks to matrix convexity and since the sum of convex functions is also convex, we find that the weight loss is convex.

Finally, this is a function on demeaned representational similarity. Changing the mean changes $\boldsymbol{Q}$, but doesn't change the weight loss, so as a function of the full weight loss the readout weight are first a demeaning projection (it is easy to check this is a projection):

$$P : \boldsymbol{Q} \to \boldsymbol{Q} - \frac{1}{T}\mathbf{1}\mathbf{1}^T \boldsymbol{Q} - \frac{1}{T}\boldsymbol{Q}\mathbf{1}\mathbf{1}^T + \frac{1}{T^2}\mathbf{1}\mathbf{1}^T(\mathbf{1}^T \boldsymbol{Q}\mathbf{1}) \qquad \|\boldsymbol{W}_{\text{out}}\|_F^2 = \text{Tr}[\boldsymbol{S}^2 \boldsymbol{V}^T P(\boldsymbol{Q})\boldsymbol{V}] \tag{50}$$

Our previous convexity results then apply to the full similarity matrix.

**O5: Regularised Affine Reconstruction**  Given a fixed $\boldsymbol{Z}$, we consider the sum of an L2 affine reconstruction error and a weight regularisation, again using bar to denote demeaned:

$$\frac{1}{T}\|\boldsymbol{W}_{\text{out}}\bar{\boldsymbol{Z}} - \bar{\boldsymbol{Y}}\|_F^2 + \lambda_w \|\boldsymbol{W}_{\text{out}}\|_F^2 \tag{51}$$

One can optimise this with respect to the readout weights to find:

$$\boldsymbol{W}_{\text{out}}^* = \bar{\boldsymbol{Y}}\bar{\boldsymbol{Z}}^T(\lambda_w T\mathbb{1} + \bar{\boldsymbol{Z}}\bar{\boldsymbol{Z}}^T)^{-1} = \bar{\boldsymbol{Y}}(\lambda_w T\mathbb{1} + \bar{\boldsymbol{Z}}^T \bar{\boldsymbol{Z}})^{-1}\bar{\boldsymbol{Z}}^T \tag{52}$$

Inserting this into the objective and writing it in terms of $\bar{\boldsymbol{Q}}$ gives:

$$\text{Tr}\Big[\underbrace{\lambda_w(\lambda_w T\mathbb{1} + \bar{\boldsymbol{Q}})^{-1}\bar{\boldsymbol{Y}}^T \bar{\boldsymbol{Y}}(\lambda_w T\mathbb{1} + \bar{\boldsymbol{Q}})^{-1}\bar{\boldsymbol{Q}}}_{\text{weight loss}} + \frac{1}{T}\underbrace{((\lambda_w T\mathbb{1} + \bar{\boldsymbol{Q}})^{-1}\bar{\boldsymbol{Q}} - \mathbb{1})^T \bar{\boldsymbol{Y}}^T \bar{\boldsymbol{Y}}((\lambda_w T\mathbb{1} + \bar{\boldsymbol{Q}})^{-1}\bar{\boldsymbol{Q}} - \mathbb{1})}_{\text{reconstruction loss}}\Big]$$

$$\tag{53}$$

Inserting $\bar{Q} = \lambda_w T \mathbb{1} + \bar{Q} - \lambda_w T \mathbb{1}$ in the place of one of $\bar{Q}$s in the reconstruction loss let's you cancel the weight loss, leaving:

$$\mathcal{L}(Q) = \lambda_w \operatorname{Tr}[(\lambda_w T \mathbb{1} + \bar{Q})^{-1} \bar{Y}^T \bar{Y}] \tag{54}$$

To prove convexity we modify the proof technique of maths stack exchange user Robert Israel (Israel, 2013). A function, $\mathcal{L}$, of positive semi-definite matrix, $\bar{Q}$ is convex if for $\bar{Q}(t) = \bar{Q} + tS$ where $S$ is a symmetric matrix:

$$\left.\frac{d^2 \mathcal{L}(\bar{Q}(t))}{dt^2}\right|_{t=0} \geq 0 \tag{55}$$

Using the fact that:

$$(A + tS)^{-1} = A^{-1} - tA^{-1}SA^{-1} + t^2 A^{-1}SA^{-1}SA^{-1} \tag{56}$$

with the identification $A = \lambda_w T \mathbb{1} + \bar{Q}$ we find:

$$\left.\frac{d^2 \mathcal{L}(\bar{Q}(t))}{dt^2}\right|_{t=0} = \lambda_w \operatorname{Tr}[A^{-1}SA^{-1}SA^{-1}\bar{Y}^T \bar{Y}] \tag{57}$$

Finally, writing $C = SA^{-1}\bar{Y}^T$ this is:

$$\left.\frac{d^2 \mathcal{L}(\bar{Q}(t))}{dt^2}\right|_{t=0} = \lambda_w \operatorname{Tr}[C^T A^{-1} C] \tag{58}$$

But we know $A^{-1} = (\lambda_w T \mathbb{1} + Q)^{-1}$ is positive definite, hence this quantity is not only nonnegative, but positive. This means the loss is not just convex, but strictly convex, guaranteeing a single globally optimal $Q$ matrix.

**O6: Similarity Matching Objective**    In terms of the representational similarity this is:

$$\operatorname{Tr}[(X^T X - \alpha \mathbf{1}\mathbf{1}^T)Q] \tag{59}$$

This is a linear function of $Q$ so is convex.

**O7: Nonlinear Similarity Matching**    We consider the nonlinear similarity matching objective:

$$\operatorname{Tr}[G e^{Z^T Z}] = \operatorname{Tr}[G e^{Q}] \tag{60}$$

for a positive-definite input similarity kernel $G \in \mathbb{R}^{T \times T}$. We can take the second derivative of this with respect to $Q$:

$$\frac{\partial^2 \mathcal{L}}{\partial Q_{ab} \partial Q_{cd}} = \delta_{ac}\delta_{bd} e^{Q_{ab}} G_{ab} \tag{61}$$

Schoenberg (1942) showed that elementwise functions of a positive definite matrix are positive definite if the function is analytic with all positive coefficients. This tells us that the elementwise exponential of a postive semidefinite matrix is positive semidefinite. Similarly, the elementwise product of two postive semi-definite matrices is postive semi-definite. Hence this derivative is positive and the function is convex.

# B  NECESSARY AND SUFFICIENT IDENTIFIABILITY OF NONNEGATIVE AFFINE AUTOENCODING

Let's recall our optimisation problem:

**Problem 1** (Nonnegative Affine Autoencoding). *Let $s^{[i]} \in \mathbb{R}^{d_s}$, $x^{[i]} \in \mathbb{R}^{d_x}$, $z^{[i]} \in \mathbb{R}^{d_z}$, $W_{\text{in}} \in \mathbb{R}^{d_z \times d_x}$, $b_{\text{in}} \in \mathbb{R}^{d_z}$, $W_{\text{out}} \in \mathbb{R}^{d_x \times d_z}$, $b_{\text{out}} \in \mathbb{R}^{d_x}$, and $A \in \mathbb{R}^{d_x \times d_s}$ where $d_z > d_s$ and $d_x \geq d_s$, and $A$ is rank $d_s$. We seek the solution to the following the constrained optimization problem.*

$$\min_{W_{\text{in}}, b_{\text{in}}, W_{\text{out}}, b_{\text{out}}} \quad \left\langle ||z^{[i]}||_2^2 \right\rangle_i + \lambda \left( ||W_{\text{in}}||_F^2 + ||W_{\text{out}}||_F^2 \right)$$

$$\text{s.t.} \quad z^{[i]} = W_{\text{in}} x^{[i]} + b_{\text{in}}, \ x^{[i]} = W_{\text{out}} z^{[i]} + b_{\text{out}}, \ z^{[i]} \geq 0,$$

(62)

*where $i$ indexes a finite set of samples of $s$, and $x^{[i]} = As^{[i]}$.*

When will this optimisation problem produce a modular solution? In other words, when will each latent neuron be a function of only one source: $z_n^{[i]} = d_n(s_{k_n}^{[i]} - \min_i s_{k_n}^{[i]})$? We will present tight conditions on the dataset $\{s^{[i]}\}_i$ and mixing matrix $A$ such that, if this algorithm is properly optimised, the neurons will recover the sources. This is known as an identifiability result: a statement about what has to be true about the data such that an algorithm will, in principle, succeed in extracting the groundtruth sources.

**Tight Scattering Condition**  In this work we show that a necessary and sufficient condition for the nonnegative affine autoencoder to recover the true sources is the following tight scattering condition:

**Definition 1** (Tight Scattering). *Generate the de-meaned sources: $\bar{S} = S - \frac{1}{T} S \mathbf{1} \mathbf{1}^T \in \mathbb{R}^{d_S \times T}$ with elements $\bar{S}_{dt} = \bar{s}_d^{[t]}$. Assume wlog that $|\min_t \bar{s}_d^{[t]}| \leq \max_t \bar{s}_d^{[t]}$ for each dimension $d$ (if not satisfied simply redefine this source as $-s_d$). Construct the diagonal matrix $D \in \mathbb{R}^{d_S \times d_S}$ and the symmetric matrix $F^{d_S \times d_S}$:*

$$D_{jj} = \sqrt[4]{\frac{\lambda (A^T A)_{jj}}{\langle (\bar{s}_j^{[i]})^2 \rangle_i + (\min_i \bar{s}_j^{[i]})^2 + \lambda ((A^T A)^{-1})_{jj}}} \quad F = \lambda D^2 (A^T A) D^2 - \lambda (A^T A)^{-1} - \Sigma$$

(63)

*where $\Sigma = \frac{1}{T} \bar{S} \bar{S}^T$ is the covariance. Use these matrices to construct the following set:*

$$E = \{x | x^T F^{-1} x = 1\}$$

(64)

*$S$ is tightly scattered with respect to a generating matrix $A$ if the following conditions hold:*

- *$\text{Conv}(\bar{S}) \supseteq E$*

- *$\text{Conv}(\bar{S})^* \cap bdE^* = \{\lambda e_k, \lambda \neq 0 \in \mathbb{R}, k = 1, ..., d_S\}$ where the $*$ denotes the dual cone, and bd the boundary.*

These two conditions relate closely to the sufficient scattering conditions previously derived (Hu & Huang, 2023; Tatli & Erdogan, 2021a;b; Huang et al., 2013), but are adaptive to the structure of $A$. Finally, it is easy to check that the diagonal elements of $F$ are $F_{jj} = (\min_i [\bar{s}_j^{[i]}])^2$, which will be useful later.

**Main Result**  Our main theorem is that tight scattering is a necessary and sufficient condition for biological linear autoencoders to recover linearly mixed sources.

**Theorem 1** (Identifiability of Nonnegative Affine Autoencoders). *Given a dataset $X = AS$, if the matrix $\bar{S}$ is tightly scattered with respect to $A$ then the optimal biological linear autoencoder recovers the sources - each neuron's activity is an affine function of one source and every source has at least one neuron encoding it:*

$$z_n^{[i]} = d_n(s_n^{[i]} - \min_i [s_n^i])$$

(65)

Our proof takes the following logic: we find the optimal modular solution (appendix B.1), then we locally perturb into the nearby mixed representations. We derive the above conditions as those that

determine when the optimal modular solution is in fact also a local minima of the full representation, appendix B.2. Since the problem is convex, section 2, it is also then the global minima. We then show that if these conditions hold there are no non-modular global minima, i.e. the globally minimum representations are all modular. Finally, we show that in the special case where the sources are mixed orthogonally we recover the results found by Dorrell et al. (2025) (appendix B.4).

## B.1 Optimal Modular Solution

An optimal modular representation can be implemented by $d_s$ neurons, one for each source. Let's first demean the sources, which doesn't change the problem (since we can add or remove a bias arbitrarily), but makes notation easier. Then:

$$z_M^{[i]} = \sum_{j=1}^{d_S} d_j(\bar{s}_j^{[i]} - \min_i \bar{s}_j^{[i]})e_j \tag{66}$$

Where $e_j$ are the canonical basis vectors, $d_j$ represents the strength of the encoding of each source, and the subscript M denotes the fact it is modular. Stacking the $d_j$ elements into a diagonal matrix $D$ we can derive each of the three loss terms. We know that:

$$z_M^{[i]} = W_{\text{in}}x^{[i]} + b_{\text{in}} = W_{\text{in}}A\bar{s}^{[i]} + b_{\text{in}} = D\bar{s}^{[i]} + b_{\text{in}} \tag{67}$$

Hence the min L2 norm weight matrices are:

$$W_{\text{in}} = DA^\dagger \qquad W_{\text{out}} = W_{\text{in}}^\dagger = AD^{-1} \tag{68}$$

where $\dagger$ denotes the pseudoinverse. Hence:

$$||W_{\text{in}}||_F^2 = \text{Tr}[W_{\text{in}}^T W_{\text{in}}] = \text{Tr}[D^2 A^\dagger (A^\dagger)^T] = \text{Tr}[D^2 O_A^{-1}] \tag{69}$$

$$||W_{\text{out}}||_F^2 = \text{Tr}[D^{-2} O_A] \tag{70}$$

Where we've defined $O_A = A^T A$, and then $A^\dagger (A^\dagger)^T = O_A^{-1}$, as can be seen using the SVD.

Denote with $\alpha_j^2$ the diagonal elements of $O_A$, positive numbers representing how strongly a particular source is encoded in the data. Further, denote with $(\alpha_j')^2$ the diagonal elements of $O_A^{-1}$, these represent the lengths of the 'pseudoinverse vectors'. If the encodings of each source are orthogonal they are just the inverse of the encoding sizes, $\alpha_j$, if not, they also tell you how 'entangled' a source's representation is. In terms of these variables we can write the whole loss as:

$$\mathcal{L} = \sum_{j=1}^{d_s} d_j^2[\langle (s_j^{[i]})^2 \rangle_i + (\min_i s_j^{[i]})^2 + \lambda(\alpha_j')^2] + \frac{\lambda \alpha_j^2}{d_j^2} \tag{71}$$

Then the optimal choice of encoding sizes is:

$$(d_j^*)^4 = \frac{\lambda \alpha_j^2}{\langle (s_j^{[i]})^2 \rangle_i + (\min_i s_j^{[i]})^2 + \lambda(\alpha_j')^2} \tag{72}$$

This corresponds to the elements of the matrix $D$ from the definition of tight scattering, definition 1, hence why we gave them the same name.

## B.2 Tight Scattering is a Necessary and Sufficient Condition for the Modular Representation to be a Global Minima

Now let's just go balls to the wall and take the derivative of the whole loss with respect to the linear map that relates the neurons to the sources, while being careful to preserve the positivity of the representation.

$$z^{[i]} = W\bar{s}^{[i]} - \min_i[W\bar{s}^{[i]}] \tag{73}$$

where the minima is taken neuron-wise. Then the min-norm weight matrices are: $W_{\text{in}} = WA^\dagger$, $W_{\text{out}} = AW$, so, defining the covariance matrix $\Sigma = \langle \bar{s}^{[i]} \bar{s}^{[i],T} \rangle_i$, we can write the loss as:

$$\mathcal{L}(W) = \lambda \text{Tr}[W^T W O_A^{-1}] + \lambda \text{Tr}[(W^T W)^{-1} O_A] + \text{Tr}[\Sigma W^T W] + \sum_n (\min_i[w_n \bar{s}^{[i]}])^2 \tag{74}$$

where the min is now just the min of a set of scalars, and we've denoted with $\boldsymbol{w}_n$ the $n$th row of $\boldsymbol{W}$. Then the (sub)derivative is:

$$\frac{1}{2}\frac{\partial \mathcal{L}}{\partial \boldsymbol{W}} = \lambda \boldsymbol{W} \boldsymbol{O}_A^{-1} - \lambda \boldsymbol{W}(\boldsymbol{W}^T\boldsymbol{W})^{-1}\boldsymbol{O}_A(\boldsymbol{W}^T\boldsymbol{W})^{-1} + \boldsymbol{W}\boldsymbol{\Sigma} + \sum_n \min_i[\boldsymbol{w}_n^T\boldsymbol{s}^{[i]}]\frac{\partial \min_i[\boldsymbol{w}_n^T\boldsymbol{s}^{[i]}]}{\partial \boldsymbol{W}} \tag{75}$$

Let's evaluate this at the optimal modular representation, which with a slight abuse of notation is $\boldsymbol{W} = \boldsymbol{D}$ (previously $\boldsymbol{D} \in \mathbb{R}^{d_S \times d_S}$, but now we'll pad it with zeros so it is $\boldsymbol{D} \in \mathbb{R}^{N \times d_S}$ like $\boldsymbol{W}$). The biggest immediate change is in the minima term; since:

$$\text{for} \quad n < d_S \quad \boldsymbol{w}_n = \boldsymbol{e}_n d_n \qquad \text{so} \quad \min_i[\boldsymbol{w}_n^T\bar{\boldsymbol{s}}^{[i]}] = d_n \min_i[\bar{s}_n^{[i]}] \tag{76}$$

Further, the subderivative of the minima of a set of convex functions is within the convex hull of the derivatives of the functions that achieve that minima:

$$f(x) = \min_i f_i(x) \qquad \partial f(x) \in \text{Convex Hull}(\{\partial f_i(x)|f_i(x) = f(x)\}) \tag{77}$$

So we know that:

$$\frac{\partial \min_i[\boldsymbol{w}_n^T\boldsymbol{s}^{[i]}]}{\partial \boldsymbol{w}_{n'}} = \delta_{n,n'}\boldsymbol{a}_n \qquad \boldsymbol{a}_n \in \text{Convex Hull}(\{\boldsymbol{s}^{[i]}|s_n^{[i]} = \min_j s_n^{[j]}\}) \tag{78}$$

i.e. the subdifferential vector $\boldsymbol{a}_n$ lives in the convex hull of the bounding datapoints in direction $n$. Now we want to know if the derivative is zero. Let's look at the gradient with respect to one row of the matrix $\boldsymbol{W}$, $\boldsymbol{w}_n$:

$$\frac{1}{2}\frac{\partial \mathcal{L}}{\partial \boldsymbol{w}_n}\bigg|_{\boldsymbol{W}=\boldsymbol{D}} = \lambda d_n \boldsymbol{e}_n^T \boldsymbol{O}_A^{-1} - \lambda d_n \boldsymbol{e}_n^T(\boldsymbol{D}^T\boldsymbol{D})^{-1}\boldsymbol{O}_A(\boldsymbol{D}^T\boldsymbol{D})^{-1} + d_n \boldsymbol{e}_n^T\boldsymbol{\Sigma} + d_n \min_i[s_n^{[i]}]\boldsymbol{a}_n \tag{79}$$

Therefore, an equivalent question to establishing whether this gradient is zero is the following question, is this vector in the convex hull of the axis-bounding points?

$$\frac{1}{-\min_i[s_n^{[i]}]}\underbrace{\left(\lambda(\boldsymbol{D}^T\boldsymbol{D})^{-1}\boldsymbol{O}_A(\boldsymbol{D}^T\boldsymbol{D})^{-1} - \lambda\boldsymbol{O}_A^{-1} - \boldsymbol{\Sigma}\right)}_{\boldsymbol{F}}\boldsymbol{e}_n \in \text{Convex Hull}(\{\boldsymbol{s}^{[i]}|s_n^{[i]} = \min_j s_n^{[j]}\}) \tag{80}$$

We'll now show that, if the sources are tightly scattered, definition 1, the answer is yes. First we do some work to identify the vector on the left-hand side as a point on the ellipse is maximally displaced in the direction $-\boldsymbol{e}_n$. We consider a vector on the ellipse, $\boldsymbol{v}_{\boldsymbol{x}}$, that is maximally displaced along some vector $\boldsymbol{x}$. To find $\boldsymbol{v}_{\boldsymbol{x}}$ we maximise $\boldsymbol{v}^T\boldsymbol{x}$ subject to the constraint that $\boldsymbol{v}_{\boldsymbol{x}}^T\boldsymbol{F}^{-1}\boldsymbol{v}_{\boldsymbol{x}} = 1$. Performing the resulting lagrange optimisation tells us that:

$$\boldsymbol{v}_{\boldsymbol{x}} = \frac{\boldsymbol{F}\boldsymbol{x}}{\sqrt{\boldsymbol{x}^T\boldsymbol{F}\boldsymbol{x}}} \tag{81}$$

Hence using $\boldsymbol{x} = -\boldsymbol{e}_n$ and the definition of $\boldsymbol{F}$, we get:

$$\boldsymbol{v}_{-\boldsymbol{e}_n} = \frac{1}{-\min_i[s_n^{[i]}]}\boldsymbol{F}\boldsymbol{e}_n \qquad [\boldsymbol{v}_{-\boldsymbol{e}_n}]_n = \min_i[s_n^{[i]}] \tag{82}$$

and we see that this point touches the bounding box $\min_i[s_n^i]$ along the $n$th axis. Therefore, if the ellipse is contained in the convex hull of the data, in particular if the ellipse kisses the ellipse at the minimising point along each axis, which it has to, then the gradient is zero.

So that's good, but we have to go to second order unfortunately. We can derive the horrendous second derivative in index notation, using Einstein summation convention, and $\delta_{ij}$ - the Kronecker delta:

$$\frac{\frac{1}{2}\partial^2 \mathcal{L}}{\partial W_{ij}\partial W_{kl}} = -\lambda\delta_{ik}(\boldsymbol{W}^T\boldsymbol{W})_{l\beta}^{-1}(\boldsymbol{O}_A)_{\beta\gamma}(\boldsymbol{W}^T\boldsymbol{W})_{\gamma j}^{-1} + +\delta_{ik}(\Sigma_{lj} + \lambda(\boldsymbol{O}_A)_{lj}^{-1})$$

$$+ 2\lambda W_{i\alpha}W_{k\delta}(\boldsymbol{W}^T\boldsymbol{W})_{\delta\alpha}^{-1}(\boldsymbol{W}^T\boldsymbol{W})_{l\beta}^{-1}(\boldsymbol{O}_A)_{\beta\gamma}(\boldsymbol{W}^T\boldsymbol{W})_{\gamma j}^{-1}$$

$$+2\lambda W_{i\alpha}(\boldsymbol{W}^T\boldsymbol{W})_{\alpha\beta}^{-1}(\boldsymbol{O}_A)_{\beta\gamma}W_{k\delta}(\boldsymbol{W}^T\boldsymbol{W})_{\delta\gamma}^{-1}(\boldsymbol{W}^T\boldsymbol{W})_{lj}^{-1} + \sum_n \delta_{ik}\frac{\partial \min_i[\boldsymbol{w}_n^T\boldsymbol{s}^{[i]}]}{\partial W_{ij}}\frac{\partial \min_i[\boldsymbol{w}_n^T\boldsymbol{s}^{[i]}]}{\partial W_{kl}} \tag{83}$$

Now, this is a godawful mess. Fortunately, I derived it not you, and we don't have to conclude much from it. First notice that most terms have a $\delta_{ik}$, this means they do not couple the perturbations of different neurons. There are only two terms that do. But, notice another aspect of both these two terms: they contain terms of the type $\boldsymbol{W}_{i\alpha}$ and $\boldsymbol{W}_{k\delta}$. We are going to evaluate this second derivative at $\boldsymbol{W} = \boldsymbol{D}$, and recall that the first $d_S$ rows of $\boldsymbol{D}$ are a diagonal matrix, and the rest are zeros. So, if both $i$ and $k$, the neuron indices, are below $d_S$, the loss will couple, but for all extra neurons, $i > d_S$, these two coupling terms will be zero, and they will contribute independently!

Therefore we split our evaluation of this quantity at the point $\boldsymbol{W} = \boldsymbol{D}$ into an analysis of the indices $i, k \leq d_s$, and the others. Let's begin with the more boring, $i, k \leq d_s$. Now there's no summation convention being applied any more, every quantity is just a scalar.

$$
\begin{aligned}
\frac{\frac{1}{2}\partial^2 \mathcal{L}}{\partial W_{ij}\partial W_{kl}} = {} & \delta_{ik}(\Sigma_{lj} + \lambda(\boldsymbol{O}_A)^{-1}_{lj}) + \lambda\delta_{ik}(d_l^*)^{-2}(\boldsymbol{O}_A)_{lj}(d_j^*)^{-2} \\
& + 2\lambda\delta_{lj}(d_j^*)^{-2}(d_i^*)^{-1}(\boldsymbol{O}_A)_{ik}(d_k^*)^{-1} + \sum_k \delta_{ik}[\boldsymbol{a}_k]_j[\boldsymbol{a}_k]_l
\end{aligned}
\tag{84}
$$

First notice that every term contains a kronecker delta over two of the indices and a positive semi-definite matrix over the other two[3]. This is all a bit of a pain to look at because we currently have a tensor with four indices. To turn it clean matrices we have to vectorise over the two matrix indices. Then each term is just the Kronecker product of two positive semi-definite matrices, since the Kronecker delta is just the identity matrix in index land. And, since the Kronecker product of two positive definite matrices is itself positive definite, this quantity will be too! Finally, we see that the 2nd derivative of the loss with respect to this portion of the linear map is positive definite, therefore all perturbations in these direction increase the loss.

Alternatively, we can take the case of a new neuron, $i = k = n$:

$$
\frac{\frac{1}{2}\partial^2 \mathcal{L}}{\partial W_{ni}\partial W_{nj}} = \underbrace{\Sigma_{ij} - \lambda(\boldsymbol{O}_A)^{-1}_{ij} + \lambda(d_i^*)^{-2}(\boldsymbol{O}_A)_{ij}(d_j^*)^{-2}}_{-F_{ij}} + \boldsymbol{a}_n\boldsymbol{a}_n^T
\tag{85}
$$

Again, $\boldsymbol{a}_n$ lives in the convex hull of the datapoints that minimise $\min_i(\boldsymbol{w}_n^T\boldsymbol{s}^{[i]})$, but we evaluated the derivative at the point $\boldsymbol{W} = \boldsymbol{D}$, i.e. where $\boldsymbol{w}_n = \boldsymbol{0}$. As such, all datapoints satisfy this minimia, and $\boldsymbol{a}_n$ is just a point that lives in the convex hull of the data.

To check whether this quantity will ever be zero we take the directional derivative along some direction $\boldsymbol{w}$:

$$
\boldsymbol{w}^T \frac{\frac{1}{2}\partial^2 \mathcal{L}}{\partial \boldsymbol{w}_n \partial \boldsymbol{w}_n} \boldsymbol{w} = -\boldsymbol{w}^T \boldsymbol{F}\boldsymbol{w} + (\boldsymbol{w}^T\boldsymbol{a}_n)^2
\tag{86}
$$

So whether this quantity is negative, positive, or 0, all depends on the convex hull of the data, within which $\boldsymbol{a}_n$ lies. If we are able to choose $\boldsymbol{a}_n$ as the maximising point on the ellipse in the direction $\boldsymbol{w}$:

$$
\boldsymbol{a}_n = \frac{\boldsymbol{F}\boldsymbol{w}}{\sqrt{\boldsymbol{w}^T \boldsymbol{F}\boldsymbol{w}}}
\tag{87}
$$

Then the derivative is clearly zero. If $\boldsymbol{a}_n$ is larger in the same direction the gradient is positive, and smaller it is negative.

Therefore the first sufficient scattering condition, theorem 1, which guarantees that $E \subseteq$ Convex Hull$(\{\boldsymbol{s}^{[i]}\}_i)$, means that this quantity is nonnegative. Further, were this scattering condition not satisfied we would have found a direction which would reduce the loss, hence this condition is a necessary condition for the modular solution to be optimal.

And it is the second scattering condition that circumscribes the set of cases in which this gradient can be 0. The only allowed points of tangency between the convex hull and $E$ are those that are extremal along a basis direction. This is saying two things. First, in all other directions the gradient is positive, and we cannot decrease the loss. Second, we can create a new neuron with $\boldsymbol{w} \propto \boldsymbol{e}_k \forall k = 1, ..., d_S$ and it won't change the loss, since the gradient is zero. Since this new neuron is a function of only

---

[3]$\boldsymbol{\Sigma}$ is positive definite, each $d_l^*$ is a positive number, $\boldsymbol{O}_A = \boldsymbol{A}^T\boldsymbol{A}$ so it and its inverse are positive definite too, and $\boldsymbol{a}_k\boldsymbol{a}_k^T$ is clearly positive semi-definite

a single source, this preserves the optimal representation's modularity. Were there more points of tangency, i.e. the second scattering condition was broken, there would be other perturbations that would leave the loss unchanged but introduce non-modularity, so we find that this condition is also necessary.

In sum, we find that, if both conditions hold, the modular representation is a local optima, while each condition is necessary. Hence the pair of conditions are the necessary and sufficient conditions for the optimal modular representation to be a local minima of the loss. Since the problem is a convex optimisation problem over the set of representational similarity matrices, $Q = Z^T Z$, and a locally optimal linear map implies a locally optimal $Q$, we know that, when these conditions hold, the optimal modular representation is not just a local minima but a global minima.

### B.3 WHEN IS THE OPTIMAL MODULAR SOLUTION *the* GLOBAL MINIMA

In section 2 we saw that out problem is a convex optimisation problem. We have found a globally minimal solution, our question is now: are there any other globally minimal solutions and if so are they also modular? We will show that under the tight scattering conditions all global minima are just permutations of the optimal modular solution.

All our representations are affine function of the demeaned data: $Z = W\bar{X} + b\mathbf{1}^T$. Further, we saw in appendix A that, using the reduced SVD of the demeaned data matrix, $\bar{X} = USV^T$, the readout weight loss could be written:

$$||W_{\text{out}}||_F^2 = \text{Tr}[S^2 V^T \bar{Q}^\dagger V] \tag{88}$$

where, denoting the rank of $\bar{X}$ with $\rho$, $U \in \mathbb{R}^{d_x \times \rho}$, $S \in \mathbb{R}^{\rho \times \rho}$, $V \in \mathbb{R}^{d_z \times \rho}$. Using the properties of the pseudoinverse of products:

$$\bar{Q}^\dagger = (\bar{X}^T W^T W \bar{X})^\dagger = (VSU^T W^T WUSV^T)^\dagger = V^T S^{-1}(U^T W^T WU)^\dagger S^{-1} V \tag{89}$$

Affine decodability ensures that $U^T W^T WU$ is full rank, leading to a simplified form of the loss:

$$||W_{\text{out}}||_F^2 = \text{Tr}[(U^T W^T WU)^{-1}] \tag{90}$$

Further, let's break $W$ into two parts, those aligning with $U$ and those orthogonal:

$$W = W_\rho U + W_\perp U_\perp \tag{91}$$

The labels are zero along the $U_\perp$ directions, and so there is no value to a non-zero $W_\perp$, hence, the loss becomes simply:

$$||W_{\text{out}}||_F^2 = \text{Tr}[(W_\rho^T W_\rho)^{-1}] \tag{92}$$

But, thanks to the positive definiteness of $W_\rho^T W_\rho$ this is now not just a convex function of $W^T W$, but a strictly convex one (Israel, 2013). This means that there is a uniquely optimal $W^T W$: any other globally optimal representation $Z'$ must therefore have an input linear map that is a(n) (semi-)orthogonal transform of the optimal modular solution's, $W^*$:

$$Z' = OW^* \bar{X} + b'\mathbf{1}^T = \geq 0 \tag{93}$$

Further, in order to be optimal $Z$ must have the same loss as $Z_M$. We already know that they have the same readin and readout loss, since these are both fixed by the choice of readin transform $OW^*$, therefore they must also have the same activity loss. The activity loss of this new representation is:

$$\frac{1}{T}\text{Tr}[Z'^T Z'] = \frac{1}{T}\text{Tr}[\bar{X}\bar{X}^T W^{*,T} W^*] + ||b'||^2 \tag{94}$$

Therefore we conclude that in order to have the same activity loss the new optimal solution's bias vector must have the same length as the original: $b' = O'b^*$. Further, we know that in order for $Z'$ to be an optimal representation $b' = -\min_i[OW^* x^{[i]}] = -\min_i[O\bar{z}_M^{[i]}]$, where $\bar{z}_M^{[i]}$ is the demeaned optimal modular latent. Therefore, the puzzle becomes which rotations and reflections, $O$, can be apply to $\bar{z}_M^{[i]}$ such that the length of the minimal bias vector doesn't change? We'll now show that if the sources are tightly scattered $O$ must be a permutation.

Since the sources obey the tight scattering assumptions, the optimal modular representation is tightly scattered with respect to a different ellipse, $E_z$:

$$E_z = DE = \{\bar{z}|\bar{z} = Dy, \quad y^T F^{-1} y = 1\} = \{\bar{z}|\bar{z}^T D^{-1} F^{-1} D^{-1}\bar{z} = 1\} \tag{95}$$

Let's call the $n$th row of the orthogonal matrix $\boldsymbol{o}_n$, then we need to find $-\min_i[\boldsymbol{o}_n^T \bar{v} z^{[i]}]$. Since the convex hull of the demeaned latents supersets $E_z$, we can upper bound this quantity $-\min_{\boldsymbol{y} \in E_z}[\boldsymbol{o}_n^T \boldsymbol{y}]$. The element $\boldsymbol{y}$ that achieves this can be calculated through lagrange optimisation to be:

$$-\frac{\boldsymbol{DFDo}_n}{\sqrt{\boldsymbol{o}_n^T \boldsymbol{DFDo}_n}}, \quad \text{hence} \quad -\min_{\boldsymbol{y} \in E_z}[\boldsymbol{o}_n^T \boldsymbol{y}] = \sqrt{\boldsymbol{o}_n^T \boldsymbol{DFDo}_n} \tag{96}$$

Therefore we can calculate a lower bound on $||\boldsymbol{b}'||$, which can be simplified using the orthogonality of $\boldsymbol{O}$:

$$||\boldsymbol{b}'||^2 \geq \sum_n (-\min_{\boldsymbol{y} \in E_z}[\boldsymbol{o}_n^T \boldsymbol{y}])^2 = \sum_n \boldsymbol{o}_n^T \boldsymbol{DFDo}_n = \text{Tr}[\boldsymbol{D}^2 \boldsymbol{F}] = \sum_j D_{jj}^2 F_{jj} = \sum_j (d_j^*)^2 (\min_i[\bar{s}_j^{[i]}])^2 = ||\boldsymbol{b}^*||^2 \tag{97}$$

where the penultimate equality follows from the fact that the diagonal elements of $\boldsymbol{F}$ are $(\min_i[\bar{s}_j^{[i]}])^2$, definition 1. So in fact this lower bound has to be achieved.

Further, we can now use the second tight scattering condition on the intersection of the boundary to constrain this orthogonal matrix. The convex hull of the data only touches the ellipse at points whose tangents are orthogonal to a basis direction. If $\boldsymbol{o}_n$ is not a basis direction then the inequality is strict: $(\min_i[\boldsymbol{o}_n^T \boldsymbol{z}^{[i]}])^2 > (\min_{\boldsymbol{y} \in E_Z}[\boldsymbol{o}_n^T \boldsymbol{y}])^2$, the bias vectors cannot be the same length, and the solution cannot be optimal. Therefore we derive that each $\boldsymbol{o}_n$ must be along a basis vector. The only way to achieve this is if the orthogonal matrix is a permutation.

## B.4 ORTHOGONAL ENCODING SPECIAL CASE

Finally, as a corollary we show that the original orthogonally result recovered by Dorrell et al. (2025) fall out under the right conditions. To do this we assume $\boldsymbol{x}^{[i]} = \boldsymbol{s}^{[i]}$. This occurs when $\boldsymbol{A} = \boldsymbol{I}$, $\boldsymbol{O}_A = \boldsymbol{O}_A^{-1} = \boldsymbol{I}$; then:

$$(d_j^*)^4 = \frac{\lambda}{\lambda + \langle (s_j^{[i]})^2 \rangle_i + (\min_i s_j^{[i]})^2} \qquad (d_j^*)^{-4} = 1 + \frac{1}{\lambda}(\langle (s_j^{[i]})^2 \rangle_i + (\min_i s_j^{[i]})^2) \tag{98}$$

Putting these into the definition of $\boldsymbol{F}$:

$$\boldsymbol{F} = \lambda \boldsymbol{D}^{-4} - \lambda - \langle \bar{\boldsymbol{s}}^{[i]}(\bar{\boldsymbol{s}}^{[i]})^T \rangle_i = \begin{bmatrix} (\min_i s_1^{[i]})^2 & -\langle s_1^{[i]} s_2^{[i]} \rangle_i & \cdots \\ -\langle s_1^{[i]} s_2^{[i]} \rangle_i & (\min_i s_2^{[i]})^2 & \cdots \\ \vdots & \vdots & \ddots \end{bmatrix} \tag{99}$$

Exactly the $\boldsymbol{F}$ matrix from (Dorrell et al., 2025).

## B.5 RELAXATION OF NUMBER OF NEURONS CONSTRAINT

To show the convexity of our problems we had to assume the number of neurons was larger than the number of datapoints: $d_Z \geq T$. Restricting the number of neurons further will never improve the globally optimal solution. Therefore, if we can find a globally optimal solution to the many-neuron convex problem that uses a small number of neurons, $n$, it will also be a globally optimal solution to any version of the problem with the neuron a neuron constraint $d_Z \geq n$. This is the case in this problem: under the tight scattering conditions a modular solution using only $d_S$ neurons is optimal. Further, if the tight scattering conditions are not satisfied all we need is one extra neuron whose activity we can perturb to reduce the loss. This shows that tight scattering is still the necessary and sufficient condition for modularising when as long as $d_Z > d_S$, as stated in theorem 1.

## C    EXTENSION TO IMPERFECT RECONSTRUCTION

Here we extend our identifiability results to the imperfect reconstruction setting, i.e. fitting a non-negative affine autoencoder to data using a combination of a reconstruction and regularisation loss

**Problem 2** (Imperfect Nonnegative Affine Autoencoding). *Let $s^{[i]} \in \mathbb{R}^{d_s}$, $x^{[i]} \in \mathbb{R}^{d_x}$, $z^{[i]} \in \mathbb{R}^{d_z}$, $W_{\text{in}} \in \mathbb{R}^{d_z \times d_x}$, $b_{\text{in}} \in \mathbb{R}^{d_z}$, $W_{\text{out}} \in \mathbb{R}^{d_x \times d_z}$, $b_{\text{out}} \in \mathbb{R}^{d_x}$, and $A \in \mathbb{R}^{d_x \times d_s}$ where $d_z > d_s$ and $d_x \geq d_s$, and $A$ is rank $d_s$. We seek the solution to the following the constrained optimization problem.*

$$\min_{W_{\text{in}},b_{\text{in}},W_{\text{out}},b_{\text{out}}} \quad \lambda_a \left\langle ||z^{[i]}||_2^2 \right\rangle_i + \lambda_w \left( ||W_{\text{in}}||_F^2 + ||W_{\text{out}}||_F^2 \right) + \langle ||W_{out}z^{[i]} + b_{out} - x^{[i]}||^2 \rangle_i$$
$$\text{s.t.} \quad z^{[i]} = W_{\text{in}}x^{[i]} + b_{\text{in}}, \ z^{[i]} \geq 0,$$
$$\tag{100}$$

*where $i$ indexes a finite set of samples of $s$, and $x^{[i]} = As^{[i]}$.*

We ask when the optimal solution modularises, i.e. under which conditions does the optimal solution use a disjoint set of neurons to encode each source. We derive a related tight scattering condition, adapts to the optimal imperfect reconstruction.

**Definition 2** (Tight Scattering). *Generate the de-meaned sources: $\bar{S} = S - \frac{1}{T}S11^T \in \mathbb{R}^{d_S \times T}$ with elements $\bar{S}_{dt} = \bar{s}_d^{[t]}$. Assume wlog that $|\min_t \bar{s}_d^{[t]}| \leq \max_t \bar{s}_d^{[t]}$ for each dimension $d$ (if not satisfied simply redefine this source as $-s_d$). Numerically find the diagonal positive definite matrix, $D \in \mathbb{R}^{d_S \times d_S}$ that minimises the following loss:*

$$\mathcal{L} = \text{Tr}[D^2(\lambda_a M^2 + \lambda_a \Sigma + \lambda_w(A^T A)^{-1})] + \lambda_w \text{Tr}[(\lambda_w \Sigma^{-1} + D^2)^{-1}(A^T A)] \tag{101}$$

*Use that to construct the following symmetric matrix, $F^{d_S \times d_S}$:*

$$F = \frac{\lambda_w}{\lambda_a}(\lambda_w \Sigma^{-1} + D^2)^{-1}A^T A(\lambda_w \Sigma^{-1} + D^2)^{-1} - \frac{\lambda_w}{\lambda_a}(A^T A)^{-1} - \Sigma \tag{102}$$

*$S$ is tightly scattered with respect to a generating matrix $A$ if the following conditions hold with respect to the set $E = \{x | x^T F^{-1} x = 1\}$.*

- *$\text{Conv}(\bar{S}) \supseteq E$*

- *$\text{Conv}(\bar{S})^* \cap bdE^* = \{\lambda e_k, \lambda \neq 0 \in \mathbb{R}, k = 1, ..., d_S\}$ where the $*$ denotes the dual cone, and bd the boundary.*

Our theorem is the natural extension of theorem 1. If the data is tightly scattered with respect to this set, the optimal imperfect nonnegative affine autoencoder, as in problem 2, is modular:

**Theorem 3** (Identifiability of Imperfect Nonnegative Affine Autoencoders). *Given a dataset $X = AS$, if the matrix $\bar{S}$ is tightly scattered with respect to $A$ then the optimal biological linear autoencoder recovers the sources - each neuron's activity is an affine function of one source and every source has at least one neuron encoding it:*

$$z_n^{[i]} = d_n(s_n^{[i]} - \min_i[s_n^i]) \tag{103}$$

The proof logic is as before, appendix B, we first derive the optimal modular solution, which is encoded in the matrix $D$. Then we perturb the solution and show the above condition arises as the one that makes the optimal modular a local and therefore global solution. Finally, we take the $\lambda_a$ and $\lambda_w$ small, but constant ratio, limit to recover the previous result.

### C.1    OPTIMAL MODULAR SOLUTION

We consider the set of modular representation:

$$z_M^{[i]} = \sum_{j=1}^{d_S} d_j(\bar{s}_j^{[i]} - \min_i \bar{s}_j^{[i]})e_j \tag{104}$$

Denote with $\boldsymbol{D}$ the diagonal matrix made from these weightings. We can write the loss in terms of $\boldsymbol{D}$. First note that using $\bar{\boldsymbol{Y}} = \boldsymbol{A}\bar{\boldsymbol{S}}$ and $\bar{\boldsymbol{Z}} = \boldsymbol{D}\bar{\boldsymbol{S}}$, then, using the SVD or otherwise, we can rewrite the combination of output weight loss and reconstruction loss, O5 in appendix A.3, as:

$$\mathrm{Tr}[(\lambda_w T \mathbb{1} + \bar{\boldsymbol{Q}})^{-1}\bar{\boldsymbol{Y}}^T\bar{\boldsymbol{Y}}] = \mathrm{Tr}[(\lambda_w \Sigma^{-1} + \boldsymbol{D}^2)^{-1}(\boldsymbol{A}^T\boldsymbol{A})] \tag{105}$$

Constructing a diagonal matrix $\boldsymbol{M}$ with diagonal elements equal to $(\min_i \bar{s}_j^{[i]})^2$ the full loss is:

$$\mathcal{L} = \mathrm{Tr}[\boldsymbol{D}^2(\lambda_a \boldsymbol{M}^2 + \lambda_a \Sigma + \lambda_w (\boldsymbol{A}^T\boldsymbol{A})^{-1})] + \lambda_w \mathrm{Tr}[(\lambda_w \Sigma^{-1} + \boldsymbol{D}^2)^{-1}(\boldsymbol{A}^T\boldsymbol{A})] \tag{106}$$

We optimise numerically, though finding a simpler closed form solution does not seem impossible.

We find one further implicit definition of $\boldsymbol{D}$, which we'll use later, by taking the derivative with respect to (the diagonal elements of) $\boldsymbol{D}^2$. We find:

$$\frac{\partial \mathcal{L}}{\partial \boldsymbol{D}^2} = \underbrace{(\lambda_a \boldsymbol{M}^2 + \lambda_a \Sigma + \lambda_w (\boldsymbol{A}^T\boldsymbol{A})^{-1})}_{\boldsymbol{K}} - \lambda_w (\lambda_w \Sigma^{-1} + \boldsymbol{D}^2)^{-1}(\boldsymbol{A}^T\boldsymbol{A})(\lambda_w \Sigma^{-1} + \boldsymbol{D}^2)^{-1} \tag{107}$$

So the diagonal elements of this matrix equation equal zero at the optimal modular solution.

## C.2 Perturbing Optimal Modular Solution

In general our representations can be written using a linear map from the demeaned sources, against which we can then optimise. Calling this map $\boldsymbol{W}$:

$$\boldsymbol{z}^{[i]} = \boldsymbol{W}\bar{\boldsymbol{s}}^{[i]} - \min_i[\boldsymbol{W}\bar{\boldsymbol{s}}^{[i]}] \tag{108}$$

Then the loss divided by $\lambda_a$ is:

$$\mathrm{Tr}[\boldsymbol{W}^T\boldsymbol{W}\Sigma] + \sum_{j=1}^{d_s}(\min_i \boldsymbol{w}_n^T\bar{\boldsymbol{s}}_j^{[i]})^2 + \frac{\lambda_w}{\lambda_a}\mathrm{Tr}[\boldsymbol{W}^T\boldsymbol{W}(\boldsymbol{A}^T\boldsymbol{A})^{-1} + (\lambda_w\Sigma^{-1} + \boldsymbol{W}^T\boldsymbol{W})^{-1}\boldsymbol{A}^T\boldsymbol{A}] \tag{109}$$

Taking the derivative with respect to one of the rows of $\boldsymbol{W}$, $\boldsymbol{w}_n$, and dividing by two gives:

$$\boldsymbol{w}_n^T\Sigma + \boldsymbol{a}_n^T\min_i \boldsymbol{w}_n^T\bar{\boldsymbol{s}}^{[i]} + \frac{\lambda_w \boldsymbol{w}_n^T}{\lambda_a}((\boldsymbol{A}^T\boldsymbol{A})^{-1} - (\lambda_w\Sigma + \boldsymbol{W}^T\boldsymbol{W})^{-1}\boldsymbol{A}^T\boldsymbol{A}(\lambda_w\Sigma^{-1} + \boldsymbol{W}^T\boldsymbol{W})^{-1}) \tag{110}$$

Where, as in appendix B:

$$\frac{\partial \min_i[\boldsymbol{w}_n^T\boldsymbol{s}^{[i]}]}{\partial \boldsymbol{w}_{n'}} = \delta_{n,n'}\boldsymbol{a}_n \qquad \boldsymbol{a}_n \in \mathrm{Convex\ Hull}(\{\boldsymbol{s}^{[i]}|s_n^{[i]} = \min_j s_n^{[j]}\}) \tag{111}$$

Evaluating this at the optimal modular solution and setting it to zero we find that the condition that this is a local minima is:

$$\frac{1}{\min_i s_n^{[i]}}\underbrace{\left[\frac{\lambda_w}{\lambda_a}(\lambda_w\Sigma^{-1} + \boldsymbol{D}^2)^{-1}\boldsymbol{A}^T\boldsymbol{A}(\lambda_w\Sigma^{-1} + \boldsymbol{D}^2)^{-1} - \frac{\lambda_w}{\lambda_a}(\boldsymbol{A}^T\boldsymbol{A})^{-1} - \Sigma\right]}_{\boldsymbol{F}}\boldsymbol{e}_n \in \boldsymbol{a}_n \tag{112}$$

Now identical logic to the previous case follows through: if the sources are tightly scattered with respect to the matrix defined by $\boldsymbol{F}$, this is a minima, else it is not. We just need to show one final thing, that the diagonals of $\boldsymbol{F}$ are the square minima, i.e. $\mathrm{diag}(\boldsymbol{F}) = \boldsymbol{M}^2$. This can be seen as follows, rearranging the definition of $\boldsymbol{F}$ we get:

$$\lambda_a \boldsymbol{F} + \lambda_a \Sigma + \lambda_w (\boldsymbol{A}^T\boldsymbol{A})^{-1} = \lambda_w(\lambda_w\Sigma^{-1} + \boldsymbol{D}^2)^{-1}\boldsymbol{A}^T\boldsymbol{A}(\lambda_w\Sigma^{-1} + \boldsymbol{D}^2)^{-1} \tag{113}$$

pattern matching to eq. (107) we can see that the diagonals of this equation are satisfied if $\boldsymbol{F} = \boldsymbol{M}^2$.

Hence, tight scattering relative to this newly defined $\boldsymbol{F}$ again drives identifiability. We show some numerical verification of these results in fig. 4.

**Recovering Initial Results**  Finally, we can see that taking the limit that $\lambda_a$ and $\lambda_w$ are very small (but their ratio is not) let's us approximate $(\lambda_w\Sigma^{-1} + \boldsymbol{D}^2)^{-1}$ as simply $\boldsymbol{D}^{-2}$, in which case we recover the previous definition of $\boldsymbol{F}$, appendix B with $\lambda = \frac{\lambda_w}{\lambda_a}$.

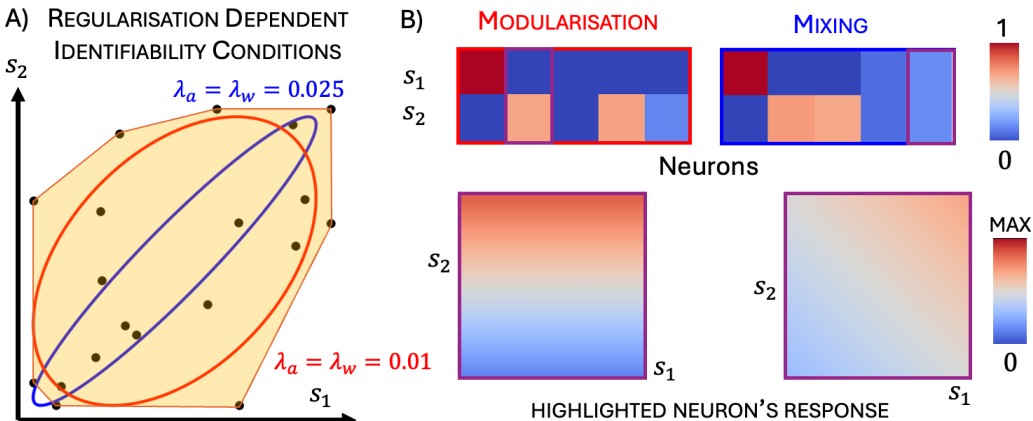

Figure 4: Similarly to fig. 1, in **(A)** We schematise identifiability conditions for two sources; which are either satisfied (red) or brocken (blue) by this dataset depending on the regularisation hyperaparameter. **(B)** Matching the theory, numerical solutions are modular for the less regularised network (left), but not for the more regularised (right). We plot the linear conditional mutual information (Hsu et al., 2023; Dorrell et al., 2025) between each neuron and source scaled by the neuron's peak activity. Below we display a highlighted (purple) neuron's tuning to sources.

# D  IDENTIFIABILITY OF NEURAL TUNING CURVES

We consider strictly convex optimisations over the set of representational dot-product similarity matrices with the constraint of nonnegative neural activities. In this setting there is a globally optimal similarity matrix, $\boldsymbol{Q}^*$. Let's imagine you measure a nonnegative representation that is optimal, $\boldsymbol{Z} \geq 0$, $\boldsymbol{Z}^T\boldsymbol{Z} = \boldsymbol{Q}^*$. What has to be true about the single neuron properties of $\boldsymbol{Z}$ such that all optimal representations contain the same neurons?

In general, any other optimal representation must be an orthogonal transform of your original representation, $\boldsymbol{Z}' = \boldsymbol{OZ}$. We consider constraints on $\boldsymbol{Z}$ such that, in order to preserve positivity, $\boldsymbol{O}$ must be a permutation matrix. In this case, the same single neuron responses appear in all representations, tieing them precisely to the optima.

We begin with a looser but simpler condition that has occured repeatedly in the literature. Then we show that we can significantly loosen this condition.

## D.1  SUFFICIENT SCATTERING

Following classic work on nonnegative matrix factorisation (Donoho & Stodden, 2003; Fu et al., 2018; Tatli & Erdogan, 2021a;b), define the cone:

$$\mathcal{C} = \{\boldsymbol{x}|\boldsymbol{x}^T\boldsymbol{1} \geq \sqrt{N-1}||\boldsymbol{x}||_2\} \tag{114}$$

Which has its corresponding dual cone[4]:

$$\mathcal{C}^* = \{\boldsymbol{x}|\boldsymbol{x}^T\boldsymbol{1} \geq ||\boldsymbol{x}||_2\} \tag{115}$$

Then our sufficient scattering condition is two conditions. We will use the notion of a cone of a matrix, all conic combinations of the columns of the matrix:

$$\text{cone}(\boldsymbol{X}) = \{\boldsymbol{x}|\boldsymbol{x} = \boldsymbol{Xa}, \forall \boldsymbol{a} \geq 0\} \tag{116}$$

1. First, a sufficient scattering condition that says that the datapoints, i.e. the rows of the matrix $\boldsymbol{Z}^*$, are spread around the positive orthant sufficiently:

$$\mathcal{C} \subseteq \text{cone}[\boldsymbol{Z}^T] \tag{117}$$

2. Second, there are few points at which the cone of the datapoints actually touches the edge of the cone $\mathcal{C}$. This can be formalised as:

$$\text{cone}\boldsymbol{Z}^{T^*} \cap \text{bd}\mathcal{C}^* = \{\lambda\boldsymbol{e}_k|\lambda \geq 0, k = 1, ..., N\} \tag{118}$$

Where $\text{bd}(\mathcal{C}^*)$ is the boundary of $\mathcal{C}^*$.

Our claim is that, if these two sufficient scattering conditions are satisfied, then the single neuron response properties are unique.

*Proof.* It is a classic result from convex optimisation stuff, for two cones $\mathcal{K}_1$ and $\mathcal{K}_2$, if $\mathcal{K}_1 \subseteq \mathcal{K}_2$, then their duals satisfy the opposite relation: $\mathcal{K}_1^* \supseteq \mathcal{K}_2^*$. Since $\mathcal{C} \subseteq \text{cone}[\boldsymbol{Z}^T]$ we therefore have that $\mathcal{C}^* \supseteq \text{cone}[\boldsymbol{Z}^T]^*$

Now, imagine you have some other neural representation: $\boldsymbol{Z} = \boldsymbol{OZ}^* \geq 0$ for some orthogonal matrix $\boldsymbol{O}$. Now, our first claim is about what has to be true about the rows of $\boldsymbol{O}$ such that $\boldsymbol{Z} \geq 0$. Call the $n$th row, $\boldsymbol{o}_n$, then, since $\boldsymbol{Z}^T\boldsymbol{o}_n \geq \boldsymbol{0}$:

$$\boldsymbol{o}_n \in \text{cone}(\boldsymbol{Z}^T)^* \subseteq \mathcal{C}^* \tag{119}$$

Now, membership in $\mathcal{C}^*$ means that $\boldsymbol{o}_n^T\boldsymbol{1} \geq ||\boldsymbol{o}_n||_2$. Further, we know that $||\boldsymbol{o}_n||_2 = 1$ because $\boldsymbol{O}$ is an orthogonal matrix, therefore $\boldsymbol{o}_n^T\boldsymbol{1} \geq 1$. In fact, we can go further and show this has to be an equality:

$$N = \boldsymbol{1}^T\boldsymbol{1} = \boldsymbol{1}^T\boldsymbol{O}^T\boldsymbol{O}\boldsymbol{1} = \sum_n(\boldsymbol{o}_n^T\boldsymbol{1})^2 \geq N \tag{120}$$

---

[4]A dual cone for cone $\mathcal{C}$ is defined as $\mathcal{C}^* = \{\boldsymbol{y}|\boldsymbol{y}^T\boldsymbol{x} \geq 0 \forall \boldsymbol{x} \in \mathcal{C}\}$

So, in order to satisfy the orthogonal matrix property, each $\boldsymbol{o}_n^T \boldsymbol{1} = 1$. This is great, because it means that $\boldsymbol{o}_n \in \mathrm{bd}(\mathcal{C}^*)$. But now the vectors are trapped, we can use the second part of the scattering condition. Since each row $\boldsymbol{o}_n$ is both in $\mathrm{bd}(\mathcal{C}^*)$ and $\mathrm{cone}(\boldsymbol{Z}^T)^*$, it is in its intersection, and this is very prescribed by assumption:

$$\boldsymbol{o}_n \in \mathrm{cone}\boldsymbol{Z}^{T^*} \cap \mathrm{bd}\mathcal{C}^* = \{\lambda \boldsymbol{e}_k | \lambda \geq 0, k = 1, ..., N\} \tag{121}$$

Therefore, each row of $\boldsymbol{O}$ is a scaled version of the a basis element. But in order to be an orthogonal matrix, in fact this must just be a permutation matrix. $\qquad\square$

### D.2 A FAMILY OF SUFFICIENT CONDITIONS

**Theorem 2** (Tight Scattering Implies Unique Neurons). *If your neural data, $\boldsymbol{z}^{[i]}$, satisfies the following two tight scattering constraints with respect to a set $E = \{\boldsymbol{x} + \langle \boldsymbol{z}^{[i]} \rangle_i | \boldsymbol{x}^T \boldsymbol{F}^{-1} \boldsymbol{x} = 1\}$ for a positive definite matrix $\boldsymbol{F}$ with diagonals equal to $(\langle \boldsymbol{z}^{[i]} \rangle_i)^2$ then all orthogonal matrices such that $\boldsymbol{O}\boldsymbol{z}^{[i]} \geq 0$ are permutation matrices.*

- *$Conv(\{\boldsymbol{z}_i^{[i]}\}) \supseteq E$*

- *$Conv(\{\boldsymbol{z}^{[i]}\}_i)^* \cap bd(E^*) = \{\lambda \boldsymbol{e}_k | \lambda \in \mathbb{R}, k = 1, ..., d_S\}$*

*Proof.* The key constraint is that the transformed data must be positive: $\boldsymbol{Z}' = \boldsymbol{O}\boldsymbol{Z} \geq 0$. Consider a row of $\boldsymbol{O}$, $\boldsymbol{o}_n$. This has to satisfy:

$$\boldsymbol{o}_n^T \boldsymbol{z}^{[i]} \geq 0 \quad \forall i = 1...T \tag{122}$$

A necessary condition for this to be true is the same statement but for members of the ellipse $E$:

$$\boldsymbol{o}_n^T \boldsymbol{x} \geq 0 \quad \forall \boldsymbol{x} \in E \tag{123}$$

Given a putative transform vector $\boldsymbol{o}_n$, the member of $E$ that will most tax its ability to preserve this defining inequality is the member of the ellipse $E$ that has the largest projection along the $-\boldsymbol{o}_n$ direction. Using lagrange optimisation we can find that this is the vector:

$$\frac{-\boldsymbol{F}\boldsymbol{o}_n}{\sqrt{\boldsymbol{o}_n^T \boldsymbol{F}\boldsymbol{o}_n}} + \langle \boldsymbol{z}^{[i]} \rangle_i \tag{124}$$

Inserting this into eq. (123):

$$\boldsymbol{o}_n^T \left( \frac{-\boldsymbol{F}\boldsymbol{o}_n}{\sqrt{\boldsymbol{o}_n^T \boldsymbol{F}\boldsymbol{o}_n}} + \langle \boldsymbol{z}^{[i]} \rangle_i \right) = -\sqrt{\boldsymbol{o}_n^T \boldsymbol{F}\boldsymbol{o}_n} + \boldsymbol{o}_n^T \langle \boldsymbol{z}^{[i]} \rangle_i \geq 0 \tag{125}$$

If this inequality is satisfied then all those in eq. (123) are as well. Let's rewrite this once more as:

$$(\boldsymbol{o}_n^T \langle \boldsymbol{z}^{[i]} \rangle_i)^2 \geq \boldsymbol{o}_n^T \boldsymbol{F}\boldsymbol{o}_n \tag{126}$$

And this must hold concurrently for all the different rows of $\boldsymbol{O}$. Since $\boldsymbol{O}$ is an orthogonal matrix we know a lot about these rows, i.e. $\boldsymbol{o}_n^T \boldsymbol{o}_m = \delta_{nm}$. Let's put this information to use by considering the sum of these inequalities over the population:

$$\sum_n (\boldsymbol{o}_n^T \langle \boldsymbol{z}^{[i]} \rangle_i)^2 = \langle \boldsymbol{z}^{[i]} \rangle_i^T \boldsymbol{O}^T \boldsymbol{O} \langle \boldsymbol{z}^{[i]} \rangle_i = ||\langle \boldsymbol{z}^{[i]} \rangle_i||_2^2 \geq \sum_n \boldsymbol{o}_n^T \boldsymbol{F}\boldsymbol{o}_n = \mathrm{Tr}[\boldsymbol{O}^T \boldsymbol{F}\boldsymbol{O}] = \mathrm{Tr}[\boldsymbol{F}] = ||\langle \boldsymbol{z}^{[i]} \rangle_i||_2^2 \tag{127}$$

Where the last inequality follows from the fact that the diagonal elements of $\boldsymbol{F}$ are $(\langle \boldsymbol{z}^{[i]} \rangle_i)^2$. Further, since the sums are equal, each of the elements must be equal. Hence, we see that if the first sufficient scattering condition is satisfied, then this inequality constraint must be an equality:

$$\boldsymbol{o}_n^T \underbrace{\left( \frac{-\boldsymbol{F}\boldsymbol{o}_n}{\sqrt{\boldsymbol{o}_n^T \boldsymbol{F}\boldsymbol{o}_n}} + \langle \boldsymbol{z}^{[i]} \rangle_i \right)}_{\in E} = 0 \tag{128}$$

Now we use the second scattering condition. The positivity constraint, $\boldsymbol{o}_n^T \boldsymbol{z}^{[i]} \geq 0 \forall i$, is the defining property of membership of the dual cone of the data: $\boldsymbol{o}_n \in \mathrm{Conv}(\{\boldsymbol{z}_i\}_i)^*$. Further, since $\boldsymbol{o}_n^T \boldsymbol{x} \geq$

$0 \forall \boldsymbol{x} \in E$, it is also in the ellipse's dual cone $\boldsymbol{o}_n \in \text{Conv}(E)^*$. Finally, not only is $\boldsymbol{o}_n$ in the dual cone of the convex hull of $E$, it is on the boundary, because, as shown in the equation above, $\boldsymbol{o}_n^T \boldsymbol{x} = 0$ for a point $\boldsymbol{x} \in E$. Therefore:

$$\boldsymbol{o}_n \in \text{Conv}(\{\boldsymbol{z}_i\}_i)^* \cap \text{bd}(E^*) = \{\lambda \boldsymbol{e}_k | \lambda \in \mathbb{R}, k = 1, ..., d_S\} \tag{129}$$

Where the last equality is the second scattering condition. Hence we have our final result. In order to have the same dot product structure and preserve positivity each $\boldsymbol{o}_n$ must align with a basis direction, and since it is unit norm, it must be a basis direction. So each new neuron $\boldsymbol{z}_n'^{[i]} = \boldsymbol{o}_n \boldsymbol{z}^{[i]} = z_k^{[i]}$: all single unit tuning properties are preserved. □

### D.3 PARTIAL IDENTIFIABLE POPULATIONS

In practice, it might be inconvenient to expect every neuron in a population to be identifiable - perhaps some set of neurons are together encoding one variable and can be easily rotated amongst themselves, but not mixed with another population. We study a setting in which such statements can also be proven. We consider an optimisation problem that is strictly convex over the de-meaned representational similarity matrices:

$$\bar{\boldsymbol{Q}} = \bar{\boldsymbol{Z}}^T \bar{\boldsymbol{Z}}, \quad \bar{\boldsymbol{Z}} = \boldsymbol{Z} - \boldsymbol{Z} \mathbf{1} \mathbf{1}^T \tag{130}$$

In this setting, all optimal representations have to be rotations of the demeaned sources plus some bias. Assuming positivity and activity regularisation, you can further prove that to be optimal the minimal bias of the original and rotated sources have to have equal length (see previous section). We now show that, if two sets of sources are range-independent, no representation that mixes them will be optimal.

Let's consider a matrix, $\boldsymbol{O}$, with orthogonal columns that transforms the de-meaned representation while preserving its dot product structure:

$$\tilde{\boldsymbol{z}}_i = \boldsymbol{O} \bar{\boldsymbol{z}}_i \qquad \tilde{\boldsymbol{z}}_j^T \tilde{\boldsymbol{z}}_i = \bar{\boldsymbol{z}}_j \boldsymbol{O}^T \boldsymbol{O} \bar{\boldsymbol{z}}_i = \bar{\boldsymbol{z}}_j \bar{\boldsymbol{z}}_i \tag{131}$$

Let's say that:

$$\bar{\boldsymbol{z}}_i = \begin{bmatrix} \boldsymbol{x}_i \\ \boldsymbol{y}_i \end{bmatrix} \tag{132}$$

For two range independent variables $\boldsymbol{x}_i$ and $\boldsymbol{y}_i$. Consider an orthogonal matrix that mixes the representation: $\boldsymbol{O}$. Then consider breaking the mixed parts into two separate populations of neurons:

$$\boldsymbol{O} = [\boldsymbol{O}_x \quad \boldsymbol{O}_y] \qquad \boldsymbol{O}' = \begin{bmatrix} \boldsymbol{O}_x & \mathbf{0} \\ \mathbf{0} & \boldsymbol{O}_y \end{bmatrix} \tag{133}$$

A representation made from $\boldsymbol{O}'$ still has the optimal dot-product similarity, because the columns of $\boldsymbol{O}'$ remain orthogonal. However, we will now show that the representation made from $\boldsymbol{O}'$ must have a shorter bias vector, and therefore the mixed representation cannot be an optimal one. Let's calculate the two bias vectors:

$$||\boldsymbol{b}||^2 = ||-\min_i [\boldsymbol{O}_x \boldsymbol{x}_i + \boldsymbol{O}_y \boldsymbol{y}_i]||^2 = ||\min_i [\boldsymbol{O}_x \boldsymbol{x}_i] + \min_i [\boldsymbol{O}_y \boldsymbol{y}_i]||^2 \geq ||\min_i [\boldsymbol{O}_x \boldsymbol{x}_i]||^2 + ||\min_i [\boldsymbol{O}_y \boldsymbol{y}_i]||^2 \tag{134}$$

And the final term is exactly the length of the bias required for the representation constructed from $\boldsymbol{O}'$. The equality only occurs if $\min_i [\boldsymbol{O}_x \boldsymbol{x}_i]^T \min_i [\boldsymbol{O}_y \boldsymbol{y}_i] = 0$. Since the variables are mean-zero, this only happens if $\boldsymbol{O}$ already kept the representation separate. Therefore, any time we think we find an optimal solution, it can be improved by keeping range-independent variables in different neurons.

# E A TRACTABLE NONLINEAR THEORY OF ON-OFF CODING

We consider a representation, $z^{[i]}$ that is nonnegative and from which a single stimulus, $x^{[i]}$, can be decoded using an affine readout:

$$\boldsymbol{R}z^{[i]} + \boldsymbol{r} = x^{[i]} \tag{135}$$

Subject to this encoding constraint and the nonnegativity of the representation we minimise an energy loss:

$$\mathcal{L} = \langle ||z^{[i]}||^2 \rangle_i + \lambda ||\boldsymbol{R}||_F^2 \tag{136}$$

In section 2 we showed that this problem is convex when reframed over the set of similarity matrices. We will use the KKT conditions to find necessary conditions on the optimal representations $\boldsymbol{Z}$ ($\boldsymbol{Z}$ denotes the stacked representation matrix, $[\boldsymbol{Z}]_{:,i} = z^{[i]}$ and similarly $[\boldsymbol{x}]_i = x^{[i]}$). We will find that the representation either comprises both an ON and an OFF channel, or just a single channel, and we will show that sparsity delimits these two scenarios. For these simple representations, it's easy to use the results in section 4 to show that ON-OFF coding cannot be rotated.

## E.1 KKT CONDITIONS LEAD TO ON-OFF CODING

The min-norm readout vector, $\boldsymbol{R}$, is the pseudoinverse that maps the de-meaned representation, $\bar{\boldsymbol{Z}}$, to the de-meaned data, $\bar{\boldsymbol{x}}$. Using this, we can write the loss as:

$$\mathcal{L} = \frac{1}{T}\text{Tr}[\boldsymbol{Z}^T\boldsymbol{Z}] + \lambda\,\text{Tr}[\bar{\boldsymbol{Z}}^\dagger\bar{\boldsymbol{x}}\bar{\boldsymbol{x}}^T(\bar{\boldsymbol{Z}}^\dagger)^T] = \frac{1}{T}\text{Tr}[\boldsymbol{Z}^T\boldsymbol{Z}] + \lambda\bar{\boldsymbol{x}}^T(\bar{\boldsymbol{Z}}^\dagger)^T\bar{\boldsymbol{Z}}^\dagger\bar{\boldsymbol{x}} \tag{137}$$

This has to be minimised subject to non-negativity, so we construct the following lagrangian:

$$\mathcal{L} = \frac{1}{T}\text{Tr}[\boldsymbol{Z}^T\boldsymbol{Z}] + \lambda\bar{\boldsymbol{x}}^T(\bar{\boldsymbol{Z}}^\dagger)^T\bar{\boldsymbol{Z}}^\dagger\bar{\boldsymbol{x}} - \text{Tr}[\boldsymbol{P}^T\boldsymbol{Z}] \tag{138}$$

Slightly non-rigorously, we take the derivative with respect to $\boldsymbol{Z}$ and, using the expression for the derivative of the pseudoinverse of constant rank (Golub & Pereyra, 1973), and we find that:

$$\frac{1}{T}\boldsymbol{Z} - \lambda\boldsymbol{Z}\bar{\boldsymbol{Z}}^\dagger(\bar{\boldsymbol{Z}}^\dagger)^T\bar{\boldsymbol{x}}\bar{\boldsymbol{x}}^T\bar{\boldsymbol{Z}}^\dagger(\bar{\boldsymbol{Z}}^\dagger)^T = \frac{1}{2}\boldsymbol{P} \tag{139}$$

And further, if $Z_{ij} > 0$ then $P_{ij} = 0$, or if $P_{ij} > 0$ then $Z_{ij} = 0$. If the former case:

$$Z_{ij} = T\lambda[\boldsymbol{Z}\bar{\boldsymbol{Z}}^\dagger(\bar{\boldsymbol{Z}}^\dagger)^T\bar{\boldsymbol{x}}\bar{\boldsymbol{x}}^T\bar{\boldsymbol{Z}}^\dagger(\bar{\boldsymbol{Z}}^\dagger)^T]_{ij} \tag{140}$$

Else it is zero. Since $P_{ij} \geq 0$ in these cases we know that $\boldsymbol{Z}\bar{\boldsymbol{Z}}^\dagger(\bar{\boldsymbol{Z}}^\dagger)^T\bar{\boldsymbol{x}}\bar{\boldsymbol{x}}^T\bar{\boldsymbol{Z}}^\dagger(\bar{\boldsymbol{Z}}^\dagger)^T \leq 0$. Hence, this can all be summarised by the following equation:

$$Z_{ij} = T\lambda[\boldsymbol{Z}\bar{\boldsymbol{Z}}^\dagger(\bar{\boldsymbol{Z}}^\dagger)^T\bar{\boldsymbol{x}}\bar{\boldsymbol{x}}^T\bar{\boldsymbol{Z}}^\dagger(\bar{\boldsymbol{Z}}^\dagger)^T]_+ \tag{141}$$

Where $[]_+$ denotes the elementwise operation $[x]_+ = \max(x, 0)$.

We can study the firing of a single neuron, $z_n \in \mathbb{R}^T$:

$$z_n = T\lambda[\bar{\boldsymbol{Z}}^\dagger(\bar{\boldsymbol{Z}}^\dagger)^T\bar{\boldsymbol{x}}\bar{\boldsymbol{x}}^T\bar{\boldsymbol{Z}}^\dagger(\bar{\boldsymbol{Z}}^\dagger)^T z_n]_+ \tag{142}$$

Noticing that the same vector appears twice: $\boldsymbol{a} = \bar{\boldsymbol{Z}} \dagger (\bar{\boldsymbol{Z}}^\dagger)^T\bar{z}$, we can rewrite much more simply:

$$z_n = T\lambda[\boldsymbol{a}\boldsymbol{a}^T z_n]_+ \tag{143}$$

We can break $\boldsymbol{a}$ into two parts, its positive and negative components, $\boldsymbol{a}_+ = [\boldsymbol{a}]_+$, $\boldsymbol{a}_- = [-\boldsymbol{a}]_+$, then this equation already tells us that in the optimal population there are only two types of neural responses, governed by the sign of $\boldsymbol{a}^T z_n$:

$$z_n = \begin{cases} T\lambda\boldsymbol{a}^T z_n\boldsymbol{a}_+ & \text{if} \quad \boldsymbol{a}^T z_n > 0 \\ T\lambda|\boldsymbol{a}^T z_n|\boldsymbol{a}_- & \text{if} \quad \boldsymbol{a}^T z_n < 0 \end{cases} \tag{144}$$

Now we derive these responses using the definition of $\boldsymbol{a}$: $\boldsymbol{a} = \bar{\boldsymbol{Z}}^\dagger(\bar{\boldsymbol{Z}}^\dagger)^T\bar{\boldsymbol{x}}$. Since $\bar{\boldsymbol{x}}$ is in the span of $\bar{\boldsymbol{Z}}$:

$$\bar{\boldsymbol{Z}}^T\bar{\boldsymbol{Z}}\boldsymbol{a} = \bar{\boldsymbol{x}} \tag{145}$$

Now, all neurons belong to these two firing patterns. Call the set of positive neuron indices $S_+$ then define the sum of all positive weightings $\alpha_+ = \sum_{n \in S_+} |\boldsymbol{a}^T \boldsymbol{z}_n|$, and the same for negative. Then:

$$\bar{\boldsymbol{Z}}^T \bar{\boldsymbol{Z}} \boldsymbol{a} = \alpha_+ \bar{\boldsymbol{z}}_+ \bar{\boldsymbol{z}}_+^T \boldsymbol{a} + \alpha_- \bar{\boldsymbol{z}}_- \bar{\boldsymbol{z}}_-^T \boldsymbol{a} \tag{146}$$

Hence:

$$T^2 \lambda^2 (\alpha_+ \bar{\boldsymbol{a}}_+ \bar{\boldsymbol{a}}_+^T \boldsymbol{a} + \alpha_- \bar{\boldsymbol{a}}_- \bar{\boldsymbol{a}}_-^T \boldsymbol{a}) = \bar{\boldsymbol{x}} \tag{147}$$

Where we've used the de-meaned variables $\bar{\boldsymbol{a}}_\pm$. Expanding these in terms of their means, $\mu_\pm$: $\bar{\boldsymbol{a}}_\pm = \boldsymbol{a}_\pm + \mu_\pm \mathbf{1}$ and writing the mean of $\boldsymbol{a}$ as $\mu_a$, we find that, with some rearrangement:

$$T^2 \lambda^2 \left( (\frac{1}{T\lambda} - T\mu_a)\boldsymbol{a}_+ - (\frac{1}{T\lambda} + T\mu_a)\boldsymbol{a}_- \right) = \bar{\boldsymbol{x}} - T^2 \lambda^2 (\mu_a T(\mu_+ + \mu_-) + \frac{\mu_+ - \mu_-}{T\lambda})\mathbf{1} = \bar{\boldsymbol{x}} + b\mathbf{1} \tag{148}$$

Since $\boldsymbol{a}_+$ and $\boldsymbol{a}_-$ are never concurrently non-zero, this tells us that they each code for a portion of $\bar{\boldsymbol{x}} + b\mathbf{1}$, $\boldsymbol{a}_+$ the positive, $\boldsymbol{a}_-$ the negative. We could wade further through this mess, but instead we can skip to a 2 neuron representation with exactly this form and optimise the remaining parameters directly:

$$\boldsymbol{z}^{[i]} = \begin{bmatrix} \alpha_+ [x^{[i]} - b]_+ \\ \alpha_- [-(x^{[i]} - b)]_+ \end{bmatrix} \tag{149}$$

Then we can calculate the loss:

$$\mathcal{L} = \langle ||\boldsymbol{z}^{[i]}||_2^2 \rangle_i + \lambda ||\boldsymbol{R}||_F^2 = \alpha_+^2 \langle [x^{[i]} - b]_+^2 \rangle_i + \alpha_-^2 \langle [-(x^{[i]} - b)]_+^2 \rangle_i + \lambda(\frac{1}{\alpha_+^2} + \frac{1}{\alpha_-^2}) \tag{150}$$

We will assume each $b$ lies within the range of $x$ (i.e. we are using an ON-OFF code) and take the derivative with respect to each of the $\alpha$ to get:

$$\alpha_+^4 = \frac{\lambda}{\langle [x^{[i]} - b]_+^2 \rangle_i} \qquad \alpha_-^4 = \frac{\lambda}{\langle [-(x^{[i]} - b)]_+^2 \rangle_i} \tag{151}$$

And with respect to the bias we end up with the implicit equation:

$$\frac{\int_b^\infty (x - b)dp(x)}{\int_b^\infty (x - b)^2 dp(x)} = \frac{\int_{-\infty}^b (b - x)dp(x)}{\int_{-\infty}^b (b - x)^2 dp(x)} \tag{152}$$

So, in general, solve this implicit equation for $b$ (even numerically), then calculate $\alpha_+$ and $\alpha_-$ and you have your representation. A simpler solution is that if $p(x)$ is symmetric you can see that $b = 0$ is a solution.

### E.2 Direct vs. ON-OFF Coding

The previous result assumed that we had an ON and an OFF neuron each with non-zero firing. The other alternative is a single channel, for which we have two choices, ON or OFF:

$$\text{either: } z^{[i]} = x^{[i]} - \min_i x^{[i]} \text{ or: } z^{[i]} = -x^{[i]} - \max_i x^{[i]} \tag{153}$$

We only have to consider the optimal one, which will be the lower energy option. This will be the ON channel if:

$$\langle (x^{[i]} - \min_i x^{[i]})^2 \rangle_i \leq \langle (-x^{[i]} - \max_i x^{[i]})^2 \rangle_i \tag{154}$$

and the OFF channel otherwise.

For simplicity let's assume the ON channel is the preferred single neuron response (all the arguments can be switched to the OFF if needed). Let's calculate what has to be true for the one neuron solution to be stable to perturbations into two neuron coding.

To do that we compare the one neuron loss for the optimal value of $\alpha_+ = \sqrt{\frac{\lambda}{\langle (x^{[i]} - \min_i x^{[i]})^2 \rangle_i}}$:

$$\mathcal{L}_+ = 2\sqrt{\lambda \langle (x^{[i]} - \min_i x^{[i]})^2 \rangle_i} \tag{155}$$

To one where we shift the bias the tiny-iest amount positive to create an ON and OFF neuron. Since the dataset is finite, we can choose this tiny bias to sit between the minimal $x$ value and the

next smallest. Then the OFF neuron will only activate for the minimal $x$. Let's denote with $S$ the probability of the minimal $x$, then the loss:

$$\mathcal{L} = \alpha_+^2 \langle [x^{[i]} - \min_i x^{[i]} - b]_+^2 \rangle_i + \alpha_-^2 b^2 S + \lambda(\frac{1}{\alpha_+^2} + \frac{1}{\alpha_-^2}) \tag{156}$$

We can study the second moment:

$$\langle [x^{[i]} - \min_i x^{[i]} - b]_+^2 \rangle_i = \langle (x^{[i]} - \min_i x^{[i]} - b)^2 \rangle_i - b^2 S = \langle (x^{[i]} - \min_i x^{[i]})^2 \rangle_i - 2b\langle x^{[i]} - \min_i x^{[i]} \rangle_i + b^2(1 - S^2) \tag{157}$$

And then, using this expression, calculate the optimal $\alpha_+$ and $\alpha_-$ and find, to first order in $b$:

$$\mathcal{L} = 2\sqrt{\lambda b^2 S} + 2\sqrt{\lambda \langle (x^{[i]} - \min_i x^{[i]})^2 \rangle_i - 2b\langle x^{[i]} - \min_i x^{[i]} \rangle_i} \tag{158}$$

Comparing this to the previous loss and series expanding in $b$:

$$\frac{\mathcal{L} - \mathcal{L}_+}{2\sqrt{\lambda}} = b\sqrt{S} - b\frac{\langle x^{[i]} - \min_i x^{[i]} \rangle_i}{\sqrt{\langle (x^{[i]} - \min_i x^{[i]})^2 \rangle_i}} \tag{159}$$

Hence, dual ON-OFF channel coding is better if:

$$S < \frac{\langle x^{[i]} - \min_i x^{[i]} \rangle_i^2}{\langle (x^{[i]} - \min_i x^{[i]})^2 \rangle_i} \tag{160}$$

else, when the representation is sparse enough, single channel coding is better.

### E.3 NUMERICAL DETAILS

To produce fig. 3 we numerically optimised eq. (136) subject to decoding and nonnegativity constraints for a one-dimensional readout, and plotted the resulting representation. fig. 3C shows a measure of linear coding. Specifically, for each neuron we calculate the set of datapoints on which it has significantly non-zero activity. We then quantify what proportion of total firing those datapoints encapsulate by taking the norm of the representation matrix restricted to those points divided by the norm of the total representation matrix, $\mathbf{Z}$. We then take the max of this quantity across neurons.