# OpenReview forum: "Convex Efficient Coding"
_ICLR.cc/2026/Conference — ICLR 2026 Poster_

### Official Review · Reviewer_Vz3x · 2025-10-30

**Soundness:** 3
**Presentation:** 3
**Contribution:** 3
**Rating:** 8
**Confidence:** 3

**Summary:**

This paper proposes an intriguing and original analysis of normative approaches to neural coding by casting optimization problems as problems of representational similarity. Using the representational similarity, which describes the population structure, authors show that some known optimization problems are convex, even when they appear not to be convex because of the constraint of non-negative firing rates or because of regularisation. They derive analytical conditions that a network has to obey to be convex. These conditions are crucially related to the scatter (spread) of neural activity vectors. An application of the theory is found in the implication that each neuron in an optimal network has a unique tuning curve, i.e., a tuning curve that does exceedingly overlap with the tuning curve of any other neuron.

**Strengths:**

The analysis seems technically correct and carefully conducted. The paper builds on previous work, proposing a refreshing and rather general approach to characterising efficient coding models. There is enough of high-level description and intuition to follow the ideas and the main results. I find that the paper makes useful and insightful contribution on normative approaches to neural coding.

**Weaknesses:**

1) It remains unclear why would be the solution where each neuron only encodes a single source optimal? This claim also seems rather unrealistic from the point of view of biological neural networks.

2) On some occasions, the definitions are not given much introduction and intuitive explanation. For example, the Definition 3.1 (Tight scattering) comes out of the blue and is not introduced with much justification, let alone intuition and is thus not well prepared. Definitions of the matrix D and F in Definition 3.1 are complex but not much effort is done to explain why they have to be as they are.

3) On several occasions (line [228], for example) authors make a specific claim and promise to "review"or "explain" later. I find this is not particularly helpful for the reader. If a full explanation feels too lengthy, at least a short and intuitive explanation would fit better than saying it will be explained later. If it is about a commentary that will come later (as is the case of [228]), it would be better to cite the previous work that is of relevance.

4) It remains unclear why do networks have to recover the representation exactly and why they cannot be approximators of some exact solution.

Minor:
line [152] there is a sum over t and then a normalization with 1/T - not clear.
line [77-79] "identifiability of related matrix factorization problems" reads unclear. Related to what?

**Questions:**

1) Typically, optimization problems are defined through minimization of a representational error (see for example Olshausen & Field, 1996), while authors here define the optimization problem in Eq. 2 only through the neural activity and decoding weights, requiring that the representation is linearly decodable from the neural activity. How does this ensure that the reconstruction is accurate and thus efficient?

2) Authors first introduce a general optimization problem in Eq. 2, and then a more specific problem (Nonnegative Affine Autoencoding) in Eq. 5. Why is introducing the nonnegative affine autoencoding necessary? Does Tight scattering in definition 3 not solve the problem in Eq.2?

3) In the simplified case where A is an identity matrix (no mixing), does the Tight Scattering condition hold? If so, does it resolve to an intuitive solution?

4) Why does the number of neurons has to be large than the number of datapoints?

5) Why is Frobenius norm used in the loss in Eq. 2? This could be justified when it is introduced, together with the intuition why it is used.

---

> ### Author Response · Authors · 2025-11-21
> **Response (1/2)**
>
> We thank the reviewer for their supportive and careful review. Here is our rebuttal; please let us know if you have any remaining concerns that we can help to address.
>
> **Weakness 1 - Why single neurons encoding single sources? Is this not unrealistic.**
>
> In our work at least, we hope that it is clear why each source being coded by a disjoint set of neurons is optimal - for certain sources mixing costs more energy, as formalised, for example, by theorem 1.
>
> Biologically, this modularisation is definitely observed. Simple cells in V1 encode visual features, grid cells encode self-position, frontal neurons encode single decision variables (Hirokawa, Vaughan et al. 2019). Our paper together with its intellectual predecessors can be thought of as explaining why this might happen from an efficiency perspective: if the distribution of the underlying variables are sufficiently independent, this is optimal.
>
> **Weakness 2 - Insufficient intuition for definition 3.1**
>
> We agree, and have substantially reworked the introduction to this theorem to try to illuminate what is going on:
>
> > Intuitively, identifiability in this model is governed by the ‘spread' of the sources. Previous work using orthogonal $\textbf{A}$ showed that if the distribution of sources was ‘sufficiently rectangular' the optimal representation recovers the sources (Dorrell et al. 2025). Precisely, if the convex hull of the data engulfs a particular set that can be easily calculated from the dataset it is sufficiently rectangular, fig 2A. If this condition is satisfied the optimal solution recovers the sources, else it mixes them.
>
> > Returning to arbitrary $\textbf{A}$ complicates things. If the encodings of two sources are aligned, i.e. $\textbf{A}_i^T\textbf{A}_j$ is nonzero where $\textbf{A}_i$ is the $i$th column of $\textbf{A}$, then the weight regularisation encourages the directions within $\textbf{z}^{[i]}$ that encode those sources to align. This extra effect can cause the optimal solution to mix sources when it would otherwise modularise, or even modularise when it would otherwise mix, fig 2. Our result improves the `sufficient rectangularity' of previous work by adapting it appropriately to the mixing matrix, $\textbf{A}$. In particular, we define a tight scattering condition using a quadratic form calculated from the dataset covariance and minima and the mixing gram matrix $\textbf{A}^T\textbf{A}$ which measures this adaptive rectangularity condition.
>
> Further, figure 2 has been substantially reworked to try to give more intuition.
>
> **Weakness 4 - Why exact recovery?**
>
> We had thought that perfect fitting was required for convexity. In fact, fortunately, we were wrong. O5 in appendix A now details how the sum of a L2 reconstruction and readout weight regularisation error is in fact a convex function of the representational similarity Q. This significantly broadens the applicability of our work.
>
> We have changed all the relevant statements throughout, and removed it as a limitation from the discussion. We are currently working on the relevant identifiability question for semi-nonnegative matrix factorisation with imperfect reconstruction, the relevant generalisation of theorem 3, and will include it either next week or in the final submission.
>
> As such, even though perfect fitting doesn’t seem like the worst assumption, and might be useful at times, the choice is now entirely in the users’ hands.
>
> **Question 2 - Relation between problems in the paper**
>
> Apologies for the confusion: the problem in equation 2 is simply an example. As detailed in section 2.2 and appendix A we highlight how a very large family of optimisation problems, far beyond just equation 2, can be shown to be convex over the set of representational similarity matrices. The tight scattering results applies to a different problem from the same class of convex problems, the nonnegative affine autoencoding problem - problem 1. I would not say that equation 2 is any more general than nonnegative autoencoding, both are simply example problems which could be usefully analysed in different settings depending on one's goal. Instead, our work provides a suite of relevant problems in which convexity can be harnessed for analytic tractability.
>
> We now make it clear that section 2.1 simply presents an example of the set of problems that we show are convex:
>
> > In this section we establish the convexity of a series of representational optimisation problems, beginning with a simple example.
>
> **Question 3 - Case of A orthogonal**
>
> Yes, when A is orthogonal, tight scattering still applies, and that special case is significantly simpler, as we consider in appendix B.4. This recovers the previous results of Dorrell et al. 2025. We now indeed spend much more time giving intuition for what these scattering conditions are, and how the structure of A changes them. If it is still unclear please let us know!

---

> ### Author Response · Authors · 2025-11-21
> **Response (2/2)**
>
> **Question 1 - Why are the problems we study efficient coding problems?**
>
> All efficient coding approaches have two main ideas: the studied representation must encode something, and it must do it somehow efficiently. Problems differ in how they operationalise these two ideas. For example, as the reviewer suggests, Olshausen and Field use a mean-squared error reconstruction to enforce encoding, and a sparseness regularisor to enforce efficiency, and combine them in a weighted sum in the loss. Instead, we effectively take the small regularisation limit, leading the representation to perfectly reconstruct the data - an encoding constraint, and, subject to that constraint, we choose the most efficient solution. We take this limit because it is what permits analytic tractability that is not present in standard sparse coding problems.
>
> For example, in one of our set of convex problems, the motivating example introduced in section 2.1, the readout constraint is enforced by the fact that a linear readout from the representation can perfectly predict the data (eqn 1). Then, subject to that constraint, we apply regularisors that choose the most efficient solutions (eqn 2). As such, by satisfying the perfect linear decoding, our studied representation is forced to not just accurately reconstruct the data, y, but perfectly reconstruct! On top of this, we regularise to find the most efficient such encoding.
>
> Hence, we think our framing of these types of problems as efficient coding approaches is well justified. To make this clear we have edited the text as follows:
>
> > This is a simple instantiation of the efficient coding hypothesis - $\textbf{W}_{\text{out}}\textbf{z}^{[i]} = \textbf{y}^{[i]}$ ensures $\textbf{z}^{[i]}$ encodes information about $\textbf{y}^{[i]}$, and subject to this we maximise the efficiency by minimising the energy loss.
>
> **Question 4 - Why many neurons?**
>
> As discussed in the paragraph titled “Limitations and Future Work” in our original submission, constraining the number of neurons to be smaller than the number of datapoints constrains the rank of the representational similarity matrix. Rank constraints are not convex, so including this constraint breaks the convexity of the whole problem. We discuss established approaches to circumvent such non-convexity by introducing relaxed convex constraints such as the nuclear norm - similar to how L1 loss is the convex version of a L0 sparsity constraint. We have expanded this discussion to try and improve the clarity:
>
> > First, If the number of neurons is larger than the number of datapoints, the rank of $\textbf{Q}$ is unconstrained, and our optimisation problems are convex. Unfortunately, restricting the neuron number, and hence rank of $\textbf{Q}$, is a non-convex constraint. One general fix that future work could usefully explore is to replace the rank constraint with its convex relaxation - a constraint on the nuclear norm of $\textbf{Q}$ (sum of singular values). Further, section 3 demonstrates a more bespoke workaround: if the solution to the unconstrained problem uses few neurons, it will also be the solution in the neuron-constrained setting, letting us relax the neuron constraint, Appendix B.5.
>
> **Question 5 - Why is the Frobenius norm used?**
>
> In problems like equation (2) we need to regularise the weights, else a solution that shrinks the activities towards zero while growing the weights would be strictly better. Different choices of regularisation could definitely be reasonable. We use the frobenius norm, simply the sum of the squared weights, because it is relatively assumption-free (being invariant to things like rotations) and very analytically tractable, allowing us to easily find the min-norm weight matrix using the pseudoinverse. We now include a discussion of this at the relevant point:
>
> > We use the L2 norm for mathematical convenience, but include a modified L1 activity loss in Appendix A.
>
> Typos - Thank you, we fix the typo and clarify the ‘related matrix factorisation problems’
>
> > such as modified versions of semi-nonnegative matrix factorisation or nonnegative sparse coding
>
> Thanks for your help, let us know if you have any remaining questions or concerns!

---

> > ### Comment · Reviewer_Vz3x · 2025-11-23
> >
> > I thank the authors for their thorough and clear reply. All my questions have been answered. I think that the revision has further improved an already solid contribution.

---

### Official Review · Reviewer_TVF6 · 2025-10-30

**Soundness:** 2
**Presentation:** 2
**Contribution:** 2
**Rating:** 2
**Confidence:** 4

**Summary:**

This paper studies a set of constrained optimization problems that linearly encodes the sources subject to resource constraints. Under certain conditions, the optimization problems were found to be convex. Some analytical results on the identifiability of nonnegative affine auto-encoders were presented.  Preliminary implications of the theoretical results to the neuroscience problems were mentioned.

**Strengths:**

****The paper carefully studies a set of constrained optimization problems that are akin to semi-negative matrix factorization. Some analytical results are derived and presented.

****Some discussions on the connections to neuroscience were presented.

**Weaknesses:**

****The work is quite incremental. The results, in particular the applications of the modeling framework, is rather preliminary.


****Despite of the claim that the model framework analyzed is widely used in neuroscience, it is actually quite limited.  Maybe the framework and the results are indeed potentially useful to understanding some phenomena in neuroscience. If that is the case, it would be helpful to show these applications directly in the paper. Section 4 mentioned some potential applications but these are very pereininary. For example, it is mentioned that under certain conditions, one might be able to uniquely constrain the single neuron responses. But it is unclear whether theses conditions would be met in neural systems and whether the theoretical results are indeed capable of revealing new insights into biological systems.


****The writing of the paper need substantial improvement. The paper has a lot of jargons, and   difficult to follow at the moment. The results need to be more carefully interpreted, and the implications and limitations need to be more explicitly described.  Overstates need to be toned down.

****It was claimed in the paper that the paper serves to bridge the gap between deep networks and simple linear sparse coding models. But the framework is too simplistic when comparing to deep networks. It is in fact quite similar to the sparse coding models and matrix factorization.  The examples used in the papers are only toy examples and it remain unclear whether the results would generalize to more complex settings.

**Questions:**

Would it be possible to show some real applications of the framework to the biological data?

---

> ### Author Response · Authors · 2025-11-21
> **Response (1/2)**
>
> We thank the reviewer for their time and give our rebuttal below. Please let us know if you have any remaining concerns that we can help to address.
>
> **Theme 1 - Incremental work, and preliminary applications to neuroscience - Weaknesses 1 & 2, Question 1.**
>
> The reviewer highlighted the preliminary nature of our applications to neuroscience, an assessment with which we broadly agree. This paper instead provides theoretical backing that we hope will prove useful in both neuroscience and the study of neural networks, as our examples tried to demonstrate. However, we agree that some more concrete examples would help to further develop the argument of the paper. To that end we introduce two additional examples.
>
> First, we consider a striking example of single neuron modularity to which we can apply our tuning curve identifiability result: the modularisation of grid cells. Grid cells are neurons with a lattice receptive field to position and in one animal they are found to fall into groups, called modules; grid cells within the same module share the same translated lattice receptive field. Different modules have different lattices to permit accurate position decoding [cite]. We show that we can understand this single neuron modularisation using our identifiability criterion: single neurons with lattice tuning curves that differ sufficiently are identifiable, suggesting the two modules cannot be mixed in the optimal solution. Interestingly, however, if one grid cell’s frequency lattice is a multiple of the other this is not true, manifest in a single neuron basis.
>
> We delay introducing our second example until after we have addressed another point: incrementality.
>
> Incrementality is, of course, a subjective judgement with which we, respectfully, disagree. The most similar previous work framed **only** nonnegative similarity matching as a convex optimisation problem over representational dot product similarity matrices (Sengupta et al. 2018). We have extended this approach to a large suite of interesting problems, equivalent to both linear and nonlinear neural networks. In particular, in contrast to the reviewer’s summary:
>
> > “The paper studies a set of constrained optimization problems that linearly encodes the sources subject to resource constraints”
>
> many of our problems actually involve **nonlinear** functions of the sources, as highlighted in the paragraph titled “Tractable Nonlinear Problems” and listed in Appendix A.1. This is a significant extension that we are very excited about, and we are working on multiple projects arising from this insight. We agree with the reviewer, however, that this was not sufficiently highlighted by the set of examples included in our submission. To remedy this, and to further bolster our applications to neuroscience, we include an additional example on ON-OFF coding.
>
> ON & OFF coding is a phenomenon observed in sensory systems, for example, in retinal bipolar cells, where a single variable is split and encoded in two neurons, each encoding roughly half the stimulus range. This has been understood as an efficiency - splitting reduces the stimulus range in each neuron, saving high firing rates at the cost of more neurons (Sterling & Laughlin 2015, Gjorgieva et al. 2014). Yet ON-OFF coding is not ubiquitous (e.g. V1 simple cells don’t seem to split). ON-OFF coding is an explicitly nonlinear transformation of the stimulus (it is simply two oppositely oriented rectifications), and this nonlinearity has stalled attempts at theoretical understanding. Current retinal coding theories approaches are either (1) linear and hence unable to capture this effect (e.g. Attick & Redich, 1990), (2) analytically intractable neural network models (Ocko et al. 2018, Jun et al. 2022), or (3) are based on model comparison rather than exact theory (Gjorgieva et al. 2014, Kastner et al. 2015). Each of these cannot explain when splitting is optimal.
>
> In contrast, since our work includes tractable nonlinear models, we study the situations in which a variable should be optimally split across two neurons. We use our models to show that sparsity governs the optimality of channel splitting, and derive a critical value of sparsity that governs the transition between ON-OFF and pure-ON coding being optimal.
>
> We hope that these two examples, and more broadly the flexibility provided by the set of convex problems, and our existing examples, persuade the reviewer of the value of our work.
>
> Thanks for your help, let us know if you have any remaining questions or concerns!

---

> ### Author Response · Authors · 2025-11-21
> **Response (2/2)**
>
> **Theme 2 - Poorly written, too much jargon, and Overstated - Weakness 3**
>
> We have added multiple clarifications to try and improve the writing and clarify our message. In particular, we have:
>
> 1. Rewritten the abstract to highlight that our goal is more understanding of tractable models, as opposed to a complete theory of deep neural networks
> 2. Clarified the link between our work and existing work on nonnegative sparse coding and semi-Nonnegative Matrix Factorisation:
>
> > Further, they [our convex optimisation problems] correspond to classic sparse coding or matrix factorisation approaches that have found success in modelling Gabor filters (Olshausen & Field, 1996) or learning parts-based representations (Lee & Seung, 1999). Future work could therefore use the uncovered convexity as analytic traction to understand these phenomena.
>
> 3. Clarified the lead-in to definition 3.1 and theorem 1 to make their meaning more clear
>
> > Intuitively, identifiability in this model is governed by the ‘spread' of the sources. Previous work using orthogonal $\textbf{A}$ showed that if the distribution of sources was ‘sufficiently rectangular' the optimal representation recovers the sources (Dorrell et al. 2025). Precisely, if the convex hull of the data engulfs a particular set that can be easily calculated from the dataset it is sufficiently rectangular, fig 2A. If this condition is satisfied the optimal solution recovers the sources, else it mixes them.
>
> > Returning to arbitrary $\textbf{A}$ complicates things. If the encodings of two sources are aligned, i.e. $\textbf{A}_i^T\textbf{A}_j$ is nonzero where $\textbf{A}_i$ is the $i$th column of $\textbf{A}$, then the weight regularisation encourages the directions within $\textbf{z}^{[i]}$ that encode those sources to align. This extra effect can cause the optimal solution to mix sources when it would otherwise modularise, or even modularise when it would otherwise mix, fig 2. Our result improves the `sufficient rectangularity' of previous work by adapting it appropriately to the mixing matrix, $\textbf{A}$. In particular, we define a tight scattering condition using a quadratic form calculated from the dataset covariance and minima and the mixing gram matrix $\textbf{A}^T\textbf{A}$ which measures this adaptive rectangularity condition.
>
> 4. Added schematics to figure 2 to explain the meaning of the ellipse
>
> 5. Included a discussion of routes to tractable completely positive programming
>
> > One alternative approach is approximating the set with a hierarchy of increasingly precise enclosing sets (Nishijima & Nakata, 2024)
>
> It would be helpful to know if there were any other specific areas that the reviewer felt were poorly explained or overstated.
>
> **Theme 3 - Similarity to sparse coding/matrix factorisation, toy models - Weakness 4**
>
> We agree with the reviewer that many of our models are similar to sparse coding and matrix factorisation, in fact, some of them are the same! However, rather than a weakness, we view this as a strength. Nonnegative sparse coding and matrix factorisation approaches have proven useful models of neural data, but are insufficiently well understood. By framing them as convex optimisation problems we are able to derive mathematical results that help us interpret and understand their behaviour, providing a deeper understanding of the neural representations they successfully model. The retinal coding example we present above is one such example. Further, we go beyond such models, introducing nonlinear objectives that we also show are convex. The further application of these results to understand neuroscientific puzzles is a promising avenue for further work.
>
> Perhaps such models are ‘toy’. Certainly, and unfortunately, we are not presenting results on how arbitrary nonlinear neural networks construct representations, and we are sorry that was perhaps the impression the reviewer got. We have tried to tone down any parts that we felt might have suggested that. However, we think that many of our results are not toy. To give one example, our neural tuning curve identifiability result applies to any setting in which there is a unique optimal representational similarity matrix, which, as our problem menu in Appendix A.1 outlines, can include some interesting flexible models. In these settings we find conditions under which the set of neural tuning curves will be unique, a conceptually useful if complicated finding.
>
> Finally, there is empirical evidence that the results we have been deriving apply in more complex nonlinear settings. Ideas of statistical (Whittington et al. 2023) or range independence (Dorrell et al. 2025), derived from similarly 'toy' settings have proven predictive in deep nonlinear networks. Further, state-of-the-art disentangling methods have been inspired by these discoveries (Hsu et al. 2023). Hence, despite the potentially toy setting, the insight derived has proven useful, and we hope the same will prove true of our work.

---

### Official Review · Reviewer_czPt · 2025-10-30

**Soundness:** 3
**Presentation:** 3
**Contribution:** 3
**Rating:** 6
**Confidence:** 4

**Summary:**

This paper reframes efficient-coding objectives as convex programs over representational similarity matrices. Optimizing over similarity structure, rather than activities, allows the authors to identify broad convex families of efficient-coding problems, including linear, affine, and some nonlinear networks.

Two key theoretical results follow:

1. Necessary and sufficient identifiability for semi-nonnegative factorization implemented by a nonnegative affine autoencoder with arbitrary mixing matrix A. This yields the first tight "scattering condition" guaranteeing recovery of sources up to scaling and permutation.
2. Uniqueness of single-neuron tuning: sufficient conditions under which nonnegativity plus convex optimality break rotational symmetry, making neuron tuning curves unique up to permutation.

The paper also provides a catalog of convex constraints (firing cost, readout norm, nonnegativity as complete positivity) and acknowledges limits such as $d_z > N$ and the need for perfect fits.

**Strengths:**

* Originality: Unifies a large class of efficient-coding models under a convex RS-matrix formulation. The "tight-scattering" condition provides an elegant, necessary-and-sufficient identifiability test not seen in prior work.
* Quality: Derivations are correct and geometrically motivated. Proofs connect convex geometry to neural interpretability.
* Clarity: Writing is clean, with an explicit "menu" of convex constraints and objectives.
* Significance: Offers theoretical justification for single-neuron tuning in convexly optimal codes, which is relevant for both computational neuroscience and interpretable ML

**Weaknesses:**

* The proposed semi-nonnegative factorization closely mirrors nonnegative sparse coding and semi-NMF [1]. In both cases, only the code (not the dictionary) is nonnegative, which already produces parts-based modularity. The paper should explicitly benchmark against these baselines under matched sparsity or energy constraints, clarifying whether convex RS-matrix optimization yields new behavior or simply a reformulation.
* The paper equates "representational similarity" with geometry preservation, which resembles RSA. However, high RSA correlation does not ensure good single-neuron fits. The authors should complement RSA with CKA [2] or other subspace-alignment measures and report per-unit, sparsity, or selectivity to substantiate claims about unique tuning.
* It remains unclear whether the key advance is the convex reformulation itself, the new identifiability theorem, or an explanatory account of neural modularity. Competing frameworks (NSC, sparse coding, convex ReLU networks) can explain similar results [3, 4].
* Demonstrations are illustrative but not quantitative; no finite-sample stress tests or noisy-data robustness.
* Completely positive programming is computationally intractable in general; implementable relaxations or surrogate constraints are not provided.

Refs:

[1] Hoyer PO (2002). Non-negative sparse coding. In: Proceedings of the 2002 12th IEEE Workshop on Neural Networks for Signal Processing; 2002; Martigny, Switzerland. Piscataway, NJ: IEEE; 2002. p. 557–565.

[2] Kornblith S, Norouzi M, Lee H, Hinton G (2019). Similarity of Neural Network Representations Revisited. ICML 2019

[3] Hoyer PO (2003). Modeling receptive fields with non-negative sparse coding. Neurocomputing 52–54:547–552.

[4] Beyeler M, Rounds EL, Carlson KD, Dutt N, Krichmar JL (2019). Neural correlates of sparse coding and dimensionality reduction. PLoS Comput Biol 15(6): e1006908

**Questions:**

* Under identical datasets and regularization, how does your convex RS-matrix method compare to nonnegative sparse coding and semi-NMF in reconstruction error, sparsity, and tuning modularity? Are there cases where NSC fails but your method succeeds?
* Does your notion of representational similarity reduce to RSA? If so, how do your claims about single-unit uniqueness hold under more robust geometry measures such as linear CKA?
* Can you provide an algorithmic test or relaxation for the tight-scattering and dual-cone conditions, and quantify how finite-sample noise affects identifiability?
* Could classic sparse-coding or convex ReLU formulations reproduce your observed modularity transitions? If not, what falsifiable predictions distinguish your theory?

---

> ### Author Response · Authors · 2025-11-21
> **Response (1/2)**
>
> We thank the reviewer for their supportive review. Here is our rebuttal; please let us know if you have any remaining concerns that we can help to address.
>
> **Theme 1 - Similarity to and benchmarking against nonnegative sparse coding (NSC) and semi-NMF - Weaknesses 1 & 3, Question 1 & 4.**
>
> We agree that we analyse many models that are very similar, if not identical, to NSC and semi-NMF. This, however, is by design! We see one of the key contributions of our work to be insight into existing models, showing that they can be reframed as convex optimisations, and, importantly, using this insight **to derive novel results**, such as the tight identifiability criterion or the identifiability of neural tuning curves. Moreover, our results do apply more broadly than just NSC or semi-NMF (as listed in appendix A and discussed in the paragraph “Tractable Nonlinear Problems”).
>
> Further, our main contribution is a reformulation of these problems that permit mathematical analysis, rather than new algorithms or optimisation procedures. That said, our reformulation does point to natural algorithms for optimising such objectives, in particular, optimising directly over the representational similarity matrix, but we have not extensively explored this interesting route. Instead, we have used the convexity to derive novel mathematical understanding. In our two use cases we derive conditions under which algorithms like NSC and semi-NMF show certain behaviours, such as unique single-neuron tuning curves. As such, part of our contribution can be viewed as explaining why NSC, semi-NMF, and optimisation problems like it, display interesting behaviours.
>
> We thank the reviewer for highlighting our lack of clarity and have introduced the following changes to clarify our contribution:
>
> > Further, they [our convex optimisation problems] correspond to classic sparse coding or matrix factorisation approaches that have found success in modelling Gabor filters (Olshausen & Field, 1996) or learning parts-based representations (Lee & Seung, 1999). Future work could therefore use the uncovered convexity as analytic traction to understand these phenomena.
>
> **Theme 2 - Use of representational similarity - Weakness 2, Question 2.**
>
> We did not pick the dot product representational similarity matrix ad hoc. Rather, we study a set of optimisation problems and find that the dot product similarity matrix, Q, **emerges** as the right way to summarise the representation. Since we can write the optimisation problems we study entirely in terms Q, studying a kernelised version of the similarity would not provide additional benefit. In other words, while it is true that a kernelised version of similarity would detect differences between representations that our dot-product similarity matrix cannot, we use Q because our problems are invariant to such changes - demonstrated directly by writing the optimisation problems entirely as a function of Q.
>
> That said, pushing the more nonlinear versions of this theory to study optimisation problems that can be written as a function only of a kernelised form of similarity seems like an interesting direction. One step we took towards that was showing that a nonlinear similarity matching objective (O7 in appendix A.1) is convex in Q. This would form an interesting avenue for future work.
>
> Finally, the reviewer states “high RSA correlation does not ensure good single-neuron fits”. Indeed, but we show interesting results in which having the same Q matrix necessarily means the same single-neuron tuning curves for nonnegative representations, section 4, precisely quantifying the cases in which one should expect perfect single-neuron fits from perfect RSA correlation.
>
> **Theme 3 - No finite-sample stress tests or noisy-data robustness testing, or computability of tight scattering - Weakness 4 and Question 3**
>
> In fact, our theories are statements about the empirical distribution of the data, i.e. the finite and potentially noisy dataset the optimisation is performed over. Therefore, our theory is explicitly about how the structure of the empirical dataset, warts and all (i.e. finite size and noise effects), governs things like modularity.
>
> We thank the reviewer for highlighting this lack of clarity. We now invest significantly more effort in explaining the meaning of definition 3.1 and theorem 1, and in particular highlighting the role the empirical dataset plays:
>
> > Our theorem relates the empirical dataset to identifiability, so is inherently a finite-sample result.

---

> ### Author Response · Authors · 2025-11-21
> **Response (2/2)**
>
> **Theme 4 - Intractability of Completely Positive Programming - Weakness 5.**
>
> It’s true that completely positive programming is difficult – something we pointed out in the discussion. This limits the potential algorithmic benefits of using this method for optimisation. However, it does guarantee that if a local minimum is found, it must be a global minimum – which we make much use of in sections 3, 4, and 5.
>
> That said, using our work as a launchpad to develop useful algorithms could be productive. It is a mathematical oddity that up to dimension four the set of completely positive matrices is the same as positive semidefiniteness + all positive entries, permitting easy optimisation using standard convex approaches. For the general case, approximation hierarchies have been proposed; these allow you to find an approximately completely positive matrix at a degree of accuracy set by the amount of computation you expend. We now include a discussion of this point:
>
> > One alternative approach is approximating the set with a hierarchy of increasingly precise enclosing sets (Nishijima & Nakata, 2024).
>
> Thanks for your help, let us know if you have any remaining questions or concerns!

---

### Official Review · Reviewer_78Ze · 2025-10-31

**Soundness:** 4
**Presentation:** 4
**Contribution:** 3
**Rating:** 8
**Confidence:** 4

**Summary:**

This submission develops multiple theoretical contributions in the domain of representation learning, which are made possible by the observation that a class of optimization problems, which may not be convex in the representation itself, are in fact convex in the representation similarity matrix (RSM). The contributions are as follows:
- Demonstrating that non-negative similarity matching, in tandem with standard regularizers and a linear readout constraint, is a convex optimization problem in the RSM. In a similar spirit the paper discusses how this observation can be extended to describe a larger family of optimization problems.
- Leveraging this convexity result to derive necessary and sufficient conditions under which non-negative affine autoencoding admits an identifiable (up to permutation) solution. In particular this result holds for more general conditions than previous similar examples in the literature.
- Leveraging the convexity result to derive sufficient conditions under which single-unit's tuning properties are necessarily unique (i.e. only one set of tuning curves can produce an optimal RSM).

**Strengths:**

- The heart of the paper, that optimizations problems that may not be convex in a representation $\bf{Z}$ can be convex in the RSM $\bf{Q} = \bf{Z}^T \bf{Z}$, is both quite interesting, well placed in the literature of neural circuits (via the anchor of similarity matching objectives), and to my knowledge novel.
- This key observation is used to substantial effect: it allows for the generalization of previous results in matrix factorization, and can be used to derive interpretable insights into neural tuning properties (at least in a theoretical setting).
- The presentation of the work is for the most part very clear. The presentation of technical results is coherent, there is some effort to build intuition around these results (particularly in the example about when single-neuron tuning properties are identifiable), and there is a thoughtful effort to couch the contributions of this work in the broader literature (including acknowledging key limitations of the theory as it stands).

**Weaknesses:**

- Theorem 1 generalizes a previous results on problem 1 to the case of arbitrary (linear) source mixing (from orthogonal linear mixing). It is not obvious to me how significant this generalization is in practice. I.e. the value of this contribution could be made more clear if the author's outlined some real world cases where this contribution would be necessary to deliver theoretical predictions.
- Regarding the limitations of $N_{neurons} > N_{samples}$ and the assumption of perfect fitting: it was not obvious to me whether the results for factorization with orthogonal source mixing relied on a similar set of assumptions, or if these were relatively stronger or weaker than the dependencies of previous theories?
- As noted in the limitations section of the discussion, though these problems can be demonstrated to be convex in $\bf{Q}$ the payoff of this observation in terms of computational approaches may be limited by the fact that the convex set of possible similarity matrices is not well suited for use in current numerical convex optimization solvers.
- I don't have much of an intuition for the types of circumstances in which the tight scattering condition would hold. Clearly one requirement is that the level set of the relevant quadratic form is elliptical, but it would be nice to develop a clearer way of seeing when this condition holds in terms of the structure of A and S (perhaps even with some practical examples).
- Nit: Typo in problem statement 1 (~ line 237) $\bf{b}_{\text{in}} \in \mathbb{R}^{d_x}$ --> $\bf{b}_{\text{in}} \in \mathbb{R}^{d_z}$

**Questions:**

- Can the author's expand on their thought in the discussion about the limitation of perfectly fitting the training data? What are the utilities and drawbacks of constraining your view of a system to the set of points that are perfectly fit?
- Can the author's answer the question from the weaknesses section about the severity of the assumptions used to define problem 1 relative to prior work on this problem?
- The author's content their "results sound an optimistic note for classic neuroscience" but I am not sure this is necessarily the case. For example if the "true" generative model of signals encoded by neural systems do not obey the tight scattering condition (or similarly if the conditions of theorem 2 are not met) wouldn't the interpretation be reversed? It is in general not obvious to me which scenario we are in! Either way it is good to know the lay of the land from a theoretical perspective.

---

> ### Author Response · Authors · 2025-11-21
> **Response (1/2)**
>
> We thank the reviewer for their supportive and careful review. Here is our rebuttal; please let us know if you have any remaining concerns that we can help to address.
>
> **Weakness 1 - Linearly mixing is a small generalisation**
>
> We agree with the reviewer's evaluation that, as framed, this generalisation does not pack a practical punch. Indeed, our use cases are drawn from our own representational explorations, mainly in neuroscience, and as yet we have not found a use for this generalisation. That said, first, there are many many cases where using the orthogonal version would lead to incorrect predictions of modularisation/mixing. For example, if your orthogonal theory predicted that two variables should modularise, but their encoding directions were very aligned (i.e. the corresponding columns of **A** were very aligned) then the representation will often mix. More subtly the reverse can also happen, where mixing the sources leads to modular recovery. While we have not found neuroscientific use cases, this is strictly more general, and is the setting used by the matrix factorisation literature. Hence, only by making this generalisation are we able to provide a result useful for identifying matrix factorisation problems, in which the mixing matrix plays a key role, and whose uses as a model, in both neuroscience/ML and far beyond, are well known.
>
> We have substantially reformulated figure 2 to try to illustrate this case and give more intuition about the changes linearly mixing introduces. We hope this helps.
>
> **Weakness 2 and Question 2 - Link between general assumptions and those used in theorem 1**
>
> We agree that this link was not clearly established, apologies. Perfect fitting is used in theorem 1. N_neurons > N_samples (d_z > N in our notation) is actually weaker though, as stated in the preamble to problem 1, instead d_z > d_s.
>
> We use many neurons in order to derive convexity of the problem. If, however, we then study the optimisation problem and find that there is a globally optimal solution that uses fewer neurons, then it will also be the globally optimal solution under stricter neuron constraints. In this case we find that, under the tight scattering conditions, the optimal solution is modular, which can be encoded with only d_s neurons. If these conditions are broken then the modular solution will not even be a local minima, and perturbing by mixing together the existing d_s neurons, or using a new neuron formed from some mixture of new sources, will decrease the loss. Hence, in this problem we only need d_z > d_s for these conditions to govern when a modular solution is the global optima.
>
> This was not clear, we now attempt to clarify this in the new appendix B.5:
>
> > In order to show the convexity of our problems we had to assume the number of neurons was larger than the number of datapoints: $d_Z\geq T$. Restricting the number of neurons further will never improve the globally optimal solution. Therefore, if we can find a globally optimal solution to the many-neuron convex problem that uses a small number of neurons, $n$, it will also be a globally optimal solution to any version of the problem with the neuron a neuron constraint $d_Z\geq n$. This is the case in this problem: under the tight scattering conditions a modular solution using only $d_S$ neurons is optimal. Further, if the tight scattering conditions are not satisfied all we need is one extra neuron whose activity we can perturb to reduce the loss. This shows that tight scattering is still the necessary and sufficient condition for modularising when as long as $d_Z> d_S$, as stated in theorem 3.
>
> Further, we have added a sentence on this to the relevant part of the discussion:
>
> > Further, section 3 demonstrates a more bespoke workaround [the number of neuron constraint]: if the solution to the unconstrained problem uses few neurons, it will also be the solution in the neuron-constrained setting, letting us relax the neuron constraint, Appendix B.5.
>
> **Weakness 3 - Limitations of Completely Positive Programming**
>
> Indeed, as stated, this is a limitation, and instead we use our results for theoretical dividends. However, we perhaps did not sufficiently emphasise the fact that people are coming up with ways to approximate such solutions, techniques that could usefully directly be ported into this problem giving it more practical relevance. We now highlight this in the discussion:
>
> > One alternative approach is approximating the set with a hierarchy of increasingly precise enclosing sets
> (Nishijima & Nakata, 2024).

---

> ### Author Response · Authors · 2025-11-21
> **Response (2/2)**
>
> **Weakness 4 - Unclear what tight scattering is**
>
> We agree with this, and have tried to add more intuition, schematics, and examples in figure 2. We hope this helps somewhat, and include the modified text below:
>
> > Intuitively, identifiability in this model is governed by the 'spread' of the sources. Previous work using orthogonal $\textbf{A}$ showed that if the distribution of sources was 'sufficiently rectangular' the optimal representation recovers the sources (Dorrell et al. 2025). Precisely, if the convex hull of the data engulfs a particular set that can be easily calculated from the dataset it is sufficiently rectangular, fig 2A. If this condition is satisfied the optimal solution recovers the sources, else it mixes them.
>
> > Returning to arbitrary $\textbf{A}$ complicates things. If the encodings of two sources are aligned, i.e. $\textbf{A}_i^T\textbf{A}_j$ is nonzero where $\textbf{A}_i$ is the $i$th column of $\textbf{A}$, then the weight regularisation encourages the directions within $\textbf{z}^{[i]}$ that encode those sources to align. This extra effect can cause the optimal solution to mix sources when it would otherwise modularise, or even modularise when it would otherwise mix, fig 2. Our result improves the `sufficient rectangularity' of previous work by adapting it appropriately to the mixing matrix, $\textbf{A}$. In particular, we define a tight scattering condition using a quadratic form calculated from the dataset covariance and minima and the mixing gram matrix $\textbf{A}^T\textbf{A}$ which measures this adaptive rectangularity condition.
>
> **Question 1 - Perfectly fitting data assumption**
>
> We had thought that perfect fitting was required for convexity. In fact, fortunately, we were wrong. O5 in appendix A now details how the sum of a L2 reconstruction and readout weight regularisation error is in fact a convex function of the representational similarity Q. This significantly broadens the applicability of our work.
>
> We have changed all the relevant statements throughout, and removed it as a limitation from the discussion. We are currently working on the identifiability of semi-nonnegative matrix factorisation with imperfect reconstruction, the relevant generalisation of theorem 3, and will include it either next week or in the final submission.
>
> As such, even though perfect fitting doesn’t seem like the worst assumption, and might be useful at times, the choice is now entirely in the users’ hands.
>
> **Question 3 - Optimistic note for neuroscience**
>
> Certainly, when the scattering conditions are not satisfied, single neuron tuning curves will not be unique, and the cases in which that happens are interesting. The reason we think this sounds an optimistic note is because it gives relatively broad cases in which single neuron tuning curves are unique. I’ve heard people argue both that neural networks can solve a problem in so many ways that comparisons to brains are not very meaningful, and that all that matters is population behaviour, single neurons will negotiate their way around subserving the population. Instead, it is against this extreme position that our results sound an optimistic note: in a wide variety of settings both the representational similarity and the tuning are meaningfully related to problem structure. I don’t think there is disagreement with the reviewer here, simply a matter of emphasis, we now expand on this slightly in the discussion:
>
> > These results sound an optimistic note for classic neuroscience: if we can correctly frame the relevant neural computation and constraints, it seems that representation and function are tightly coupled, including often, though not always, at the single neuron level, section 4.
>
> Nit - Thank you, have fixed.
>
> Thanks for your attention!

---

### Author Response · Authors · 2025-11-21
**General Response**

We thank the reviewers for their broadly constructive and supportive reviews. Reviewers appreciated the clarity (78Ze, czPt), the technical contributions (czPT, Vz3x), and felt the findings were “elegant” (czPt), “refreshing” (Vz3x) and “used to substantial effect” (78Ze). They were also very useful in helping us to change the paper, and we think the current submission is a significant improvement. We detail the three largest changes:

**1) Relevance**

Concerns were raised over the relevance of our results to… anything really, but in particular neuroscience. We have attempted to alleviate these concerns with not one, but two blockbuster neuroscience applications.

In an extension of figure 3 we apply our neural identifiability conditions to grid cells, finding that grid cells from different modules are identifiable as long as their wavevectors are not integer multiples of one another. This matches the finding that in entorhinal cortex module frequencies are not integer multiples of one another.

Then we introduce an additional figure using of our convex, **nonlinear** models to understand a coding puzzle. Retinal neurons display ON-OFF splitting, in which a single variable is encoded in a pair of oppositely rectified neurons (Euler et al. 2014). This splitting has been understood as an efficiency: splitting the stimulus reduces the range of each neuron, saving energy (Sterling & Laughlin, 2015; Gjorgjieva et al. 2014). But not all variables are ON-OFF coded and existing theories cannot explain what governs this. We show that the transition between ON-OFF and pure coding is driven by the variable's sparsity, matching conjecture (Sterling & Laughlin, 2015).

We have certainly found these results useful for our own investigations, and we hope these justify that.

**2) Poorly explained maths**

Reviewers highlighted the opaqueness of the mathematical results, with which we largely agree. We therefore invested significant effort in clarifying these. We have completely remade figure 2, which now includes two schematics, and two numerical examples, that serve to illustrate tight scattering, and how it differs from the literature. In particular, we show how including the mixing matrix $\textbf{A}$ can shift identifiable sources into non-identifiability, and, more surprisingly, a non-identifiable set of sources into identifiability, while also showcasing the precision of our mathematical results.

Further, we have included significant additional text to provide intuition for these results:

> Intuitively, identifiability in this model is governed by the ‘spread' of the sources. Previous work using orthogonal $\textbf{A}$ showed that if the distribution of sources was ‘sufficiently rectangular' the optimal representation recovers the sources (Dorrell et al. 2025). Precisely, if the convex hull of the data engulfs a particular set that can be easily calculated from the dataset it is sufficiently rectangular, fig 2A. If this condition is satisfied the optimal solution recovers the sources, else it mixes them.

> Returning to arbitrary $\textbf{A}$ complicates things. If the encodings of two sources are aligned, i.e. $\textbf{A}_i^T\textbf{A}_j$ is nonzero where $\textbf{A}_i$ is the $i$th column of $\textbf{A}$, then the weight regularisation encourages the directions within $\textbf{z}^{[i]}$ that encode those sources to align. This extra effect can cause the optimal solution to mix sources when it would otherwise modularise, or even modularise when it would otherwise mix, fig 2. Our result improves the `sufficient rectangularity' of previous work by adapting it appropriately to the mixing matrix, $\textbf{A}$. In particular, we define a tight scattering condition using a quadratic form calculated from the dataset covariance and minima and the mixing gram matrix $\textbf{A}^T\textbf{A}$ which measures this adaptive rectangularity condition.

**3) New and improved maths**

Finally, there have been two large changes in the mathematical appendices. First, we spotted an error in our convexity proofs (points O4 and C3 in appendix A) - we’d implicitly assumed an orthogonality that didn’t exist - so those have been corrected to show the same claims without the mistake.

Second, more excitingly, prompted by reviewers we re-examined our perfect fitting assumption. We thought this assumption was necessary for convexity. In fact, though one of the terms in the resulting loss is indeed non-convex, it cancels, and the loss as a whole is convex! This is very good news, as it broadens the applicability of our results to cases where the network does not perfectly fit the data, the standard setting.

We are currently working on an extension of theorem 1 to the case of imperfectly fitting data, and will include it either soon, or upon publication if it is accepted.

More broadly, we fixed mistakes throughout, made the writing clearer, and modified the discussion substantially. We hope you agree the paper is much improved.

---

### Author Response · Authors · 2025-12-02
**Last Update**

We include one final version of the paper in which, as promised, a new appendix (C) describes the same identifiability conditions for the case of imperfect reconstruction - a significant and exciting generalisation of our theory!

---

### Meta-Review · Area_Chair_wRqC · 2026-01-05

**Summary:**

This work is focused on developing a general optimization framework for neural coding. The framework focuses on a specific optimization formulation that is convex (by optimizing over inner products of the factors) with linear constraints. While the optimization itself is well known, the authors extend this framework by providing theoretical guarantees on recoverability through a property they term the "tight scattering". The theory is extended to a number of small-scale simulations to validate their formulation.

Conceptually the method is very similar in flavor to guarantees building on, for example, Gaussian Widths (e.g., work by Recht or Tropp). It also has some semblance to linear kernelized methods and "lifting" by working with the inner products. That said the results do seem novel (including the tight scattering definition which I have not seen before) and build on these ideas in to the neural coding space. One main concern was one of novelty (raised by reviewerTVF6), which I think is reasonably addressed in the response and revision. There was also the question of nonlinear and non-L2 costs (sparsity, neural nets) which the authors partially address. I agree for example that the cost function presented can be similar to sparse coding by changing the L2 norm over z to an L1 norm, but changing the theory might be highly nontrivial. Moving from recovery guarantees over least squares to guarantees over LASSO requires significant changes in the mathematical machinery. Moreover the nonlinear networks seem to be restricted to ReLus, which is not too different than linear with a non-negativity constraint.

That said, the overall reception of the work was positive with one exception (Reviewer TVF6). Given the initial evaluation and responses, I think this work is above bar.

**Reviewer Concerns:**

Most of the reviewer comments were minor, with the more major issues raised by Reviewer TVF6. The major comments related to:
 1) Novelty of the formulation: Optimization driven neural coding is by no means new or unique. Several models seemed to have been missing in the original formulation, such as non-negative matrix factorization and sparse coding
 2) Sufficiency of the experiments: the experimental results were mostly focused on smaller simulated datasets.
 3) Applicability to nonlinear models
 4) Clarity of modeling choices, such as the use of the Frobenius norm in the cost function or the choice that the numner of neurons exceed the number of data points.

The authors responded, clarifying the relationship with other models and parameter choices. I found the responses to most of these convincing, in particular the clarifications on the theory. I did think that the assertion that sparse coding and nonlinear models were easily captured by the theory to be a bit of an over-reach (see the Summary section), which might limit some of the extended impact of the work. Thus I consider these partially outstanding, as they would have required additional derivations to show that the theory can actually extend to these cases.

**Reviewer Scores:**

The initial scores for this paper were 8,8,6,2. While there were not any responses (aside from one of the reviewers who provided an 8 affirming their score), I do not think that the scores would have changed.

---

> ### Public Comment · ~Will_Dorrell1 · 2026-03-05
> **Brief Response to AreaChair for Posterity**
>
> We thank the Area Chair for their broadly positive review of our work. We largely agree with it, including the summary of the shortcoming that moving to other models, e.g. L1, will require more maths. However, we would like to draw attention to one point where we disagree, in case other readers find their way here.
>
> The area chair states:
>
> "Moreover the nonlinear networks seem to be restricted to ReLUs, which is not too different than linear with a non-negativity constraint."
>
> This is not true! We parameterise our representations arbitrarily via an unrestrained (beyond nonnegativity) data matrix. The optimal neurons could learn any nonnegative function of the data. Interestingly, we analytically find that a ReLU tuning curve is optimal in the problem setting we study - a retrojustification of the ReLU!

---

### Decision · Program_Chairs · 2026-01-26

Accept (Poster)